# Flexible, Efficient, and Stable Adversarial Attacks on Machine Unlearning

**Zihan Zhou**[1]   **Yang Zhou**[1]   **Zijie Zhang**[2]   **Lingjuan Lyu**[3]   **Da Yan**[4]   **Ruoming Jin**[5]   **Dejing Dou**[6,7]

## Abstract

Machine unlearning (MU) aims to remove the influence of specific data points from trained models, enhancing compliance with privacy regulations. However, the vulnerability of basic MU models to malicious unlearning requests in adversarial learning environments has been largely overlooked. Existing adversarial MU attacks suffer from three key limitations: inflexibility due to pre-defined attack targets, inefficiency in handling multiple attack requests, and instability caused by non-convex loss functions. To address these challenges, we propose a Flexible, Efficient, and Stable Attack (DDPA). First, leveraging Carathéodory's theorem, we introduce a convex polyhedral approximation to identify points in the loss landscape where convexity approximately holds, ensuring stable attack performance. Second, inspired by simplex theory and John's theorem, we develop a regular simplex detection technique that maximizes coverage over the parameter space, improving attack flexibility and efficiency. We theoretically derive the proportion of the effective parameter space occupied by the constructed simplex. We evaluate the attack success rate of our DDPA method on real datasets against state-of-the-art machine unlearning attack methods. Our source code is available at https://github.com/zzz0134/DDPA.

## 1. Introduction

Machine unlearning (MU) aims to give data holders the right to remove the influence of a certain subset of data from a trained machine learning (ML) model, while maintaining the accuracy of the ML model on remaining data (Garg et al., 2020a; Gupta et al., 2021; Nguyen et al., 2022; Wu et al., 2022). Although MU research have attracted significant attention for their ability to protect the right to be forgotten, most of existing studies focus on the improvement of effectiveness and efficiency of MU algorithms (Chowdhury et al., 2024; Aldaghri et al., 2020; Yan et al., 2022; Kumar et al., 2022; Dukler et al., 2023; Golatkar et al., 2023; Pratama & Gambetta, 2024; Yang et al., 2024).

Despite achieving remarkable performance, recent studies have shown that basic MU models are vulnerable to malicious unlearning (i.e., data removal) requests during the unlearning process in adversarial settings (Liu et al., 2024e; Di et al., 2024; ZHAO et al., 2023; Zhang et al., 2023; Huang et al., 2024b; Ma et al., 2024; Shin & Park, 2024; Zhao et al., 2024; Hu et al., 2024). An attacker can inject some carefully-designed data samples to the training dataset such that the MU model behaves benign without impact on the model prediction. Afterwards, the attacker submits a unlearning request to remove the perturbed data samples, so as to negatively affect the prediction of MU models (Liu et al., 2024e; Di et al., 2024; ZHAO et al., 2023; Shin & Park, 2024).

Current mainstream research in adversarial attacks on MU concentrates on target attacks to degrade the performance of MU models, including misclassifying specific data samples (Liu et al., 2024e; Di et al., 2024; Zhang et al., 2023; Huang et al., 2024b; Ma et al., 2024; Shin & Park, 2024) and misclassifying data samples into a specific class (ZHAO et al., 2023; Hu et al., 2024). Nevertheless, three critical challenges remain open: (1) Flexibility. These approaches require to know which attack targets before the data poisoning process. That is, these attacks are attack target-specific: the malicious unlearning requests regarding the perturbed data samples that are related to specific attack targets are only effective for specific data samples or specific class. This flexibility concern dramatically limits the applicability of such attacks in real scenarios; (2) Efficiency. In practice, a large ML model like Stable Diffusion (Rombach et al., 2022) often faces the arrival of a series of MU requests with different attack targets. In this case, the attacker need to sequentially redo the data poisoning operations and the attack processes one by one to adapt to diverse attack targets, resulting in non-trivial computational costs; and (3) Stability. The non-convexity of loss functions in ML and

---

[1]Auburn University, USA [2]University of Texas at San Antonio, USA [3]Sony AI, Japan [4]Indiana University Bloomington, USA [5]Kent State University, USA [6]Fudan University, China [7]BEDI Cloud, China. Correspondence to: Yang Zhou <yangzhou@auburn.edu>.

*Proceedings of the 42nd International Conference on Machine Learning*, Vancouver, Canada. PMLR 267, 2025. Copyright 2025 by the author(s).

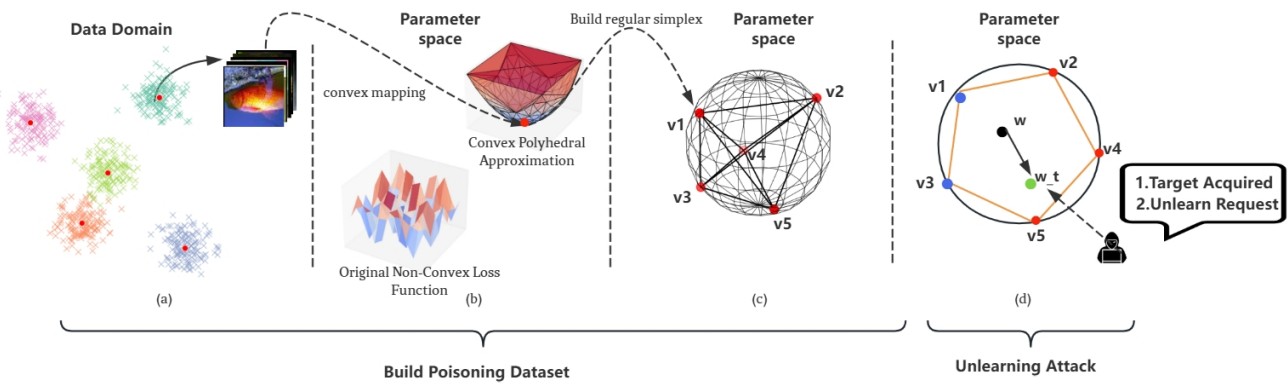

Figure 1: Attack workflow of our DDPA attack.

MU models poses significant challenges for existing MU attack methods, as they are vulnerable to the pitfalls of local minima of model parameters. These local minima can significantly degrade the performance of MU models and thus cause large deviations from the intended attack outcomes, leading to MU attack failure.

To our best knowledge, this work is the first to conduct the problem of adversarial attacks on machine unlearning, while holding the attack target-agnostic property, supports on-demand attacks to attack arbitrary targets upon attackers' demands and quickly responding to a series of MU attack requests after the MU models are deployed, and maintaining the stability of MU attacks, by leveraging the theory of thrust vector control, Simplex Geometry, and convex polyhedron.

Thrust vector control is a technique widely used in aerospace engineering that an aircraft or rocket manipulates the direction of the thrust from its engines to control the attitude or angular velocity of the vehicle (Praveen et al., 2023). This motivates us to establish a connection between thrust vector control for moving the aircraft and rocket towards arbitrary locations and thrust vector control for moving the parameter of attacked MU models towards parameters corresponding to any attack targets upon attackers' demands.

First, following (Zhang et al., 2023), given some clean instance, we randomly select multiple data samples from the clean instance as the initial group centers $V$. In order to tackle the instability issue of MU attacks due to the non-convexity of loss functions, we propose a convex polyhedral approximation method to transform the original non-convex loss function into a convex version. Notice that directly enforcing the convexity on the original loss function is impractical because the model owner has no justification for accommodating user requests to modify the model parameters and loss functions. Next, we model the group centers whose neighborhood is near convex on the original loss function, i.e., the group centers making the distance between the original and convex loss functions the smallest, as thrust

points in the thrust vector control. Since the neighborhood of these thrust points is near convex, finding the optimal parameter (near global minima of model parameters) is deterministic due to the MU, which consistently moves the unlearning gradients toward this minimum and ensures the stability of MU attacks. We theoretically derive the solution of convex polyhedral approximation through constrained optimization.

Second, by utilizing the conjugate algorithm (Ly et al., 2017), the thrust points are mapped to the corresponding parameters in parameter space. Due to the convexity of the neighborhood of the thrust points, the mapping between the thrust points and the corresponding parameters is a one-to-one mapping, since the optimal parameter in the parameter space is unique under the convexity condition. We then model the corresponding parameters in parameter space as thrust vectors in the thrust vector control. Following this mapping, an effective simplex detection technique is proposed to build a maximum regular simplex with thrust vectors as vertices. The maximum regular simplex is able to cover the parameter space as much as possible, which allows the model parameter to be moved towards parameters corresponding to any attack targets upon attackers' demands. Like multiple thrusts from the engines on a aircraft or rocket can be adjusted to steer the vehicle in any direction, based on the built maximum regular simplex, by unlearning one or more thrust vectors in parameter space, the model parameter can be moved to handle the parameter change due to arbitrary MU attack requests. We theoretically validate the regularity property of the built maximum regular simplex as well as the coverage of the simplex to the parameter space.

Figure 1 exhibits the workflow of our proposed MU attack algorithm, DDPA, with two main stages: poisoned dataset construction and MU attacks. In the first stage, Figure 1 (a) represents the random selection of multiple data samples from the clean instance as the initial group centers. In Figure 1 (b), the group centers whose neighborhood is near

convex are selected as the thrust points (red points in Figure 1 (a)) through the convex polyhedral approximation method. In Figure 1 (c), utilizing the conjugate algorithm (Ly et al., 2017), the thrust points in the poisoning dataset are mapped to the thrust vectors (red vertices $v_1$-$v_5$ in Figure 1 (c)) in parameter space via the conjugate algorithm. Around these group centers, data samples are generated within the MU budget using a predefined distribution, such as a Gaussian distribution (More & Wolkersdorfer, 2023; Oymak & Soltanolkotabi, 2021), to construct the poisoning dataset. In the second stage, in Figure 1(d), giving current parameter $w$, the attacker aims to attack a specific attack target by moving $w$ towards a target parameter $w_t$. The deviation from $w$ to $w_t$ is $\Delta w = w_t - w$. To move $w$ to $w_t$, the attacker requests the data removal with the direction $-\Delta w$ by manipulating the thrust vectors (blue vertices $v_1$ and $v_3$) to control the direction of the vehicle towards $w_t$. After addressing this malicious unlearning requests, the attacker can continue to move the parameter from $w_t$ to others in the parameter space in response to one or more unlearning requests, without sequentially redoing the data poisoning operations and the attack processes one by one.

In comparison with existing MU attack techniques, our DDPA method exhibits three compelling advantages: (1) It supports the target-agnostic MU attacks by manipulating one or more thrust vectors to move the parameter towards any direction; (2) It provides the timely response to a series of MU attack requests, as long as the amount of data removals is below the MU budget; and (3) It ensuring the stability of MU attacks based on convex approximation and optimization. Empirical evaluation on real datasets demonstrates the superior performance of our DDPA MU attack model against several state-of-the-art methods on image classification. More experiments, implementation details, and hyperparameter setting are presented in Appendices F.

## 2. Background

### 2.1. Machine Unlearning

Machine unlearning is a process designed to ensure that the influence of specific data points is effectively removed from a trained model. Formally, given a training dataset of N samples $D = \{x_i, y_i\}_{i=1}^N$, where each data point $x_i \in \mathbb{R}^d$ is associated with a label $y_i \in \{1, 2, \ldots, Y\}$, where Y is the number of classes. A classification model $M(D)$ is trained on the complete dataset $D$. Machine unlearning aims to remove the influence of a subset of data $D_u \subset D$, referred to as the forgotten data, such that the updated model behaves as if $D_u$ were never part of the training process.

When a data removal request is submitted, the dataset is conceptually split into $D_u$, the data to be forgotten, and $D_r$, the data to be retained, where $D = D_u \cup D_r$. The goal

of machine unlearning is to ensure that the model obtained after forgetting specific data has a probability distribution equivalent to a model trained without those data points. This can be expressed as:

$$Pr(D \setminus x_f) = Pr(D(X \setminus x_f; Y)) \tag{1}$$

A straightforward approach to achieve unlearning is to retrain a new classification model $M_r(D_r)$ from scratch using only $D_r$. This method ensures exact unlearning by completely removing the influence of $D_u$ from the model. However, retraining is computationally intensive and impractical for large-scale datasets and modern deep learning architectures, making it an inefficient solution.

To address these limitations, efficient algorithms aim to approximate the retrained model $M_r(D_r)$ directly from the deployed model $M(D)$. The objective is to produce a sanitized model $M_u(D, D_u, M)$ that eliminates the influence of $D_u$ while leveraging the existing model $M(D)$, thus avoiding the need for full retraining. By modifying $M(D)$ to remove the impact of $D_u$, this approach significantly reduces computational costs while achieving the desired unlearning outcome.

### 2.2. Poisoning-Based Backdoor Attacks

Poisoning-based backdoor attacks inject maliciously crafted data into the training process to implant hidden backdoors in machine learning models. Consider a training dataset $D = \{x_i, y_i\}_{i=1}^N$, where each $x_i \in \mathbb{R}^d$ is associated with a label $y_i$. The attacker introduces a poisoned subset $D_p = \{(x_p, y_t)\}$, where $x_p$ contains a trigger, and $y_t$ is the target label. The poisoned dataset becomes $D' = D \cup D_p$. The model $M(D')$ is trained to behave normally on clean inputs $x_b$, such that:

$$M(x_b) = y_b, \quad \forall (x_b, y_b) \in D, \tag{2}$$

while misclassifying inputs containing the trigger $x_t$:

$$M(x_t) = y_t, \quad \forall x_t \text{ containing the trigger.} \tag{3}$$

These attacks are stealthy and adaptable, as the poisoned samples are often indistinguishable from benign data. Recent techniques have further enhanced the stealth and robustness of triggers, making poisoning-based backdoor attacks a critical challenge for secure machine learning.

## 3. Dynamic Delayed Poisoning Attack

### 3.1. Threat Model

During training at time $t_0$, the model learns from a clean dataset $D_{c1}$, while the attacker injects a malicious dataset $D_p$. Training continues with additional clean data $D_{c2}$. At $t_1 > t_0$, the attacker submits an unlearning request targeting

$D_u \subset D_p$, thereby manipulating the model parameters $w$ toward a desired state $w_t$. The target $w_t$ is determined by searching the neighborhood of $w$, ensuring the model exhibits the intended adversarial behavior post-unlearning.

**No access to training data.** The adversary has no direct access to the training data beyond their injected poison samples. This reflects realistic scenarios, such as collaborative open-source projects where users contribute data. In such cases, an adversary can subtly introduce poisoned samples that remain indistinguishable from benign data. Through unlearning requests, these poisoned samples are strategically removed, activating the attack and compromising the model's integrity.

**Knowledge.** The attacker has access to the model architecture, training process, loss function, and parameters at a checkpoint $t_1$. This assumption is practical, as many machine learning models are built using well-documented architectures, open-source libraries, or pre-trained models. Additionally, APIs for model fine-tuning and interaction often expose certain model behaviors, which attackers can exploit.

**Real-word scenarios.** Complying with modern privacy regulations, such as the European Union's General Data Protection Regulation (GDPR) (Voigt & Bussche, 2017) and the California Consumer Privacy Act (CCPA) (Pardau, 2018), MU aims to give data holders the right to remove the influence of a certain subset of data from a trained ML model. For example, Stability AI announced it would allow artists to remove their work from the training dataset for the Stable Diffusion 3.0 release (Staff, 2022). Such requirements are increasingly relevant in settings like online learning (Hoi et al., 2018) and continual learning (Wang et al., 2024), where models are incrementally updated over time.

### 3.2. Thrust-Driven Parameter Manipulation via Simplex Geometry

To address the two key limitations of existing MU attack methods mentioned in Section 1, namely inflexibility due to pre-defined attack targets and inefficiency in handling multiple attack requests, we propose an effective simplex detection technique based on John's theorem (Lasserre, 2014). This technique constructs a maximal regular simplex using thrust vectors (group centers $V$) as vertices, ensuring that the simplex provides maximal coverage of the parameter space. This property allows model parameters to be dynamically guided toward those corresponding to any attack target specified by the attacker. By unlearning one or more thrust vectors within the parameter space, the model parameters can be adjusted to accommodate the shifts resulting from arbitrary MU attack requests. Additionally, we provide theoretical validation of the regularity of the con-

structed maximal regular simplex and its parameter space coverage, demonstrating the robustness and effectiveness of the proposed method.

A regular $n$-simplex (Dirksen, 2015) is an $n$-dimensional regular polytope with $n + 1$ vertices, where each pair of vertices is connected by an edge. To formally define the simplex and its geometric properties, we present the following:

**Definition 3.1** (Simplex). A $(n - 1)$-simplex $S_{n-1}$ is the convex hull of $n$ affinely independent group centers $v_1, v_2, \ldots, v_n \in \mathbb{R}^n$. It is defined as:

$$S_{n-1} = \left\{ s \mid s = \sum_{i=1}^{K} \lambda_i v_i, \ \sum_{i=1}^{K} \lambda_i = 1, \ \lambda_i \geq 0, \ \forall i \right\} \quad (4)$$

where $n$ represents the number of vertices in the simplex. These conditions ensure that $S_{n-1}$ is a compact and convex subset of $\mathbb{R}^n$.

**Definition 3.2.** [Regular $n-1$ Simplex] A set of $n$ group centers $\{\mathbf{v}_1, \mathbf{v}_2, \ldots, \mathbf{v}_n\} \subset \mathbb{R}^{n-1}$ forms a regular $n-1$ simplex if and only if: (1) **Centroid Condition:** The centroid of the points is at the origin, $\sum_{i=1}^{n} \mathbf{v}_i = \mathbf{0}$. (2) **Equidistant Condition:** The squared Euclidean distance between any two distinct points is constant, $\|\mathbf{v}_i - \mathbf{v}_j\|^2 = d^2, \ \forall i \neq j$. (3) **Inner Product Symmetry:** The inner product between any two distinct points is constant, $\mathbf{v}_i \cdot \mathbf{v}_j = -\frac{d^2}{n-1}, \ \forall i \neq j$. These conditions ensure that the points are symmetrically distributed in $n - 1$ dimensions, forming a regular simplex.

**Definition 3.3.** [John's Theorem] Let $K$ be a convex body in $\mathbb{R}^n$. John's Theorem states that $K$ contains a unique ellipsoid of maximal volume, denoted as $B_2^n$ (the Euclidean ball of unit radius), if and only if the following conditions are satisfied:

- $B_2^n \subseteq K$.

- There exist Euclidean unit vectors $(v_i)_{i=1}^{n}$ on the boundary of $K$ and positive coefficients $(c_i)_{i=1}^{n}$ such that:

$$\sum_{i=1}^{n} c_i v_i = 0, \text{ and } \sum_{i=1}^{n} c_i \langle x, v_i \rangle^2 = \|x\|^2, \forall x \in \mathbb{R}^n. \quad (5)$$

These conditions ensure the uniqueness of the maximal-volume ellipsoid within $K$, providing a geometric characterization of $K$ through its boundary points.

The following theoretical analysis quantifies the correctness and applicability of our poisoning dataset construction based on the regular simplex. Definitions 3.1–3.3 provide the necessary foundations for the theoretical proofs. Theorems 3.1 and 3.2 establish that the John ellipsoid of a regular simplex is its inscribed sphere. For any simplex in $\mathbb{R}^n$, the simplex is regular if and only if its John ellipsoid is the unit ball $B_2^n$. Consequently, Theorem 3.3 determines whether multiple

group centers form a regular simplex and quantifies the degree of regularity of the simplex. Theorem 3.4 measures the proportion of the effective parameter space occupied by the constructed simplex. Please refer to Appendix C for detailed proof of Theorem 3.1 - 3.4

**Theorem 3.1.** *The John ellipsoid of a regular simplex is its inscribed ball. Let $A$ be a regular simplex in $\mathbb{R}^n$ with vertices $\{A_1, A_2, \ldots, A_{n+1}\}$ and $B_2^n$ as its inscribed ball. Denote by $\{B_i, i = 1, \ldots, n+1\}$ the tangent points which is opposite to $\{A_i, i = 1, \ldots, n+1\}$ respectivelyy. For positive weights $c_i = \frac{n}{n+1}$ $(i = 1, \ldots, n+1)$, the barycentric sum satisfies:*

$$\sum_{i=1}^{n+1} c_i B_i = \left( \frac{1}{n+1}, \ldots, \frac{1}{n+1} \right) \tag{6}$$

*Additionally, the solution to the representation of any point $x \in \mathbb{R}^n$ in the simplex is:*

$$\alpha = \left( \frac{n}{n+1} \langle u_1, x \rangle, \ldots, \frac{n}{n+1} \langle u_{n+1}, x \rangle \right) \tag{7}$$

*where $\{u_1, \ldots, u_{n+1}\}$ are unit normal vectors.*

**Theorem 3.2.** *A simplex $S$ in $\mathbb{R}^n$ is regular if and only if its John ellipsoid is the unit ball $B_2^n$. If $B_2^n$ is the John ellipsoid of $S$, then $B_2^n$ must be tangent to each face $F_i$ of $S$. Conversely, if $S$ is a regular simplex, its John ellipsoid is necessarily $B_2^n$, as the regularity ensures symmetry and equal tangency conditions.*

*The volume of $S$ satisfies:*

$$Vol(S) = \frac{\sqrt{n^n (n+1)^{n+1}}}{n!} \tag{8}$$

*which is the exact volume of a regular simplex with its inscribed ball being $B_2^n$. Furthermore, for the unit normal vectors $\{v_i\}_{i=1}^{n+1}$ corresponding to the faces of $C$, the inner product between any two distinct vectors satisfies:*

$$\langle v_i, v_j \rangle = -\frac{n+1}{n^2}, \quad i \neq j \tag{9}$$

*This establishes that the regularity of a simplex is directly characterized by the tangency, volume, and inner product properties of its John ellipsoid.*

**Theorem 3.3.** *A regular simplex can be quantified by how closely it satisfies the conditions of the John ellipsoid. For a given set of group centers $V = \{v_1, v_2, \ldots, v_n\}$, let $I$ denote the identity matrix, $v_i$ represent individual group centers, and $T$ denote the total number of group centers. Define the regularity measure based on the group centers $V$ as:*

$$\phi(V) = \frac{1}{T} \sum_{t=1}^{T} \exp \left( -\frac{\left\| I - \sum_{i=1}^n c_i v_i \otimes v_i \right\|^2}{2\sigma^2} \right) \tag{10}$$

*where $\sigma^2 = \frac{1}{2\pi}$ is the variance of the Gaussian function.*

The value of $\phi(V)$ lies in the range $[0, 1]$, where $\tau(V) = 1$ indicates that the simplex is perfectly regular, satisfying all the conditions of the John ellipsoid. A value closer to $0$ reflects higher irregularity due to deviations in the summation conditions.

To ensure the simplex effectively covers the parameter space and provides precise control over model parameters, it must satisfy two criteria: (1) the simplex should be as large as possible, and (2) its centroid should be positioned at the origin. These properties enable the simplex to efficiently and effectively manipulate the parameters in diverse directions.

We achieve these goals by optimizing the following objective function:

$$\mathbb{E}_{v_1, v_2 \sim V} \|v_1 - v_2\| + \frac{1}{|V|} \sum_{i=1}^n v_i, \tag{11}$$

where $V = \{v_1, v_2, \ldots, v_n\}$ represents the set of group centers. The first term maximizes the distance between randomly sampled pairs of group centers $v_1$ and $v_2$, ensuring the simplex is as large as possible. The second term minimizes the sum of the group centers' distances to the origin, ensuring the centroid of the simplex lies at the origin.

Thus, our overall optimization objective is defined as:

$$L_s = -\phi(V) - \mathbb{E}_{v_1, v_2 \sim V} \|v_1 - v_2\| + \frac{1}{|V|} \sum_{i=1}^n v_i, \tag{12}$$

where $\phi(V)$ quantifies the regularity of the simplex formed by the group centers $V = \{v_1, v_2, \ldots, v_n\}$.

Due to the lack of access to the training data, we cannot determine the size of the effective parameter space beforehand. However, through theoretical analysis, the following theorem quantifies the proportion of the effective parameter space occupied by the constructed simplex.

**Theorem 3.4.** *Following (Li et al., 2015; Pearce et al., 2020; de G. Matthews et al., 2018), assume the parameter space follows a Gaussian distribution in $\mathbb{R}^d$. The proportion of the effective parameter space (PS) occupied by the constructed $(n-1)$-dimensional simplex (S)is:*

$$\rho = \frac{Vol(S)}{Vol(EPS)} = \frac{\sqrt{n+1} \Gamma \left( \frac{m}{2} + 1 \right)}{(n-1)! \, l \, \pi^{m/2} \sqrt{2} \sigma^m} \tag{13}$$

*where $\Gamma$ is the Gamma function, $\sigma$ represents the standard deviation of the Gaussian distribution defining the spread of the parameter space, $m$ is the dimensionality of the Gaussian parameter space, and $l$ is the edge length of the constructed simplex.*

### 3.3. Handling Instability in Non-Convex Loss via Convex Polyhedral Approximation

The non-convexity of the loss function in training models poses a fundamental challenge for existing machine unlearning attack methods, as they are all susceptible to local

minima, which can lead to deviations from the intended attack outcomes and significantly reduce attack success rates. In our scenario, this issue is particularly critical, as it introduces instability in the behavior of the constructed thrust vectors (group centers), making parameter updates unpredictable and difficult to control. In contrast, if the loss function $L(f(w, x), y)$ were convex, it would possess a unique minimum for each data sample $x$, ensuring that unlearning gradients consistently converge toward the intended update direction. However, directly enforcing convexity in the original loss function is infeasible, as it would severely degrade the model's performance, and model owners have no incentive to modify their model structure to accommodate external requests.

To mitigate instability, we propose we propose a convex polyhedral approximation method that transforms the original non-convex loss function $L_d(w)$ into its convex counterpart $L'_d(w)$. Our objective is to identify multiple group centers in the neighborhood of the original loss function that exhibit the smallest deviation from its convex approximation, ensuring that they are as close as possible to the convex regions of $L_d(w)$ These group centers act as thrust points in thrust vector control, guiding parameter updates during unlearning. Since these thrust points lie in regions where the loss function is nearly convex, the optimization process under machine unlearning (MU) becomes more predictable. This allows the unlearning gradients to reliably push the model parameters toward an optimal state, ensuring a stable and effective MU attack.

By adopting this approach, we carefully position the thrust vectors (group centers) at these optimized data points. This ensures stable and predictable behavior of the thrust vectors, enabling precise manipulation of model parameters and significantly enhancing attack effectiveness.

**Definition 3.4** (Convex Polyhedron of a Function). The convex polyhedron of a function $L_d(w)$ is defined as the intersection of all half-spaces in $\mathbb{R}^n$ that lie above the graph of $L_d(w)$. Formally, for a function $L_d : \mathbb{R}^n \to \mathbb{R}$, the convex polyhedron $P$ is given by:

$$
\begin{aligned}
P = \{(w, z) \in \mathbb{R}^n \times \mathbb{R} \mid \\
z \geq L_d(w) + \nabla L_d(w)^\top (w' - w), \forall w' \in \mathbb{R}^n\}
\end{aligned}
\tag{14}
$$

where $z$ represents the vertical coordinate in $\mathbb{R}^{n+1}$, and the inequality ensures that $P$ captures the convex hull of $L_d(w)$. This formulation represents the set of points lying above the epigraph of $L_d(w)$ in $\mathbb{R}^{n+1}$, forming a convex polyhedron.

**Definition 3.5** (Caratheodory's theorem). Given a polytope $P \subset \mathbb{R}^n$ and a lower semi-continuous function $L_d(w)$, the convex envelope of $L_d(w)$ at a point $w \in P$ is defined as :

$$
\begin{aligned}
\text{Conv}_{L,P}(\mathbf{w}) = \min \Bigg\{ \sum_{i=1}^{n+1} \lambda_i L(\mathbf{Q}_i) : \mathbf{Q}_i \in P, \\
i = 1, \ldots, n+1, \sum_{i=1}^{n+1} \lambda_i = 1, \sum_{i=1}^{n+1} \lambda_i \mathbf{Q}_i = w, \lambda_i \geq 0 \Bigg\}
\end{aligned}
\tag{15}
$$

Here, $\lambda_i$ represents the barycentric coordinates associated with the points $\mathbf{Q}_i$, ensuring that the convex combination satisfies the conditions of $w$ being in $P$.

Convex polyhedral envelopes are not necessarily differentiable at optimal points, making gradient-based optimization infeasible. Moreover, finding the convex envelope of a general function $g$ over a region $P$ is computationally challenging. It has been proven (Guo et al., 2023) that determining the convex envelope of multilinear functions over the unit hypercube is an NP-hard problem.

To address this, we adopt a pointwise supremum approach based on underestimating affine functions of $g$ over $P$. The convex envelope at any point is determined by solving a constrained optimization problem.

**Theorem 3.5.** *Let $g$ be a convex function defined over a polytope $P$, and let $(x_0, y_0) \in P$ be a reference point. The convex envelope of $g$ at $(x_0, y_0)$ is the solution to the optimization problem:*

$$
\begin{aligned}
L'_d(w) = &Conv_{g,P}(x_0, y_0) = \max c, \quad \text{subject to:} \\
&g(x_i, y_i) - [a(x_i - x_0) + b(y_i - y_0) + c] \geq 0, \\
&\min_{x \in [x_j^1, x_j^2]} [g(x, m_j x + q_j) - a(x - x_0) - \\
&b(m_j x + q_j - y_0) - c] \geq 0, \\
&\forall e_j \in E'(P), \forall (x_i, y_i) \in V'(P)
\end{aligned}
\tag{16}
$$

*where $V'(P) \subseteq V(P)$ denotes the subset of vertices of $P$ that do not belong to edges in $E'(P)$, and $E'(P)$ represents the set of edges of $P$ along which $L(x, y)$ is strictly convex.*

*In the vertex constraints, $L(x_i, y_i)$ ensures that the affine function defined by $a$, $b$, and $c$ underestimates $L(x, y)$ at each vertex. In the edge constraints, $L(x, m_j x + q_j)$ enforces the underestimation of $L(x, y)$ along each edge $e_j$, where $m_j$ and $q_j$ describe the edge as a linear function $y = m_j x + q_j$. The strict convexity of $f$ ensures that the minimum value over an edge occurs either at its endpoints or at a critical point within the interval.*

Thus, our optimization objective is defined as:

$$
L_t = \sum_{x \in G} \max_{w \in S_V} \left| L_x(w) - L'_x(w) \right|
\tag{17}
$$

where $L_x(w)$ represents the original loss function parameterized by the data sample $x$, and $L'_x(w)$ denotes its convex approximation. Here, $S_V$ is the polytope formed using $V$ as the vertices, and $G$ is the set of the group centers.

This formulation ensures that we identify a set of data samples $d$ such that $L_d(w)$ approximates a convex function as closely as possible. To ensure the model behaves predictably during unlearning, the group centers are carefully positioned on these optimized data points. This placement reduces instability caused by the non-convexity of the loss

function and ensures the parameters are adjusted smoothly and effectively throughout the process.

By assembling different pieces together, we provide the pseudo code of our DDPA method in Algorithm 1 in Appendix D.

## 4. Experimental Evaluation

In this section, we evaluate the effectiveness of our Dynamic Delayed Poisoning Attack (DDPA) method compared to several state-of-the-art machine unlearning robustness attack methods. Through comprehensive experiments on multiple representative classification tasks, we demonstrate that DDPA achieves a significantly higher attack success rate across various settings. Please refer to the appendixF for detailed experimental settings and additional results.

**Datasets and Models.** We conduct experiments on two widely-used image classification datasets and one sentiment classification dataset: CIFAR-100 (Krizhevsky, 2009), Tiny ImageNet (Le & Yang, 2015), and SST-2 (Socher et al., 2013). The classifiers are trained on their respective training sets and evaluated on their test sets. For CIFAR-100, we use the VGG16 (Simonyan & Zisserman, 2015) model for image classification. On Tiny ImageNet, we train ResNet-18 (He et al., 2015) for image classification. For sentiment classification on SST-2, we fine-tune LLaMA-3B (Grattafiori et al., 2024) using LoRA. The detailed descriptions of the datasets and models are presented in Appendix F.

**Baselines.** We compare DDPA with nine state-of-the-art machine unlearning attack methods. **AwoP** (Liu et al., 2024e) amplifies trigger effects by injecting them into the frequency domain and requesting selective instance removal. **MUECPA** (Di et al., 2024) introduces poison and camouflage points to evade detection. **SSCSF** (ZHAO et al., 2023) optimizes crafted data update requests to exploit unlearning vulnerabilities. **BAU** (Zhang et al., 2023) conceals backdoors with mitigation samples and reactivates them via unlearning. **UBA-inf** (Huang et al., 2024b) enhances stealth and attack efficacy using label correction and influence functions. **RMBMU** (Ma et al., 2024) unlearns informative benign data to destabilize the model. **DABF** (Shin & Park, 2024) injects and conceals backdoors to bypass detection. **AdvUA** (Zhao et al., 2024) selects unlearning samples near victim samples to maximize adversarial impact. **EV-MUS** (Hu et al., 2024) moves data to decision boundaries to maximize unlearning effects. To the best of our knowledge, this work is the first to determine attack targets during the unlearning phase, enabling arbitrary target attacks and multiple attacks within the unlearning data budget. For detailed descriptions of each baseline, please refer to the appendix A.

**Evaluation Metrics.** We evaluate the performance of the

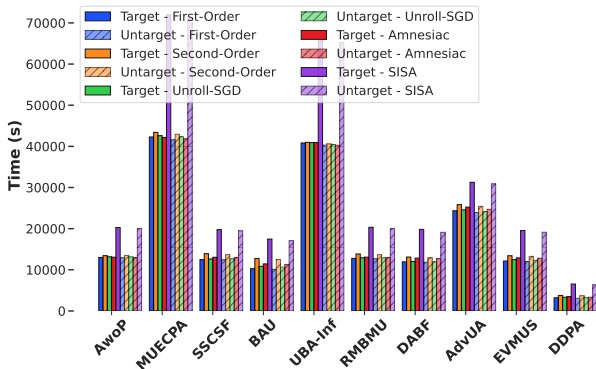

Figure 2: VGG-16 + CIFAR-100 (5 Unlearning Request) Time Comparison

attack based on the attack success rate (ASR), defined as the percentage of inputs that were successfully manipulated to achieve the objective of the adversary. For targeted attacks, ASR measures the proportion of samples misclassified into the target class. For untargeted attacks, ASR quantifies the proportion of samples misclassified into any incorrect class. In addition, we measure test accuracy (BA) and train accuracy (TA) degradation on benign inputs to assess the collateral impact on model performance.

**Variants of DDPA method.** We evaluate two variants of DDPA to highlight the advantages of different techniques. DDPA-S utilizes only the simplex method to maximize and generate an effective operational space. DDPA-C employs only Convex Polyhedral Approximation to ensure stability in constructing thrust vectors (group centers). DDPA operates with the full support of both simplex method and convex polyhedral aooriximation.

**Attack performance on various dataset with different unlearning algorithms.** Table 1 presents the TA, BA and ASR scores of five machine unlearning algorithms evaluated on test data arcoss 12 attack models. For each attack model ,we reserve 5%, 10% and 20% of the training dataset as the attack dataset. The conceal setting represents the evaluation before the unlearning attack, while unlearn refers to the results after the attack is applied.For targeted attacks, we observe that across all 12 attack methods, DDPA maintains an ASR of 0 before unlearning, while achieving relatively high test accuracy in most cases, even without data augmentation. This indicates that the poisoning dataset constructed by DDPA is highly stealthy. After the unlearning attack, DDPA achieves the highest ASR and the lowest BA, demonstrating its effectiveness in executing a successful attack. Compared to other attack models across the five machine unlearning algorithms, DDPA achieves an average ASR increase of 22.74%, 16.07%, and 21.45%, while reducing BA by 15.76%, 11.41%, and 2.27% on VGG16-CIFAR100,

Table 1: Unlearning Performance on VGG-16 with CIFAR100 (5% Unlearned)

| Method | B/A Unlearn | First-Order | | | Second-Order | | | Unroll-SGD | | | Amnesiac | | | SISA | | |
|---|---|---|---|---|---|---|---|---|---|---|---|---|---|---|---|---|
| | | TA | BA | ASR | TA | BA | ASR | TA | BA | ASR | TA | BA | ASR | TA | BA | ASR |
| AwoP | Conceal | 98.98 | 48.05 | 22.60 | 98.89 | 49.08 | 17.30 | 98.54 | 49.33 | 14.90 | 98.46 | 50.33 | 20.70 | 98.98 | 49.90 | 15.32 |
| | Unlearn | 97.64 | 47.00 | 86.00 | 97.21 | 47.74 | 89.00 | 97.37 | 47.42 | 85.00 | 96.90 | 46.67 | 85.00 | 97.81 | 46.48 | 65.00 |
| MUECPA | Conceal | 98.02 | 55.14 | 0.00 | 98.26 | 49.10 | 0.00 | 98.04 | 49.42 | 0.00 | 98.27 | 49.28 | 0.00 | 98.33 | 49.38 | 0.00 |
| | Unlearn | 95.01 | 46.43 | 80.40 | 97.25 | 47.54 | 88.10 | 97.19 | 47.37 | 86.80 | 96.54 | 45.95 | 82.20 | 97.20 | 47.10 | 85.38 |
| SSCSF | Conceal | 99.98 | 47.75 | 0.00 | 99.95 | 47.73 | 0.00 | 99.78 | 49.67 | 0.00 | 99.58 | 49.38 | 0.00 | 98.75 | 47.80 | 0.00 |
| | Unlearn | 98.72 | 45.07 | 80.00 | 97.39 | 47.75 | 90.00 | 97.46 | 47.43 | 86.93 | 97.17 | 44.82 | 88.00 | 97.22 | 45.51 | 86.26 |
| BAU | Conceal | 98.24 | 46.38 | 0.00 | 98.06 | 43.64 | 0.00 | 98.04 | 46.42 | 0.00 | 98.53 | 47.30 | 0.00 | 98.68 | 47.26 | 0.00 |
| | Unlearn | 96.28 | 44.73 | 80.00 | 96.56 | 45.42 | 78.00 | 97.90 | 43.17 | 70.80 | 96.98 | 42.73 | 79.80 | 97.45 | 45.20 | 80.00 |
| UBA-Inf | Conceal | 98.50 | 55.40 | 15.00 | 98.83 | 56.41 | 9.13 | 98.97 | 55.46 | 16.57 | 98.34 | 56.86 | 13.18 | 98.40 | 51.27 | 16.27 |
| | Unlearn | 97.23 | 54.41 | 89.00 | 97.34 | 54.67 | 85.83 | 97.41 | 52.18 | 84.98 | 96.68 | 50.17 | 84.62 | 96.24 | 46.26 | 81.52 |
| RMBMU | Conceal | 97.64 | 47.73 | 0.00 | 97.82 | 47.14 | 0.00 | 97.69 | 47.83 | 0.00 | 97.72 | 47.89 | 0.00 | 97.15 | 47.20 | 0.00 |
| | Unlearn | 95.28 | 46.24 | 85.00 | 96.38 | 46.67 | 86.00 | 96.76 | 45.40 | 84.38 | 95.02 | 44.89 | 85.00 | 96.90 | 45.25 | 86.74 |
| DABF | Conceal | 98.46 | 48.74 | 0.63 | 98.16 | 48.02 | 0.13 | 98.39 | 48.97 | 0.52 | 98.88 | 48.04 | 0.00 | 98.28 | 48.14 | 0.67 |
| | Unlearn | 97.64 | 46.69 | 86.00 | 97.47 | 47.68 | 81.21 | 97.24 | 45.48 | 85.00 | 96.81 | 42.44 | 87.11 | 97.14 | 46.14 | 84.45 |
| AdvUA | Conceal | 98.86 | 46.37 | 0.00 | 98.55 | 47.14 | 0.00 | 98.73 | 47.73 | 0.00 | 98.49 | 47.44 | 0.00 | 98.56 | 47.19 | 0.00 |
| | Unlearn | 96.49 | 45.29 | 80.20 | 96.38 | 45.42 | 85.52 | 97.07 | 45.39 | 83.57 | 96.63 | 45.57 | 85.34 | 96.69 | 45.83 | 83.68 |
| EVMUS | Conceal | 99.37 | 51.71 | 0.00 | 98.89 | 50.70 | 0.28 | 98.31 | 50.51 | 0.37 | 98.60 | 51.02 | 0.33 | 98.79 | 46.02 | 1.89 |
| | Unlearn | 98.97 | 50.62 | 79.20 | 96.07 | 46.99 | 81.77 | 97.90 | 46.36 | 67.66 | 94.31 | 47.62 | 73.61 | 97.29 | 45.36 | 79.26 |
| DDPA | Conceal | 96.97 | 48.26 | 0.00 | 96.87 | 48.74 | 0.00 | 98.36 | 48.64 | 0.00 | 98.27 | 47.11 | 0.00 | 98.45 | 47.76 | 0.00 |
| | Unlearn | 94.54 | 44.09 | 92.00 | 95.60 | 44.43 | 91.00 | 97.89 | 44.25 | 88.00 | 95.48 | 41.56 | 89.40 | 95.81 | 43.18 | 89.69 |
| DDPA-C | Conceal | 97.57 | 48.60 | 0.00 | 96.77 | 47.13 | 0.00 | 98.23 | 47.55 | 0.00 | 97.85 | 46.65 | 0.00 | 97.92 | 46.32 | 0.00 |
| | Unlearn | 95.45 | 45.58 | 80.00 | 95.35 | 45.93 | 80.20 | 96.86 | 46.86 | 80.40 | 95.58 | 43.59 | 81.20 | 95.16 | 44.11 | 80.60 |
| DDPA-S | Conceal | 96.42 | 48.37 | 0.00 | 96.90 | 47.38 | 0.00 | 98.19 | 47.33 | 0.00 | 98.63 | 46.42 | 0.00 | 98.01 | 46.06 | 0.00 |
| | Unlearn | 95.70 | 45.18 | 81.30 | 95.30 | 45.15 | 82.30 | 96.16 | 45.58 | 81.60 | 96.46 | 43.53 | 82.40 | 95.65 | 44.75 | 81.90 |

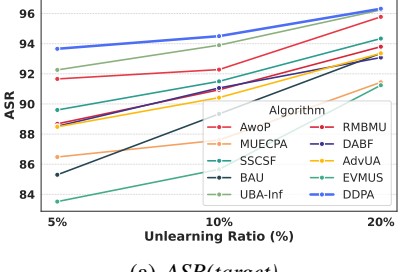

(a) *ASR(target)*

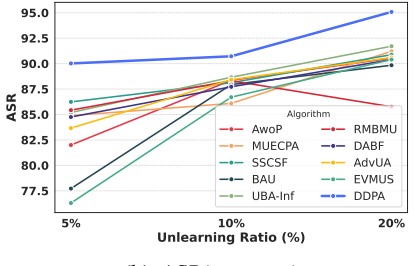

(b) *ASR(untarget)*

Figure 3: Attack Success Rate (ASR) comparison for target and untarget attacks.

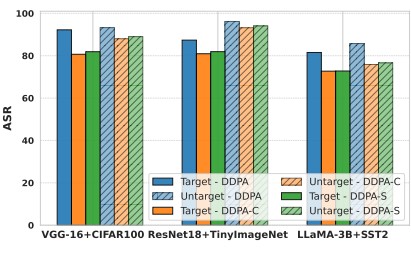

Figure 4: Time comparison for different methods in the ablation study.

ResNet18-Tiny-ImageNet, and LLaMA-3B-SST-2 respectively. In addition, the promising performance of DDPA with all machine unlearning algorithms implies that DDPA has greate potential as a general attack solution to other machine unlearning methods, which is desirable in practice.

**Evaluation of target-agnostic attack performance.** Figure 5 evaluates the flexibility of our method in a target-agnostic attack setting, where the attack target is unknown during the construction of the poisoned dataset. Since other attack methods require predefining a single target during poisoning and cannot adjust the target during the unlearning attack phase, we relax this constraint for comparison. Specifically, we assume they have prior knowledge of 5, 10, or 20 potential target classes, forcing them to distribute their poisoning budget across all potential targets rather than focusing on a single one. As the number of potential target classes increases, we observe a significant drop in ASR for

other attack methods, while DDPA maintains a high ASR. In the targeted attack setting, DDPA achieves a maximum ASR of 88.3%, whereas the lowest ASR among other methods is 5.9%. In the untargeted attack setting, DDPA reaches 91.6% ASR, while the lowest-performing method achieves only 6.3%. These results demonstrate DDPA's flexibility in adapting to different attack targets, allowing it to effectively execute unlearning attacks against any target.

**Running time with multi-attacks** Figure 2 evaluates the efficiency of our method in executing multiple attacks within a predefined poisoning budget. The attacker submits 2, 3, or 5 unlearning requests, each targeting a different attack objective. Since other attack methods predefine a single target and cannot dynamically adjust to multiple attacks, they must reconstruct a new poisoned dataset for each target, leading to significant time overhead. In contrast, DDPA uses a single pre-constructed dataset, eliminating the need for additional poisoning steps. As a result, DDPA efficiently

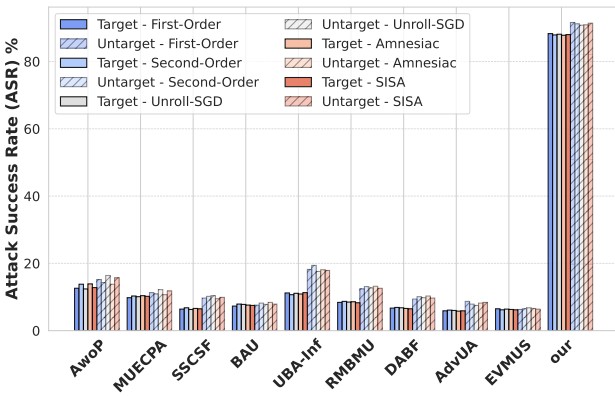

Figure 5: VGG-16 + CIFAR-100 (20 target) ASR Comparison

Table 2: Distribution Test Results on Two Model-Dataset Combinations

| | VGG16 + CIFAR-100 | | |
| --- | --- | --- | --- |
| | Shapiro-Wilk | D'Agostino's $K^2$ $p$-value | Anderson-Darling |
| Value | 0.55669 | 0.30275 | 0.24373 |
| | ResNet-18 + Tiny ImageNet | | |
| | Shapiro-Wilk | D'Agostino's $K^2$ $p$-value | Anderson-Darling |
| Value | 0.67146 | 0.35562 | 0.35510 |

executes multiple attacks across different datasets with minimal time cost. Compared to other methods, DDPA achieves the lowest running time, demonstrating its scalability and efficiency in multi-target attack scenarios.

**Ablation study.** Figure 4 presents the attack performance of two DDPA variants across five unlearning algorithms on CIFAR-100, Tiny-ImageNet, and SST-2. We observe that the full DDPA method achieves the lowest BA and the highest ASR in both targeted and untargeted attack settings, consistently outperforming other versions. A reasonable explanation is that our simplex method effectively maximizes the operational space while minimizing computational complexity, allowing for precise and efficient control of parameter manipulation during unlearning. In addition, our Convex Polyhedral Approximation stabilizes the behavior of thrust vectors (group centers) by mitigating the impact of loss function non-convexity, ensuring that parameter updates follow a structured and predictable trajectory.

**Impact of Unlearning Ratio** Figure 3 evaluates the impact of the unlearning ratio on ASR, ranging from 5% to 20% across CIFAR-100, Tiny-ImageNet, and SST-2. We observe that ASR increases as the percentage of unlearned samples grows, as larger unlearning sets amplify the disturbance to model parameters, making them more susceptible to attack. Notably, our method consistently achieves the highest ASR across all unlearning ratios, outperforming all other baselines.

**Validation of the Gaussian Assumption** We validate the Gaussian assumption using the Shapiro–Wilk, D'Agostino's $K^2$, and Anderson–Darling tests. For VGG16 on CIFAR-100, the $p$-values are 0.55669 (Shapiro–Wilk) and 0.30275 ($K^2$), with an Anderson–Darling statistic of 0.24373. For ResNet18 on Tiny ImageNet, the values are 0.67146, 0.35562, and 0.35510, respectively. All results fall within standard thresholds for normality: $p > 0.05$ and Anderson–Darling statistic ¡ 0.787. These consistent results across models and datasets support the Gaussian assumption, as shown in Table 2.

Table 3: Stealthiness Evaluation on ResNet-18 with Tiny ImageNet

| Method | NC | PB | AEVA |
| --- | --- | --- | --- |
| AwoP | -2.29 | 0.06 | -0.97 |
| MUECPA | -2.08 | -0.53 | -0.63 |
| SSCSF | -0.49 | -1.19 | -0.84 |
| BAU | -2.43 | -2.62 | 0.16 |
| UBA-Inf | -0.80 | -0.50 | 0.44 |
| RMBMU | -2.21 | 0.13 | -1.33 |
| DABF | -0.19 | -0.09 | -0.11 |
| AdvUA | -0.15 | -0.13 | -2.01 |
| EVMUS | -1.21 | -2.49 | -2.43 |
| our | 0.32 | 0.29 | 0.59 |

**Anomaly Detection Analysis** We report anomaly indices from three established detection methods: Neural Cleanse (NC), a perceptual similarity-based method (PB), and Autoencoder-based Variational Analysis (AEVA). These detectors assign class-wise anomaly scores, where values below –2.0 are typically considered indicative of backdoor behavior. As shown in Table 3, our method achieves scores of 0.322 (NC), 0.29 (PB), and 0.59 (AEVA), all comfortably above the standard detection threshold. These results are also consistently higher than those of baseline approaches, indicating that our attack does not leave strong or easily detectable backdoor traces.

## 5. Conclusions

In this work, we proposed a novel attack framework for machine unlearning (MU) that introduces a target-agnostic, on-demand attack strategy, enabling adversaries to dynamically specify arbitrary targets and efficiently execute multiple attack requests post-deployment. First, we leverage convex polyhedral approximation to identify stable group centers. Second, we employ the simplex method to construct a regular simplex over the group centers, maximizing parameter space coverage and allowing precise control over attack trajectories. Finally, we theoretically analyze the proportion of the parameter space occupied by the constructed simplex, providing guarantees on its effectiveness in guiding MU attacks.

## Acknowledgements

This research is partially sponsored by the National Science Foundation (NSF) under Grant No. OAC-2313191.

## Impact Statement

In this work, the two image datasets and one NLP dataset are all open-released datasets (Krizhevsky, 2009; Le & Yang, 2015; Socher et al., 2013), which allow researchers to use for non-commerical research and educational purposes. These three datasets are widely used in training/evaluating the machine unlearning. All baseline codes are open-accessed resources that are from the GitHub and locensed under the MIT License, which only requires preservation of copyright and license notices and includes the permissions of commercial use, modification, distribution, and private use.

To the best of our knowledge, this work is the first to introduce a dynamic delayed poisoning attack (DDPA) framework specifically designed for machine unlearning systems. Unlike existing methods, which require the predefined choice of attack targets during the poisoning dataset construction phase, our approach offers dynamic flexibility by delaying the selection of backdoor triggers and targeted attack objectives until after model training. Inspired by thrust vector control, a technique widely employed in aerospace engineering, our method strategically organizes data samples as "propellers" to manipulate model parameters efficiently during the unlearning process. Furthermore, we leverage convex polyhedral approximations to stabilize the loss function and ensure precise control over parameter updates, mitigating the unpredictable behavior introduced by non-convexity.

Our framework can play a critical role in evaluating and fortifying the robustness of machine unlearning systems, which are increasingly integrated into privacy-sensitive applications such as autonomous vehicles, healthcare analytics, and personalized AI systems. While primarily theoretical, we expect our findings to provide valuable insights into the vulnerabilities of unlearning systems and inspire the development of robust defense mechanisms. This paper is expected to produce a positive impact by improving the understanding of adversarial risks in unlearning scenarios, without posing immediate societal risks such as security, privacy, or fairness concerns.

An important contribution of this paper is the development of a geometry-driven poisoning strategy that dynamically adapts to changing adversarial objectives. By combining simplex-based data organization and convex approximation techniques, we ensure that the poisoning dataset remains highly effective across various unlearning configurations. Theoretical analysis supports the efficiency of this approach, with our findings showing significant attack success rates under strict data removal budgets. This work not only underscores the need for robust defenses against adversarial exploitation but also provides a foundation for future research in adversarial unlearning scenarios.

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

# A. Related Work

Trustworthy machine learning, which focuses on developing and deploying machine learning models that are not only accurate but also robust, private, fair, and explainable, has attracted active research in recent years (Palanisamy et al., 2018; Zhou et al., 2020b; Zhang et al., 2020; Zhou et al., 2021; Zhao et al., 2021; Ren et al., 2021; Zhang et al., 2021c;b;a; Zhou et al., 2022; Jin et al., 2022b; Zhang et al., 2022; Jin et al., 2022a;b; Che et al., 2022; Zhang et al., 2022; Liu et al., 2022a; Che et al., 2023b; Ren et al., 2023; Che et al., 2023a; Liu et al., 2023; 2024b;a; Zhou et al., 2024; Xiao et al., 2024; Liu et al., 2024d;c; Zhou et al., 2010; 2009; Cheng et al., 2011; Zhou & Liu, 2012; Cheng et al., 2012; Zhou & Liu, 2013; Su et al., 2013; Zhang et al., 2013; Zhou & Liu, 2014; Su et al., 2015; Zhou & Liu, 2015; Zhou et al., 2015a; Bao et al., 2015; Zhou et al., 2015b; 2016; Zhou, 2017; Zhou et al., 2018b;a; Ren et al., 2019; Zhou et al., 2019b;a;c; Zhou & Liu, 2019; Wu et al., 2020; 2021b; Zhou et al., 2020c;a; Jin et al., 2021; Wu et al., 2021c;a; Zhou et al., 2022; Guimu Guo & Zhou, 2022).

**Machine Unlearning.** Machine unlearning, also known as selective forgetting (Cao & Yang, 2015; Golatkar et al., 2020a; Shibata et al., 2021) or data removal/deletion (Ginart et al., 2019; Guo et al., 2023), focuese on eliminating the influence of specific subsets of training data on a pre-trained model (Garg et al., 2020a; Gupta et al., 2021; Nguyen et al., 2022; Wu et al., 2022). Current methods for machine unlearning can be broadly categorized into two main approaches, as outlined below.

(1) Exact machine unlearning algorithms aim to produce a model that performs identically to one trained from scratch, entirely excluding the data to be forgotten. The most straightforward approach, known as naive retraining, involves removing the data to be unlearned and retraining the model from scratch. However, this method incurs substantial computational and time costs. A notable exact unlearning method is Sharded, Isolated, Sliced, and Aggregated training (SISA) (Bourtoule et al., 2021). In SISA, the original training dataset is partitioned into multiple disjoint shards, with each training instance assigned to only one shard. Upon receiving an unlearning request, the model onwer retrains only the shard containing the affected data, significantly reducing retraining costs. The final prediction for a given instance is derived by aggregating predictions from all isolated shard models. Recent research has introduced innovative techniques to further enhance the efficiency and performance of exact unlearning. Methods such as dataset partitioning mechanisms and the use of lightweight adapters have been proposed to reduce the computational overhead while maintaining high accuracy (Chowdhury et al., 2024; Aldaghri et al., 2020; Yan et al., 2022; Kumar et al., 2022; Dukler et al., 2023; Golatkar et al., 2023; Pratama & Gambetta, 2024; Yang et al., 2024).

(2) Approximate machine unlearning methods aim to efficiently approximate the removal of specific training data's influence on a model without retraining from scratch. Notable approaches include first-order and second-order based unlearning methods (Warnecke et al., 2023), both of which transform changes in training data into closed-form parameter updates to derive the unlearned model. First-order methods leverage the first-order Taylor series expansion of the model, while second-order methods employ the inverse Hessian matrix of second-order derivatives to adjust the parameters. Another noteworthy method is UnrollSGD (Thudi et al., 2022), which formulates a gradient-based unlearning technique by extending a sequence of stochastic gradient descent (SGD) updates through a Taylor series. To reverse the effect of the unlearning data during the SGD stPS, this method adds the gradients of the unlearning data, computed with respect to the initial model weights, to the final model weights. Additionally, Amnesiac unlearning method (Graves et al., 2020) selectively reverses parameter updates associated with sensitive data by tracking which examples appeared in each training batch, providing a time-efficient mechanism with minimal impact on the model's overall performance. Recent research has introduced several innovative techniques to further improve the efficiency and effectiveness of approximate unlearning. For instance, methods based on influence functions estimate the impact of removing a specific data point by leveraging approximations, enabling computationally efficient adjustments to the model (Guo et al., 2023; Sekhari et al., 2021; Suriyakumar & Wilson, 2022; Mehta et al., 2022; Wu et al., 2022; Tanno et al., 2022). Re-optimization techniques iteratively fine-tune the model after data removal, ensuring that the influence of the data is eliminated while maintaining overall performance (Zhang et al., 2024; Park et al., 2024; Golatkar et al., 2020a;b; 2021). Gradient update methods incrementally adjust model parameters to account for the addition or removal of data points, providing a lightweight and scalable solution (Huang et al., 2024a; Hoang et al., 2023; Neel et al., 2020a; Gu et al., 2024; Cao et al., 2022; Neel et al., 2020b; Liu et al., 2022b). Additionally, graph unlearning addresses the challenges posed by graph-structured data, where inherent dependencies between nodes and edges require tailored strategies to forget specific elements without disrupting the graph's structure (Li et al., 2024; Wu et al., 2023a; Zhang, 2024; Yi & Wei, 2024;?; Wu et al., 2023b; Cheng et al., 2023; Chien et al., 2023).

**Poisoning-based Backdoor Attacks.** Poisoning-based backdoor attacks aim to embed hidden backdoors into machine learning models by manipulating the training data. In this work, we focus on the problem of poisoning attacks, which involve modifying the training data to implant backdoors into the model. A model compromised through such an attack

functions normally on benign inputs but consistently misclassifies inputs containing a specific trigger pattern to the attacker's desired target class (Gu et al., 2019; Chen et al., 2017; Nguyen & Tran, 2020; Zeng et al., 2022; Barni et al., 2019; Li et al., 2021a; Liu et al., 2018; Nguyen & Tran, 2021; Liu et al., 2020; Sarkar et al., 2020; Li et al., 2021b; Liao et al., 2018; Tan & Shokri, 2020; Cheng et al., 2021; Garg et al., 2020b; Bauman et al., 2018; Yao et al., 2019; Bagdasaryan & Shmatikov, 2021; Yang et al., 2022). These attacks typically involve the injection of poisoned samples into the training dataset, where each poisoned sample contains the predefined trigger. Consequently, the model learns to associate the trigger pattern with the target class during training, resulting in backdoor behavior at the testing stage when the same trigger pattern is present. Recent research on poisoning-based backdoor attacks can be categorized based on the type of triggers, the domain of application, and the optimization strategies employed. For trigger design, studies have proposed a variety of approaches, including adversarial noise combined with indiscriminate poisoning (Yu et al., 2024), kernel-based transformations (Gong et al., 2024), autoencoder-generated triggers (Xue et al., 2024), and low-frequency perturbations (Qiao et al., 2024). Other works emphasize invisibility and robustness by leveraging singular value decomposition (SVD) to embed imperceptible backdoors (Chen & Xu, 2024) or exploiting natural phenomena, such as fog (Ni et al., 2023), to seamlessly integrate triggers into the data. In terms of domain-specific applications, approaches like image scaling (Wu et al., 2023c) and deep steganography focus on vision tasks, while IQ sequence-based attacks target wireless communication systems (Huang et al., 2023). Backdoor attacks in generative models, targeting components like tokenizers or the language model (Vice et al., 2023), further extend the scope of these threats. Optimization-driven methods refine the effectiveness and stealthiness of attacks, including bi-level optimization with sparsity constraints (Gao et al., 2024), dynamic algorithms for manipulating decision boundaries (Ma et al., 2023), and gradient-based trigger generation techniques (Zhao et al., 2023). Additionally, innovative strategies have introduced physical-world triggers, such as uniform color space shifts (Jiang et al., 2023) and backdoor patches designed for camera inputs (Yuan et al., 2023). Some studies focus on enhancing attack stealth through natural integration of triggers or propose multi-stage frameworks that optimize attack success rates while maintaining high robustness (Rathbun et al., 2024b;a).

**Attacks on Machine Unlearning.** Machine unlearning focuse on removing the influence of specific data from a trained model, as discussed earlier. However, existing work on machine unlearning often overlook the risks associated with malicious unlearning requests, whcih can lead to model misclassifications. Recent studies have begun to explore the vulnerabilities introduced during the unlearning process and propose novel attack strategies that exploit these weaknesses.

For instance, AwoP introduces backdoor attacks by injecting triggers into the frequency domain of images and submitting malicious unlearning requests to amplify the backdoor effect, causing misclassification of triggered inputs (Liu et al., 2024e). MUECPA leverages camouflage datapoints to obscure the impact of poisoned datasets, making them more challenging to detect during training (Di et al., 2024). SSCSF explores selective forgetting attacks, addressing both static scenarios—where all malicious requests are submitted simultaneously—and sequential scenarios optimized through stochastic control frameworks (ZHAO et al., 2023). Another approach, BAU, constructs poisoned and mitigation samples to train a seemingly benign model, later exploiting unlearning requests to activate backdoors gradually (Zhang et al., 2023).

Innovative methods such as UBA-Inf use label correction and influence functions to create camouflage samples, enhancing both stealth and attack performance (Huang et al., 2024b). RMBMU takes a different approach by unlearning well-prepared benign data, causing a sudden collapse in model performance due to its reliance on these contributions during training (Ma et al., 2024). DABF employs a two-phase strategy to inject and temporarily conceal backdoors, reactivating them after partial model updates (Shin & Park, 2024). AdvUA aligns unlearning samples with adversarial directions, increasing the model's vulnerability to targeted attacks (Zhao et al., 2024). Finally, EVMUS amplifies the impact of unlearning by strategically moving data points to the model's decision boundary, maximizing the effect on the model's predictive capability (Hu et al., 2024).

While these approaches demonstrate the growing sophistication of attacks on machine unlearning, they share several limitations. Most require a predefined target when constructing the poisoning dataset, preventing flexibility in adapting attack objectives during the unlearning phase. Furthermore, these methods typically target a single objective and cannot attack multiple targets within a given unlearn data budget. Additionally, many rely on access to the training dataset, which may not always be feasible. Our proposed method effectively addresses these limitations by enabling adaptive multi-target attacks within a constrained unlearn data budget while eliminating the dependency on direct access to the training dataset.

## C. Proof of Theorems

3.1 provides a way to represent the position of a point relative to a simplex using barycentric coordinates.

**Definition 3.1** (Barycentric Coordinates in a Simplex). Let $A$ be an $n$-dimensional simplex in $\mathbb{R}^n$, with vertices denoted as $\{A_1, A_2, \ldots, A_{n+1}\}$, and let $M$ be any point in $\mathbb{R}^n$. Define $V_i$ for $i = 1, \ldots, n+1$ as the volume of the simplex formed by replacing the $i$-th vertex of $\{A_1, A_2, \ldots, A_{n+1}\}$ with $M$. The barycentric coordinates of $M$ with respect to the simplex are given as the ratios:

$$V_1 : V_2 : \cdots : V_{n+1}.$$

If the dimensions of the convex hulls $\mathrm{conv}\{A_1, \ldots, A_{i-1}, M, A_{i+1}, \ldots, A_{n+1}\}$ and $\mathrm{conv}\{A_1, \ldots, A_{n+1}\} \cap A$ are both $n$, the above ratios uniquely define the location of $M$ relative to the simplex.

Furthermore, suppose $A$ is a regular simplex, and let $B_2^n$ denote its inscribed ball. Denote the tangent points of $B_2^n$ with each face of $A$ by $\{B_i : i = 1, \ldots, n+1\}$. If the vertices of $A$ are $\{A_1, \ldots, A_{n+1}\}$, the barycentric coordinates of each tangent point $B_i$, with respect to the simplex vertices, are given by:

$$\left( \frac{1}{n}, \ldots, \frac{1}{n}, 0, \frac{1}{n}, \ldots, \frac{1}{n} \right),$$

where the 0 appears in the $i$-th position (corresponding to the vertex $A_i$) and all other components are $\frac{1}{n}$.

**Theorem 3.1.** *The John ellipsoid of a regular simplex is its inscribed ball. Let $A$ be a regular simplex in $\mathbb{R}^n$ with vertices $\{A_1, A_2, \ldots, A_{n+1}\}$ and $B_2^n$ as its inscribed ball. Denote by $\{B_i, i = 1, \ldots, n+1\}$ the tangent points which is opposite to $\{A_i, i = 1, \ldots, n+1\}$ respectivelyy. For positive weights $c_i = \frac{n}{n+1}$ $(i = 1, \ldots, n+1)$, the barycentric sum satisfies:*

$$\sum_{i=1}^{n+1} c_i B_i = \left( \frac{1}{n+1}, \ldots, \frac{1}{n+1} \right) \tag{18}$$

*Additionally, the solution to the representation of any point $x \in \mathbb{R}^n$ in the simplex is:*

$$\alpha = \left( \frac{n}{n+1} \langle u_1, x \rangle, \ldots, \frac{n}{n+1} \langle u_{n+1}, x \rangle \right) \tag{19}$$

*where $\{u_1, \ldots, u_{n+1}\}$ are unit normal vectors.*

*Proof.* According to Definition 3.3, to prove that the John ellipsoid of a regular simplex is its inscribed ball, we verify two key properties: the barycentric sum of the tangent points and the representation of any point $x \in \mathbb{R}^n$ within the simplex.

Let $A$ be a regular simplex in $\mathbb{R}^n$ with vertices $\{A_1, A_2, \ldots, A_{n+1}\}$, and let $B_2^n$ denote its inscribed ball. Denote the tangent points of $B_2^n$ opposite to $\{A_i\}$ as $\{B_i, i = 1, \ldots, n+1\}$. The barycentric coordinates of $B_i$ are:

$$B_i = \left( \frac{1}{n}, \ldots, \frac{1}{n}, 0, \frac{1}{n}, \ldots, \frac{1}{n} \right),$$

where 0 occupies the $i$-th position. Clearly, the barycentric coordinates of the origin are:

$$\left( \frac{1}{n+1}, \ldots, \frac{1}{n+1} \right).$$

Let $c_i = \frac{n}{n+1}$ for $i = 1, \ldots, n+1$. Then, we have:

$$\sum_{i=1}^{n+1} c_i B_i = \left( \frac{1}{n+1}, \ldots, \frac{1}{n+1} \right).$$

This confirms the barycentric sum property.

Next, we examine the representation of points within the simplex. For any $x \in \mathbb{R}^n$, it holds that:

$$x = \sum_{i=1}^{n+1} c_i \langle x, u_i \rangle u_i,$$

where $c_i = \frac{n}{n+1}$. Since $A$ is an $n$-dimensional simplex, the $n+1$ vectors $\{u_1, u_2, \ldots, u_{n+1}\}$ span $\mathbb{R}^n$, i.e.,

$$\text{span}\{u_1, u_2, \ldots, u_{n+1}\} = \mathbb{R}^n.$$

Define $\alpha = (\alpha_1, \ldots, \alpha_{n+1})$ and $\beta = (\langle u_1, x\rangle, \ldots, \langle u_{n+1}, x\rangle)$. The matrix $D$ is defined as:

$$D = \begin{bmatrix} \langle u_1, u_1\rangle & \langle u_1, u_2\rangle & \cdots & \langle u_1, u_{n+1}\rangle \\ \langle u_2, u_1\rangle & \langle u_2, u_2\rangle & \cdots & \langle u_2, u_{n+1}\rangle \\ \vdots & \vdots & \ddots & \vdots \\ \langle u_{n+1}, u_1\rangle & \langle u_{n+1}, u_2\rangle & \cdots & \langle u_{n+1}, u_{n+1}\rangle \end{bmatrix}.$$

This gives the equation system:

$$D\alpha^\top = \beta^\top.$$

Each element $\langle u_i, u_j\rangle$ represents the cosine of the angle between the outer normal vectors $u_i$ and $u_j$. Denote by $F_i$ and $F_j$ the faces whose outer normal vectors are $u_i$ and $u_j$, respectively. The dihedral angle $\angle(F_i, F_j)$ between $F_i$ and $F_j$ is related to $\langle u_i, u_j\rangle$ as:

$$\langle u_i, u_j\rangle = -\cos\angle(F_i, F_j).$$

The cosine of the dihedral angle satisfies:

$$\cos\angle(F_i, F_j) = \frac{S_{ji}}{S_j},$$

where $S_j$ is the $(n-1)$-dimensional volume of face $F_j$, and $S_{ji}$ is the volume of the projection of $F_j$ onto $F_i$ along $u_i$. For a regular simplex, the volumes $S_j$ and $S_{ji}$ are proportional, with:

$$\frac{S_{ji}}{S_j} = \frac{1}{n}.$$

Thus, the matrix $D$ becomes:

$$D = \begin{bmatrix} 1 & -\frac{1}{n} & \cdots & -\frac{1}{n} \\ -\frac{1}{n} & 1 & \cdots & -\frac{1}{n} \\ \vdots & \vdots & \ddots & \vdots \\ -\frac{1}{n} & -\frac{1}{n} & \cdots & 1 \end{bmatrix}.$$

Finally, let:

$$\alpha = \left(\frac{n}{n+1}\langle u_1, x\rangle, \ldots, \frac{n}{n+1}\langle u_{n+1}, x\rangle\right).$$

Since $\alpha$ satisfies the equation system, any point $x \in \mathbb{R}^n$ can be expressed as:

$$x = \sum_{i=1}^{n+1} \alpha_i u_i.$$

This completes the proof that the John ellipsoid of a regular simplex is its inscribed ball. $\qquad\square$

Theorem 4 establishes the relationship between integrable functions and unit vectors satisfying specific equality conditions, providing a bound on the integral product. Theorem 5 extends this by identifying the necessary conditions for equality, ensuring the unit vectors form an orthonormal basis in $\mathbb{R}^n$.

**Theorem 3.6** (Brascamp-Lieb inequality). *Let $\{u_i\}_{i=1}^m$ be a sequence of unit vectors in $\mathbb{R}^n$ and $\{c_i\}_{i=1}^m$ be a sequence of positive real numbers satisfying:*

$$\sum_{i=1}^m c_i u_i \otimes u_i = I_n,$$

*where $I_n$ is the identity matrix. For a sequence of integrable functions $f_i : \mathbb{R} \to [0, \infty)$, $i = 1, \ldots, m$, the following inequality holds:*

$$\int_{\mathbb{R}^n} \prod_{i=1}^{m} f_i(\langle u_i, x \rangle)^{c_i} \, dx \leq \prod_{i=1}^{m} \left( \int_{\mathbb{R}} f_i(t) dt \right)^{c_i}.$$

**Theorem 3.7** (Generalization of convolution inequality). *Let $\{u_i\}_{i=1}^{m}$ and $\{c_i\}_{i=1}^{m}$ satisfy the conditions of Theorem 4, and let $\{f_i\}_{i=1}^{m}$ be a sequence of functions, nonzero in $L_1(\mathbb{R})$ and not the density of a Gaussian distribution. The equality in Theorem 4 holds if and only if:*

$$m = n, \quad \text{and} \quad \{u_i\}_{i=1}^{m} \text{ forms an orthonormal basis of } \mathbb{R}^n.$$

**Theorem 3.2.** *A simplex $C$ in $\mathbb{R}^n$ is regular if and only if its John ellipsoid is the unit ball $B_2^n$. If $B_2^n$ is the John ellipsoid of $C$, then $B_2^n$ must be tangent to each face $F_i$ of $C$. Conversely, if $C$ is a regular simplex, its John ellipsoid is necessarily $B_2^n$, as the regularity ensures symmetry and equal tangency conditions.*

*The volume of $C$ satisfies:*

$$\text{Vol}(C) = \frac{\sqrt{n^n (n+1)^{n+1}}}{n!},$$

*which is the exact volume of a regular simplex with its inscribed ball being $B_2^n$. Furthermore, for the unit normal vectors $\{v_i\}_{i=1}^{n+1}$ corresponding to the faces of $C$, the inner product between any two distinct vectors satisfies:*

$$\langle v_i, v_j \rangle = -\frac{n+1}{n^2}, \quad i \neq j \tag{20}$$

*This establishes that the regularity of a simplex is directly characterized by the tangency, volume, and inner product properties of its John ellipsoid.*

*Proof.* To establish that a simplex $C$ in $\mathbb{R}^n$ is regular if and only if its John ellipsoid is the unit ball $B_2^n$, we proceed by proving both directions.

Assume that $B_2^n$ is the John ellipsoid of $C$. By definition, $B_2^n$ is the largest volume ellipsoid inscribed in $C$, tangent to each face $F_i$ of $C$. Denote the tangent points by $\{B_i, i = 1, \ldots, n+1\}$. Let $\{u_i, i = 1, \ldots, n+1\}$ be the outer unit normal vectors of the faces of the simplex.

The barycentric sum must satisfy:

$$\sum_{i=1}^{n+1} c_i u_i \otimes u_i = I_n,$$

and

$$\sum_{i=1}^{n+1} c_i u_i = 0,$$

where $c_i > 0$ are the weights ensuring that $B_2^n$ is the John ellipsoid.

Define the set $K = \{x \in \mathbb{R}^n : (x, u_i) \leq 1, i = 1, \ldots, n+1\}$. Since the tangent points $\{B_i\}$ are on the boundary of $C$ and $B_2^n$, we observe that $K \subseteq C$. The tangency condition implies that $K$ and $C$ share the same boundary points, hence $K = C$.

To confirm that $C$ is regular, consider $\mathbb{R}^{n+1}$ as $\mathbb{R}^n \times \mathbb{R}$. Let the vectors:

$$v_i = \sqrt{\frac{n}{n+1}} \left( -u_i, \frac{1}{\sqrt{n}} \right), \quad i = 1, \ldots, n+1,$$

and assign weights:

$$d_i = \frac{n+1}{n} c_i, \quad i = 1, \ldots, n+1.$$

The above definitions ensure that:

$$\sum_{i=1}^{n+1} d_i v_i \otimes v_i = I_{n+1}.$$

Define the function sequence $\{f_i(t)\}$ as:

$$f_i(t) = \begin{cases} e^{-t}, & t \geq 0, \\ 0, & t < 0. \end{cases}$$

For any $x \in \mathbb{R}^{n+1}$, let:

$$F(x) = \prod_{i=1}^{n+1} f_i\left(\langle v_i, x \rangle\right)^{d_i}.$$

By Theorem 4, we have:

$$\int_{\mathbb{R}^n} F(x)dx \leq \prod_{i=1}^{n+1} \left(\int_{\mathbb{R}} f_i(t)dt\right)^{d_i} = 1.$$

Following the integration, using results from [Ba2], we calculate:

$$e^{-\sqrt{n+1}r}\mathrm{Vol}(K) \leq e^{-\sqrt{n+1}r}r^n\mathrm{Vol}(K) \implies \mathrm{Vol}(K) \leq \frac{\sqrt{n^n(n+1)^{n+1}}}{n!}.$$

This matches the volume of a regular simplex with its inscribed ball being $B_2^n$.

Observe the construction of $\{f_i\}$ and Theorem 5, which shows that equality holds in the volume bound, and $\{v_i\}_{i=1}^{n+1}$ forms an orthonormal basis in $\mathbb{R}^{n+1}$. For any two vectors of this basis:

$$v_i = \sqrt{\frac{n}{n+1}}\left(-u_i, \frac{1}{\sqrt{n}}\right), \quad v_j = \sqrt{\frac{n}{n+1}}\left(-u_j, \frac{1}{\sqrt{n}}\right),$$

we compute:

$$\langle v_i, v_j \rangle = \frac{n}{n+1}\langle u_i, u_j \rangle + \frac{1}{n+1}.$$

For $i \neq j$:

$$\langle u_i, u_j \rangle = -\frac{n+1}{n^2}.$$

Because the vectors $\{u_i\}_{i=1}^{n+1}$ are normal to the $n+1$ faces of the simplex $K$, this confirms that $K$ is a regular simplex.

This completes the proof. $\qquad \square$

**Theorem 3.3.** *A regular simplex can be quantified by how closely it satisfies the conditions of the John ellipsoid. For a given set of group centers $V = \{v_1, v_2, \ldots, v_n\}$, let $I$ denote the identity matrix, $v_i$ represent individual group centers, and $T$ denote the total number of group centers. Define the regularity measure based on the group centers $V$ as:*

$$\phi(V) = \frac{1}{T}\sum_{t=1}^{T}\exp\left(-\frac{\|I - \sum_{i=1}^{n}c_iv_i \otimes v_i\|^2}{2\sigma^2}\right) \tag{21}$$

*where $\sigma^2 = \frac{1}{2\pi}$ is the variance of the Gaussian function.*

*Proof.* The goal is to establish that the defined regularity measure $\phi(V)$ quantifies how closely a given set of group centers $V = \{v_1, v_2, \ldots, v_n\}$ satisfies the conditions of a regular simplex, characterized by the John ellipsoid being the unit ball $B_2^n$.

A regular simplex aligns with the John ellipsoid if and only if its group centers satisfy specific geometric and algebraic properties. These include symmetry, tangency of the ellipsoid to the simplex faces, and uniform distribution of the group centers. To capture these properties mathematically, the Frobenius norm $\|I - \sum_{i=1}^{n}c_iv_i \otimes v_i\|^2$ measures the deviation from the identity matrix $I$, which represents an ideal configuration of the group centers. The function $\phi(V)$ is then defined as:

$$\phi(V) = \frac{1}{T}\sum_{t=1}^{T}\exp\left(-\frac{\|I - \sum_{i=1}^{n}c_iv_i \otimes v_i\|^2}{2\sigma^2}\right),$$

where $c_i > 0$ are weights, and $\sigma^2 = \frac{1}{2\pi}$ is the variance of the Gaussian function.

To prove that $\phi(V)$ serves as a valid regularity measure, consider the term $\|I - \sum_{i=1}^{n} c_i v_i \otimes v_i\|^2$. Let $M = I - \sum_{i=1}^{n} c_i v_i \otimes v_i$, where $M$ represents the deviation matrix. The Frobenius norm of $M$ is:

$$\|M\|^2 = \text{trace}(M^\top M).$$

Expanding $M^\top M$, we have:

$$M^\top M = \left( I - \sum_{i=1}^{n} c_i v_i \otimes v_i \right)^\top \left( I - \sum_{i=1}^{n} c_i v_i \otimes v_i \right).$$

Simplifying the terms, we obtain:

$$\|M\|^2 = \text{trace}(I^\top I) - 2\,\text{trace}\left( I^\top \sum_{i=1}^{n} c_i v_i \otimes v_i \right) + \text{trace}\left[ \left( \sum_{i=1}^{n} c_i v_i \otimes v_i \right)^\top \left( \sum_{j=1}^{n} c_j v_j \otimes v_j \right) \right].$$

The first term $\text{trace}(I^\top I)$ simplifies to $n$, the dimensionality of $I$. The second term evaluates to:

$$\text{trace}\left( I^\top \sum_{i=1}^{n} c_i v_i \otimes v_i \right) = \sum_{i=1}^{n} c_i.$$

The third term expands as:

$$\text{trace}\left[ \left( \sum_{i=1}^{n} c_i v_i \otimes v_i \right)^\top \left( \sum_{j=1}^{n} c_j v_j \otimes v_j \right) \right] = \sum_{i=1}^{n} \sum_{j=1}^{n} c_i c_j \langle v_i, v_j \rangle^2,$$

where $\langle v_i, v_j \rangle$ denotes the inner product between the group centers $v_i$ and $v_j$.

Combining these results, we obtain:

$$\|M\|^2 = n - 2 \sum_{i=1}^{n} c_i + \sum_{i=1}^{n} \sum_{j=1}^{n} c_i c_j \langle v_i, v_j \rangle^2.$$

The Gaussian function $\exp\left( -\frac{\|M\|^2}{2\sigma^2} \right)$ applies a penalty to configurations with higher deviation $\|M\|^2$, emphasizing those closer to the identity matrix. By summing this weighted measure over $T$ configurations and normalizing, $\phi(V)$ provides a robust measure of regularity across multiple simplex configurations.

The term $\sigma^2 = \frac{1}{2\pi}$ controls the sensitivity of the Gaussian function, ensuring that minor deviations from regularity are not overly penalized, while significant deviations are sharply discouraged.

Thus, $\phi(V)$ effectively quantifies the regularity of a simplex by penalizing deviations from the identity matrix. Its formulation ensures that the measure aligns with the geometric and algebraic conditions of the John ellipsoid, completing the proof. $\square$

The definition 3.4 establishes the property of logarithmic concavity, ensuring a specific structure of functions where the weighted averages of the inputs produce outputs bounded by a geometric mean.

**Definition 3.4** (Logarithmically Concave Function). A function $f : \mathbb{R}^n \to \mathbb{R}$ is called logarithmically concave if for any $x, y \in \mathbb{R}^n$ and $0 < \lambda < 1$,
$$f(\lambda x + (1 - \lambda)y) \geq f(x)^\lambda f(y)^{1-\lambda}.$$

The lemma 3.5 provides an inequality for integrals involving logarithmically concave functions, facilitating bounds for complex integrals in optimization and probabilistic analysis.

**Lemma 3.5.** *Let $G : \mathbb{R} \to [0, \infty)$ be a logarithmically concave function and $k > 0$. Then:*

$$G(0)^k \int_0^\infty G(x)x^k dx \leq \Gamma(k+1) \left( \int_0^\infty G(x)dx \right)^{k+1}.$$

The lemma 3.6 bounds products of weighted integrals of logarithmically concave functions, which is critical for deriving inequalities in functional analysis and probability theory.

**Lemma 3.6.** *Let $f$ be a positive logarithmically concave function on $\mathbb{R}$. Then:*

$$\left( \int_0^\infty f(x)e^{-x}dx \right) \left( \int_0^\infty f(x)(1+x)dx \right) \leq \left( \int_0^\infty f(x)dx \right)^2.$$

**Theorem 3.4.** *Following (Li et al., 2015; Pearce et al., 2020; de G. Matthews et al., 2018), assume the parameter space follows a Gaussian distribution in $\mathbb{R}^d$. The proportion of the parameter space (PS) occupied by the constructed $(n-1)$-dimensional simplex (S)is:*

$$\rho = \frac{Vol(S)}{Vol(PS)} = \frac{\sqrt{n+1}\,\Gamma\left(\frac{m}{2}+1\right)}{(n-1)!\,l\,\pi^{m/2}\sqrt{2}\sigma^m} \tag{22}$$

*where $\Gamma$ is the Gamma function, $\sigma$ represents the standard deviation of the Gaussian distribution defining the spread of the parameter space, $m$ is the dimensionality of the Gaussian parameter space, and $l$ is the edge length of the constructed simplex.*

*Proof.* To establish the proportion $\rho$ of the parameter space (PS) occupied by the constructed $(n-1)$-dimensional simplex, we calculate the ratio of the simplex volume $Vol(S)$ to the PS volume $Vol(PS)$.

The parameter space corresponds to the high-probability region of a Gaussian distribution in $\mathbb{R}^d$, defined by the probability density function:

$$f(x) = \frac{1}{(2\pi\sigma^2)^{m/2}} \exp\left( -\frac{\|x-\mu\|^2}{2\sigma^2} \right) \tag{23}$$

where $m$ is the dimensionality of the Gaussian space, $\mu$ is the mean, and $\sigma^2$ is the variance. The PS can be approximated by a ball centered at $\mu$ with radius $R$, enclosing a specified probability mass $P$. Assuming $R = \sqrt{2}\sigma$, the radius corresponds to approximately 95% of the probability mass for a standard Gaussian distribution. The volume of this $m$-dimensional ball is given by:

$$Vol(PS) = \frac{\pi^{m/2}R^m}{\Gamma\left(\frac{m}{2}+1\right)} \tag{24}$$

Substituting $R = \sqrt{2}\sigma$, we have:

$$Vol(PS) = \frac{\pi^{m/2}(\sqrt{2}\sigma)^m}{\Gamma\left(\frac{m}{2}+1\right)} \tag{25}$$

Define functions $f : \mathbb{R} \to [0, \infty)$ and $F : \mathbb{R}^{n+1} \to [0, \infty)$ by:

$$f(x) = \begin{cases} e^{-x}, & x \geq 0, \\ 0, & x < 0, \end{cases}$$

and

$$F(x) = \prod_{j=1}^{n+1} f(x_j),$$

where $x = (x_1, x_2, \ldots, x_{n+1}) \in \mathbb{R}^{n+1}$. Let $H$ be a hyperplane with $\sum_{j=1}^{n+1} a_j = 0$, where $\mathbf{a} = (a_1, a_2, \ldots, a_{n+1})$.

For a fixed $t > 0$, $F$ is constant on $S_t$, and:

$$Vol_{n-1}(H \cap S_t) = t^{n-1}Vol_{n-1}(H \cap S_1).$$

A change of coordinates gives:

$$\int_H F \, d\text{Vol}_H = \int_H \prod_{j=1}^{n+1} f(x_j) \, d\text{Vol}_H = \int_0^\infty e^{-s\sqrt{n+1}} \text{Vol}_{n-1}(H \cap S_{s/\sqrt{n+1}}) \, ds,$$

which simplifies to:

$$\int_H F \, d\text{Vol}_H = (n-1)!(n+1)^{-1/2} \text{Vol}_{n-1}(H \cap S_1).$$

Thus:

$$\text{Vol}_{n-1}(H \cap S_1) = \frac{\sqrt{n+1}}{(n-1)!} \int_H F \, d\text{Vol}_H.$$

Next, using the Fourier inversion formula, denote the characteristic function of the random variable $Y$ by $\Phi_Y$:

$$\Phi_{\sum_{j=1}^{n+1} a_j X_j}(t) = \prod_{j=1}^{n+1} \Phi_{X_j}(a_j t) = \prod_{j=1}^{n+1} \frac{1}{1 + ia_j t}.$$

From the Fourier inversion:

$$G(s) = \frac{1}{2\pi} \int_{-\infty}^{\infty} \prod_{j=1}^{n+1} \frac{1}{1 + ia_j t} e^{ist} \, dt.$$

Assuming $a_j \neq 0$ for all $j$, we simplify to:

$$\int_H F \, d\text{Vol}_H = G(0) = \frac{1}{2\pi} \int_{-\infty}^{\infty} \prod_{j=1}^{n+1} \frac{1}{1 + ia_j t} \, dt.$$

Applying Hölder's inequality, and using $\sum_{j=1}^{n+1} a_j^2 = 1$:

$$\int_H F \, d\text{Vol}_H \leq \frac{1}{2\pi} \prod_{j=1}^{n+1} \left( \int_{-\infty}^{\infty} \frac{1}{(1 + ia_j t)^2} \, dt \right)^{1/2}.$$

The integral evaluates to:

$$\int_H F \, d\text{Vol}_H \leq \frac{1}{l},$$

with equality if and only if $n-1$ of the $a_j$ are zero.

Now consider $|a_j| > \frac{1}{l}$ for some $j$. Without loss of generality, assume $a_1 > \frac{1}{l}$. The convolution of densities $(h * g)(t)$ is:

$$(h * g)(0) = \int_{-\infty}^{\infty} h(x)g(-x) \, dx.$$

Using Lemma 1.3:

$$\int_0^\infty e^{-y} g(-a_1 y) \, dy \leq \frac{1}{a_1} \left( \int_0^\infty g(-a_1 y) \, dy \right)^2.$$

Thus:

$$\int_H F \, d\text{Vol}_H \leq \frac{1}{l}.$$

Then, the constructed $(n-1)$-dimensional simplex $S$, characterized by edge length $l$, has a volume:

$$\text{Vol}(S) = \frac{\sqrt{n+1}}{(n-1)!l} \tag{26}$$

The proportion of the PS occupied by the simplex is:

$$\rho = \frac{\text{Vol}(S)}{\text{Vol}(\text{PS})} \tag{27}$$

Substituting the expressions for $\text{Vol}(S)$ and $\text{Vol}(\text{PS})$, we get:

$$\rho = \frac{\frac{\sqrt{n+1}}{(n-1)!} l^{n-1}}{\frac{\pi^{m/2}(\sqrt{2}\sigma)^m}{\Gamma\left(\frac{m}{2}+1\right)}} \tag{28}$$

Simplifying, this becomes:

$$\rho = \frac{\sqrt{n+1}\,\Gamma\left(\frac{m}{2}+1\right)}{(n-1)!\,l\,\pi^{m/2}\sqrt{2}\sigma^m} \tag{29}$$

Here, the term $\Gamma\left(\frac{m}{2}+1\right)$ arises from the volume formula of the $m$-dimensional Gaussian ball, while $\sqrt{n+1}$ reflects the geometric property of the constructed simplex. The normalization factor $(n-1)!$ accounts for the simplex dimensionality, and the terms $l$ and $\sigma$ represent the edge length and Gaussian spread, respectively. This relationship explicitly quantifies the proportion of the high-probability Gaussian region occupied by the constructed simplex. Thus, the theorem is proven. □

**Theorem 3.5.** *Let $g$ be a convex function defined over a polytope $P$, and let $(x_0, y_0) \in P$ be a reference point. The convex envelope of $g$ at $(x_0, y_0)$ is the solution to the optimization problem:*

$$Conv_{g,P}(x_0, y_0) = \max c, \quad \text{subject to:}$$
$$g(x_i, y_i) - [a(x_i - x_0) + b(y_i - y_0) + c] \geq 0, \quad \forall(x_i, y_i) \in V'(P),$$
$$\min_{x \in [x_j^1, x_j^2]} [g(x, m_j x + q_j) - a(x - x_0) - b(m_j x + q_j - y_0) - c] \geq 0, \quad \forall e_j \in E'(P),$$

*where $V'(P) \subseteq V(P)$ and $E'(P)$ denote the subsets of vertices and edges, respectively.*

*Proof.* The proof begins by establishing the constraints necessary for the convex envelope at the reference point $(x_0, y_0)$. First, consider the vertices $(x_i, y_i) \in V'(P)$, where $V'(P) \subseteq V(P)$ excludes vertices lying on edges in $E'(P)$. At each vertex, the inequality

$$g(x_i, y_i) - [a(x_i - x_0) + b(y_i - y_0) + c] \geq 0$$

ensures that the affine function defined by $a$, $b$, and $c$ underestimates $g(x, y)$. The underestimation at these discrete points guarantees feasibility of the optimization problem with respect to the vertices.

Next, we analyze the constraints along the edges $e_j \in E'(P)$, where $E'(P)$ consists of edges along which $g(x, y)$ exhibits strict convexity. Each edge $e_j$ is parameterized by a linear function:

$$y = m_j x + q_j,$$

where $m_j$ and $q_j$ define the slope and intercept of the edge. Along each edge, the strict convexity of $g(x, y)$ implies that the minimum value of the following expression:

$$g(x, m_j x + q_j) - a(x - x_0) - b(m_j x + q_j - y_0) - c$$

occurs either at one of the endpoints $x_j^1, x_j^2$ of the edge or at a critical point $s_j(a, b)$ within the interval $[x_j^1, x_j^2]$. To enforce the underestimation constraint along the edge, it is required that

$$\min_{x \in [x_j^1, x_j^2]} [g(x, m_j x + q_j) - a(x - x_0) - b(m_j x + q_j - y_0) - c] \geq 0.$$

This constraint can be reformulated by analyzing the unconstrained minimum point $s_j(a, b)$ of:

$$f_{e_j}(x) = g(x, m_j x + q_j) - a(x - x_0) - b(m_j x + q_j - y_0).$$

If $s_j(a, b)$ lies within the interval $[x_j^1, x_j^2]$, the minimum occurs at $s_j(a, b)$. Otherwise, the minimum occurs at one of the endpoints $x_j^1$ or $x_j^2$. Using the strict convexity of $g(x, y)$, the first derivative of $f_{e_j}$ with respect to $x$ determines the nature of $s_j(a, b)$:

$$\frac{\partial}{\partial x} f_{e_j}(x) = g_x(x, m_j x + q_j) + m_j g_y(x, m_j x + q_j) - a - b m_j.$$

The sign of this derivative at the endpoints $x_j^1$ and $x_j^2$ allows us to evaluate whether $s_j(a, b)$ lies inside or outside the interval $[x_j^1, x_j^2]$.

Combining the vertex and edge constraints, the convex envelope is expressed as the solution to the following optimization problem:

$$\text{Conv}_{g,P}(x_0, y_0) = \max c,$$

subject to:

$$g(x_i, y_i) - [a(x_i - x_0) + b(y_i - y_0) + c] \geq 0, \quad \forall (x_i, y_i) \in V'(P),$$
$$g_j(a, b) + a x_0 + b y_0 - b q_j \geq c, \quad \forall e_j \in E'(P),$$

where $g_j(a, b)$ is defined as:

$$g_j(a, b) = \begin{cases} f_{e_j}(x_j^1), & \text{if } f'_{e_j}(x_j^1) \geq 0, \\ f_{e_j}(x_j^2), & \text{if } f'_{e_j}(x_j^2) \leq 0, \\ f_{e_j}(s_j(a, b)), & \text{otherwise.} \end{cases}$$

where $h_j(a, b)$ corresponds to the minimum value at the critical point $s_j(a, b)$. By the strict convexity of $g(x, y)$, $s_j(a, b)$ is unique and determined by the derivative $f'_{e_j}(s_j(a, b)) = a + b m_j$. The continuity of $h_j$ and its derivatives depends on $s_j(a, b)$, which is given by:

$$\frac{\partial s_j}{\partial a}(a, b) = -s_j(a, b), \quad \frac{\partial s_j}{\partial b}(a, b) = -(m_j s_j(a, b)).$$

Substituting into the derivative of $h_j$, we have:

$$\frac{\partial h_j}{\partial a}(a, b) = f'_{e_j}(s_j(a, b)) \frac{\partial s_j}{\partial a} - s_j(a, b),$$

and similarly for $\frac{\partial h_j}{\partial b}(a, b)$. Since $f_{e_j}(x)$ is strictly convex, its derivatives are continuous, ensuring the differentiability of $h_j(a, b)$.

Finally, the convexity of $g_j(a, b)$ and the differentiability of its components $l_j^1, l_j^2, h_j$ ensure that the edge-related constraints are continuously differentiable. Combined with the vertex constraints, this proves that the entire optimization problem is convex with continuously differentiable constraints.

This concludes the proof. $\qquad\square$

# D. Algorithm

---
**Algorithm 1** Generate Poisoned Dataset

---
    **Input:** Target model $f$, initial parameters $w_0$ at $t_0$, loss function $L$, number of group centers $n$
    **Output:** Poisoned dataset $D_u$
    Construct $f'$ as a convex approximation of $f$
    Set the number of propeller groups n
    Initialize group centers $V = \{v_1, v_2, \ldots, v_n\}$
    **while** Not Converge **do**
        $V \leftarrow \arg\min_V L_t + L_s$
    **end while**
    **for** $v_i \in V$ **do**
        Sample data $D_i$ from $\mathcal{N}(v_i, \sigma^2)$
    **end forreturn** $\bigcup_{i=1}^{v} D_i$

---

Based on the methods described in Section 3.2 and Section 3.3, we summarize the stPS to construct the poisoned dataset. As described in Algorithm 1, the construction of the poisoning dataset begins with the initialization of the group centers, which serve as the key thrust points. The positions of these group centers are then optimized through convex optimization to stabilize the thrust directions and ensure precise control of the model's parameters during the unlearning process. A critical step in this process is the construction of a regular simplex using the optimized group centers, which maximizes the operational space available for parameter adjustments. This ensures that the model can efficiently and effectively control its parameters within the desired range. Finally, data is sampled around the group centers based on a predefined distribution to generate the poisoning dataset, which enables the attacker to exert precise influence on the model during the unlearning attack phase.

## E. Attack Settings

To evaluate the performance of DDPA, we use the following unlearning methods to implement the attack: first-order-based, second-order-based, Unrolling SGD, Amnesiac, and SISA. Following prior studies, the attack budget is set between 5% and 20% of the training dataset. In our experimental setup, the model is pre-trained on the clean training dataset along with the poisoning dataset generated using our proposed method. The proportion of poisoning samples in the training set is determined by the attack budget. After pre-training, our attack method is implemented under both targeted and untargeted attack scenarios. In the targeted scenario, the attacker manipulates the unlearning process to cause the model to misclassify specific inputs into a target class chosen by the attacker. In the untargeted scenario, the attacker aims to disrupt the model's predictions, causing it to produce incorrect outputs without a specific target class in mind. These scenarios evaluate the flexibility and effectiveness of the proposed method in different adversarial settings.

## F. Experimental Details

**Environment.** The experiments were conducted on a compute server running on Red Hat Enterprise Linux 7.2 with 2 CPUs of Intel Xeon E5-2650 v4 (at 2.66 GHz) and 8 GPUs of NVIDIA GeForce GTX 2080 Ti (with 11 GB of GDDR6 on a 352-bit memory bus and memory bandwidth in the neighborhood of 620GB/s) and 4 GPUs of NVIDIA H100 (each with 80GB of HBM2e memory on a 5120-bit memory bus, offering a memory bandwidth of approximately 3TB/s),256GB of RAM, and 1TB of HDD. Overall, the experiments took about 10 days in a shared resource setting. We expect that a consumer-grade single-GPU machine could complete the full set of experiments in around 21-23 days, if its full resources were dedicated. The codes were implemented in Python 3.7.10 and PyTorch 1.9.0. Since the datasets used are all public datasets and our methodologies and the hyperparameter settings are explicitly described in section 4 and F, our codes and experiments can be easily reproduced on top of a GPU server.

**Training.** We conduct experiments on three standard datasets: CIFAR-100, Tiny ImageNet, and SST-2, covering image and sentiment classification tasks. The datasets are publicly available and are widely used for non-commercial research and educational purposes. For CIFAR-100, we use 50,000 examples for training and 10,000 examples for testing, training a VGG16 model for image classification over 150 epochs. On Tiny ImageNet, we use 100,000 examples for training and 10,000 examples for testing, training a ResNet-18 model for image classification over 150 epochs. For SST-2, we use 20,000 examples for training and 872 examples for testing, fine-tuning a LLaMA-3B model with LoRA for sentiment classification over 10 epochs. All neural networks are trained using SGD optimization, starting with an initial learning rate of 0.1 and a batch size of 64. Each experiment is repeated three times to ensure stable and reliable results.

**Implementation.** For 9 state-of-art machine unlearning attack methods of AwoP (Liu et al., 2024e),MUECPA (Di et al., 2024),SSCSF (ZHAO et al., 2023),BAU (Zhang et al., 2023),UBA-Inf (Huang et al., 2024b), RMBMU (Ma et al., 2024), DABF (Shin & Park, 2024), AdvUA (Zhao et al., 2024), EVMUS (Hu et al., 2024), we utilized the same model architecture as the official open-source implementation and default parameter settings provided by the original authors in all experiments. All hyperparameters are standard values from reference codes or prior works. We validate the performance of different attack methods with a rang of unlearing ratio $\in \{5\%, 10\%, 20\%\}$, which are commonly used in related studies. For the image datasets, CIFAR-100 and Tiny ImageNet, all models were trained for 150 epochs using a batch size of 128 and a learning rate of 0.1. For the sentiment dataset SST-2, all models were trained for 50 epochs with a batch size of 8 and a learning rate of 4e-4. These training settings were chosen to align with best practices in the literature and ensure consistent comparisons across the experiments. The above open-source codes from the GitHub are licensed under the MIT License, which only requires preservation of copyright and license notices and includes the permissions of commercial use, modification, distribution, and private use.

For our DDPA method, we performed hyperparameter selection by performing a parameter unlearning ratio $\in \{5\%, 10\%, 20\%\}$, Group center $V \in \{5, 10, 15, 20, 25\}$, unlearning rate $\in \{1e^{-3}, 4e^{-3}, 1e^{-4}, 4e^{-4}, 1e^{-5}\}$. For the image datasets, CIFAR-100 and Tiny ImageNet, training epoch$\in \{30, 60, 90, 120, 150\}$, learning rate $\in \{0.001, 0.005, 0.01, 0.05, 0.1\}$. We select the bets parameters over 100 epochs of training and evaluate the model at test time. For the sentiment dataset SST-2, training epoch$\in \{10, 25, 50, 100\}$, learning rate $\in \{1e^{-4}, 3e^{-4}, 4e^{-4}, 5e^{-5}\}$. We select the bets parameters over 50 epochs of training and evaluate the model at test time.

**Hyperparameter settings.** Unless otherwise explicitly stated, we used the following default parameter settings in the experiments. As shown in Table 4

| Parameter | Value |
|---|---|
| Training data on SST-2 | 20,000 |
| Test data ratio on SST-2 | 872 |
| Training data on CIFRA100 | 5,000 |
| Test data on CIFRA100 | 1,000 |
| Training data on Tiny-ImageNet | 100,000 |
| Test data on Tiny-ImageNet | 10,000 |
| Group Center $V$ | 5 |
| Training epochs of the DDPA on image dataset | 150 |
| Training epochs of the DDPA on sentiment dataset | 50 |
| Batch size for training the model on image dataset | 128 |
| Batch size for training the model on sentiment dataset | 8 |
| Learning rate on image dataset | 0.1 |
| Learning rate on sentiment dataset | 1e-4 |

Table 4: Model parameters and settings

### F.1. Additional Experiments

**Attack performance on various dataset with different unlearning algorithms.** Table 5 - 22 presents the TA, BA and ASR scores of five machine unlearning algorithms evaluated on test data arcoss 12 attack models. For each attack model ,we reserve 5%, 10% and 20% of the training dataset as the attack dataset. The conceal setting represents the evaluation before the unlearning attack, while unlearn refers to the results after the attack is applied. For targeted attacks, we observe that across all 12 attack methods, DDPA maintains an ASR of 0 before unlearning, while achieving relatively high test accuracy in most cases. This indicates that the poisoning dataset constructed by DDPA is highly stealthy. After the unlearning attack, DDPA achieves the highest ASR and the lowest BA, demonstrating its effectiveness in executing a successful attack. Compared to other attack models across the five machine unlearning algorithms, DDPA achieves an average ASR increase of 22.74%, 16.07%, and 21.45%, while reducing BA by 15.76%, 11.41%, and 2.27% on VGG16-CIFAR100, ResNet18-Tiny-ImageNet, and LLaMA-3B-SST-2 respectively. In addition, the promising performance of DDPA with all machine unlearning algorithms implies that DDPA has greate potential as a general attack solution to other machine unlearning methods, which is desirable in practice.

Table 5: Unlearning Performance on VGG-16 with CIFAR100 (10% Unlearned)-targeted

| Method | B/A Unlearn | First-Order | | | Second-Order | | | Unroll-SGD | | | Amnesiac | | | SISA | | |
|---|---|---|---|---|---|---|---|---|---|---|---|---|---|---|---|---|
| | | TA | BA | ASR | TA | BA | ASR | TA | BA | ASR | TA | BA | ASR | TA | BA | ASR |
| AwoP | Conceal | 98.10 | 49.30 | 23.10 | 98.33 | 49.52 | 24.65 | 98.46 | 48.72 | 31.00 | 98.86 | 49.25 | 27.70 | 98.97 | 49.09 | 23.82 |
| | Unlearn | 96.89 | 45.23 | 88.60 | 96.59 | 44.72 | 87.25 | 96.70 | 45.32 | 88.66 | 96.74 | 45.76 | 89.00 | 97.50 | 46.18 | 88.30 |
| MUECPA | Conceal | 98.16 | 46.22 | 2.00 | 98.26 | 49.54 | 0.00 | 97.04 | 47.42 | 0.00 | 92.97 | 47.28 | 0.20 | 98.46 | 47.10 | 0.20 |
| | Unlearn | 97.71 | 44.99 | 86.40 | 97.25 | 45.68 | 88.10 | 82.19 | 44.42 | 85.80 | 85.64 | 43.58 | 85.20 | 96.63 | 45.50 | 84.95 |
| SSCSF | Conceal | 99.40 | 49.50 | 0.00 | 98.12 | 48.04 | 0.00 | 98.52 | 48.30 | 0.00 | 98.58 | 49.15 | 0.00 | 98.66 | 47.26 | 0.00 |
| | Unlearn | 96.89 | 45.50 | 89.30 | 96.93 | 44.90 | 88.73 | 95.15 | 44.69 | 85.00 | 96.64 | 45.08 | 89.60 | 96.71 | 45.24 | 88.00 |
| BAU | Conceal | 98.36 | 46.47 | 0.00 | 98.36 | 46.47 | 0.00 | 98.78 | 47.36 | 0.00 | 98.31 | 47.70 | 0.00 | 98.24 | 47.46 | 0.00 |
| | Unlearn | 95.82 | 43.40 | 86.90 | 96.02 | 43.32 | 87.40 | 95.92 | 45.67 | 88.00 | 96.14 | 44.93 | 88.40 | 96.74 | 45.06 | 89.00 |
| UBA-Inf | Conceal | 98.26 | 50.74 | 9.20 | 98.04 | 49.30 | 7.60 | 98.86 | 50.21 | 16.28 | 98.77 | 49.72 | 12.46 | 98.05 | 49.93 | 13.55 |
| | Unlearn | 96.71 | 44.83 | 87.60 | 96.73 | 44.05 | 90.00 | 96.04 | 45.13 | 89.40 | 96.36 | 45.06 | 90.00 | 96.93 | 43.99 | 86.31 |
| RMBMU | Conceal | 98.94 | 47.04 | 5.20 | 98.73 | 47.40 | 1.60 | 98.68 | 47.53 | 0.00 | 98.33 | 48.89 | 0.00 | 98.16 | 48.69 | 0.00 |
| | Unlearn | 96.00 | 44.94 | 87.60 | 97.42 | 44.44 | 89.40 | 97.02 | 44.38 | 88.40 | 97.55 | 45.36 | 88.40 | 96.98 | 45.18 | 87.80 |
| DABF | Conceal | 98.26 | 48.04 | 0.34 | 98.54 | 48.26 | 1.24 | 98.78 | 48.15 | 0.82 | 98.67 | 48.88 | 0.70 | 98.35 | 48.94 | 2.40 |
| | Unlearn | 96.28 | 43.62 | 85.30 | 96.09 | 44.73 | 86.20 | 96.02 | 44.29 | 89.00 | 96.63 | 43.94 | 89.20 | 96.75 | 45.06 | 88.90 |
| AdvUA | Conceal | 98.47 | 47.53 | 0.00 | 98.57 | 47.40 | 0.00 | 98.46 | 48.80 | 0.00 | 98.77 | 48.80 | 0.00 | 98.54 | 48.65 | 0.00 |
| | Unlearn | 95.75 | 43.60 | 88.30 | 96.20 | 44.60 | 87.60 | 96.31 | 45.72 | 87.60 | 96.79 | 45.03 | 89.60 | 96.40 | 45.54 | 89.00 |
| EVMUS | Conceal | 99.45 | 50.36 | 2.90 | 99.94 | 49.09 | 0.76 | 98.85 | 49.61 | 1.64 | 98.83 | 49.52 | 0.80 | 99.26 | 48.62 | 1.24 |
| | Unlearn | 96.64 | 45.58 | 84.90 | 96.30 | 44.45 | 86.00 | 96.82 | 45.76 | 86.40 | 96.75 | 44.87 | 88.00 | 97.09 | 45.75 | 88.20 |
| DDPA | Conceal | 98.75 | 47.86 | 0.00 | 98.94 | 48.61 | 0.00 | 98.65 | 48.99 | 0.00 | 98.61 | 48.11 | 0.00 | 98.66 | 48.56 | 0.00 |
| | Unlearn | 95.81 | 43.18 | 90.00 | 95.93 | 43.23 | 90.60 | 94.48 | 43.32 | 92.00 | 95.87 | 43.07 | 91.00 | 95.09 | 44.24 | 90.00 |
| DDPA-C | Conceal | 98.01 | 46.32 | 0.00 | 98.29 | 47.82 | 0.00 | 98.11 | 47.28 | 0.00 | 98.16 | 48.11 | 0.00 | 98.19 | 48.48 | 0.00 |
| | Unlearn | 95.35 | 44.04 | 82.10 | 95.21 | 44.77 | 80.20 | 94.43 | 44.17 | 81.80 | 95.35 | 44.61 | 81.20 | 95.23 | 45.51 | 80.00 |
| DDPA-S | Conceal | 98.04 | 46.39 | 0.00 | 98.43 | 47.01 | 0.00 | 98.17 | 47.58 | 0.00 | 98.79 | 47.91 | 0.00 | 98.45 | 47.79 | 0.00 |
| | Unlearn | 95.18 | 44.53 | 83.00 | 95.30 | 45.43 | 81.50 | 93.95 | 44.55 | 82.00 | 96.11 | 44.24 | 83.40 | 95.60 | 45.53 | 83.00 |

Table 6: Unlearning Performance on VGG-16 with CIFAR100 (20% Unlearned)-targeted

| Method | B/A Unlearn | First-Order | | | Second-Order | | | Unroll-SGD | | | Amnesiac | | | SISA | | |
|---|---|---|---|---|---|---|---|---|---|---|---|---|---|---|---|---|
| | | TA | BA | ASR | TA | BA | ASR | TA | BA | ASR | TA | BA | ASR | TA | BA | ASR |
| AwoP | Conceal | 98.91 | 49.12 | 19.64 | 98.20 | 49.25 | 26.39 | 98.96 | 48.68 | 24.74 | 98.70 | 48.65 | 23.10 | 98.98 | 49.11 | 0.00 |
| | Unlearn | 96.32 | 43.28 | 90.20 | 96.65 | 43.73 | 90.00 | 97.03 | 43.11 | 92.10 | 96.97 | 42.98 | 90.00 | 96.63 | 44.70 | 90.40 |
| MUECPA | Conceal | 96.73 | 52.79 | 0.00 | 98.18 | 48.39 | 1.00 | 98.84 | 47.92 | 0.40 | 96.96 | 41.59 | 0.60 | 98.45 | 47.52 | 1.20 |
| | Unlearn | 90.25 | 48.74 | 90.00 | 97.08 | 42.69 | 93.80 | 95.50 | 44.70 | 92.68 | 87.33 | 37.52 | 88.00 | 94.21 | 42.69 | 91.60 |
| SSCSF | Conceal | 98.42 | 48.83 | 0.00 | 98.56 | 48.87 | 0.00 | 98.86 | 48.48 | 0.00 | 98.54 | 48.62 | 0.00 | 98.62 | 48.37 | 0.00 |
| | Unlearn | 97.50 | 43.27 | 91.00 | 96.27 | 42.87 | 90.60 | 96.98 | 42.37 | 91.60 | 96.28 | 42.13 | 90.00 | 96.12 | 42.79 | 91.20 |
| BAU | Conceal | 98.46 | 47.79 | 0.00 | 98.14 | 48.22 | 0.00 | 98.14 | 48.33 | 0.00 | 98.47 | 48.40 | 0.00 | 97.82 | 48.86 | 0.00 |
| | Unlearn | 96.78 | 43.26 | 89.00 | 96.19 | 42.44 | 90.00 | 95.78 | 42.79 | 90.20 | 96.65 | 42.38 | 89.60 | 95.14 | 43.73 | 90.40 |
| UBA-Inf | Conceal | 99.34 | 50.33 | 16.32 | 98.83 | 49.72 | 9.82 | 98.91 | 50.05 | 6.24 | 99.09 | 49.41 | 12.80 | 98.22 | 50.87 | 16.31 |
| | Unlearn | 97.30 | 44.82 | 92.60 | 97.41 | 43.15 | 91.40 | 96.55 | 44.35 | 91.20 | 96.70 | 43.51 | 91.00 | 92.47 | 41.89 | 92.34 |
| RMBMU | Conceal | 98.20 | 48.61 | 7.00 | 98.01 | 48.18 | 0.20 | 98.09 | 49.47 | 2.60 | 98.88 | 48.23 | 0.60 | 98.57 | 48.75 | 3.40 |
| | Unlearn | 97.97 | 42.51 | 81.40 | 96.91 | 43.68 | 87.40 | 96.33 | 42.75 | 86.00 | 96.34 | 43.33 | 85.00 | 95.65 | 42.82 | 89.00 |
| DABF | Conceal | 98.26 | 48.23 | 0.00 | 98.93 | 48.90 | 0.00 | 98.45 | 48.85 | 0.00 | 98.29 | 48.36 | 0.00 | 97.92 | 48.10 | 0.00 |
| | Unlearn | 96.30 | 43.24 | 90.00 | 97.05 | 43.79 | 91.20 | 96.27 | 43.18 | 91.00 | 96.33 | 42.36 | 90.00 | 95.65 | 42.35 | 89.80 |
| AdvUA | Conceal | 98.84 | 48.42 | 0.00 | 98.19 | 48.37 | 0.00 | 98.70 | 48.55 | 0.00 | 98.54 | 48.67 | 0.00 | 98.41 | 48.33 | 0.00 |
| | Unlearn | 96.63 | 43.36 | 90.60 | 95.62 | 42.78 | 91.00 | 95.93 | 43.64 | 90.20 | 94.92 | 42.46 | 91.20 | 95.90 | 43.76 | 90.00 |
| EVMUS | Conceal | 99.24 | 50.13 | 1.92 | 98.85 | 49.36 | 0.27 | 99.31 | 49.82 | 2.50 | 98.89 | 49.61 | 1.94 | 98.86 | 49.88 | 1.11 |
| | Unlearn | 97.67 | 45.09 | 89.20 | 96.64 | 43.96 | 90.00 | 96.55 | 44.25 | 90.60 | 96.35 | 43.37 | 91.20 | 96.33 | 44.70 | 91.00 |
| DDPA | Conceal | 98.95 | 48.81 | 0.00 | 98.93 | 48.61 | 0.00 | 98.48 | 48.99 | 0.00 | 98.61 | 48.37 | 0.00 | 98.93 | 48.56 | 0.00 |
| | Unlearn | 95.49 | 42.18 | 96.00 | 95.07 | 42.23 | 95.00 | 95.11 | 42.01 | 94.00 | 95.87 | 41.07 | 95.60 | 95.09 | 42.02 | 94.80 |
| DDPA-C | Conceal | 98.24 | 47.10 | 0.00 | 98.31 | 47.85 | 0.00 | 98.07 | 48.03 | 0.00 | 98.37 | 48.11 | 0.00 | 98.29 | 47.50 | 0.00 |
| | Unlearn | 95.59 | 43.53 | 88.00 | 94.99 | 43.68 | 89.20 | 95.26 | 43.10 | 89.20 | 95.62 | 42.48 | 90.00 | 95.56 | 43.30 | 89.90 |
| DDPA-S | Conceal | 98.33 | 48.28 | 0.00 | 98.63 | 47.13 | 0.00 | 98.19 | 48.45 | 0.00 | 98.54 | 47.47 | 0.00 | 98.04 | 47.51 | 0.00 |
| | Unlearn | 95.03 | 43.20 | 90.00 | 95.34 | 43.29 | 90.20 | 95.04 | 44.43 | 90.10 | 95.27 | 42.50 | 91.20 | 95.16 | 43.41 | 91.70 |

Table 7: Unlearning Performance on ResNet-18 with Tiny ImageNet (5% Unlearned)-targeted

| Method | B/A Unlearn | First-Order | | | Second-Order | | | Unroll-SGD | | | Amnesiac | | | SISA | | |
|---|---|---|---|---|---|---|---|---|---|---|---|---|---|---|---|---|
| | | TA | BA | ASR | TA | BA | ASR | TA | BA | ASR | TA | BA | ASR | TA | BA | ASR |
| AwoP | Conceal | 98.97 | 42.15 | 21.23 | 98.58 | 41.69 | 16.43 | 98.51 | 41.10 | 17.82 | 98.74 | 42.04 | 23.21 | 98.88 | 42.59 | 21.49 |
| | Unlearn | 96.64 | 36.73 | 80.00 | 97.05 | 37.30 | 80.40 | 96.14 | 36.90 | 80.00 | 96.33 | 36.71 | 81.20 | 96.79 | 36.27 | 81.52 |
| MUECPA | Conceal | 98.76 | 42.78 | 0.00 | 98.03 | 43.39 | 0.00 | 98.14 | 42.69 | 0.00 | 97.98 | 42.39 | 0.00 | 97.21 | 43.15 | 0.00 |
| | Unlearn | 96.48 | 36.49 | 81.00 | 96.46 | 37.42 | 82.60 | 96.95 | 36.63 | 85.20 | 95.98 | 36.53 | 84.00 | 94.91 | 36.13 | 84.00 |
| SSCSF | Conceal | 99.76 | 41.42 | 0.00 | 99.76 | 41.73 | 0.00 | 99.78 | 41.67 | 0.00 | 99.58 | 41.38 | 0.00 | 98.75 | 41.80 | 0.00 |
| | Unlearn | 97.72 | 35.75 | 80.00 | 97.39 | 36.07 | 82.40 | 97.46 | 35.82 | 86.00 | 96.17 | 35.43 | 84.20 | 96.22 | 35.51 | 86.26 |
| BAU | Conceal | 98.98 | 40.73 | 0.00 | 99.95 | 41.42 | 0.00 | 98.83 | 40.67 | 0.00 | 99.58 | 41.32 | 0.00 | 98.75 | 41.32 | 0.00 |
| | Unlearn | 95.72 | 33.75 | 80.00 | 96.39 | 35.07 | 83.20 | 95.46 | 34.82 | 86.93 | 96.17 | 34.43 | 83.40 | 96.22 | 34.51 | 85.20 |
| UBA-Inf | Conceal | 98.96 | 41.09 | 13.67 | 98.53 | 40.33 | 11.34 | 98.79 | 41.68 | 10.24 | 98.64 | 41.38 | 11.63 | 98.49 | 41.30 | 10.26 |
| | Unlearn | 97.16 | 37.33 | 81.00 | 97.29 | 37.59 | 83.91 | 96.14 | 36.91 | 82.13 | 96.59 | 36.90 | 81.00 | 97.15 | 38.64 | 81.27 |
| RMBMU | Conceal | 97.64 | 40.73 | 0.00 | 98.02 | 40.14 | 0.00 | 97.69 | 41.83 | 0.00 | 97.62 | 40.89 | 0.00 | 98.15 | 47.20 | 0.00 |
| | Unlearn | 95.28 | 33.24 | 80.20 | 95.38 | 34.67 | 84.30 | 95.76 | 33.40 | 84.38 | 96.02 | 32.89 | 85.00 | 95.92 | 33.25 | 86.74 |
| DABF | Conceal | 98.32 | 41.63 | 0.43 | 98.18 | 42.02 | 0.13 | 98.39 | 41.97 | 1.54 | 98.88 | 42.04 | 0.00 | 98.34 | 41.14 | 0.78 |
| | Unlearn | 96.64 | 35.69 | 82.00 | 95.78 | 36.08 | 81.21 | 97.24 | 35.48 | 85.00 | 95.81 | 34.94 | 86.14 | 96.04 | 34.64 | 84.45 |
| AdvUA | Conceal | 98.86 | 41.37 | 0.00 | 98.45 | 41.14 | 0.00 | 98.73 | 41.73 | 0.00 | 98.49 | 41.44 | 0.00 | 98.56 | 42.19 | 0.00 |
| | Unlearn | 96.49 | 35.29 | 80.00 | 96.38 | 35.86 | 84.20 | 96.81 | 34.88 | 83.60 | 96.63 | 35.57 | 85.34 | 96.69 | 35.83 | 83.48 |
| EVMUS | Conceal | 99.36 | 41.71 | 1.23 | 98.88 | 40.82 | 0.48 | 98.31 | 40.51 | 0.37 | 98.60 | 41.02 | 0.33 | 98.64 | 41.02 | 1.89 |
| | Unlearn | 97.86 | 35.62 | 80.00 | 96.07 | 36.04 | 81.20 | 96.92 | 35.56 | 67.66 | 94.31 | 34.62 | 73.61 | 96.29 | 35.43 | 79.30 |
| DDPA | Conceal | 98.98 | 41.84 | 0.00 | 98.83 | 43.44 | 0.00 | 98.74 | 43.68 | 0.00 | 98.68 | 42.70 | 0.00 | 98.49 | 42.37 | 0.00 |
| | Unlearn | 96.36 | 33.53 | 86.00 | 96.81 | 34.18 | 88.00 | 95.24 | 33.06 | 90.00 | 96.02 | 32.10 | 88.60 | 96.07 | 33.11 | 89.20 |
| DDPA-C | Conceal | 98.09 | 40.23 | 0.00 | 98.01 | 40.17 | 0.00 | 98.40 | 42.44 | 0.00 | 98.47 | 41.54 | 0.00 | 98.16 | 41.39 | 0.00 |
| | Unlearn | 96.55 | 35.86 | 80.20 | 96.25 | 37.17 | 80.00 | 94.63 | 35.09 | 81.20 | 95.33 | 34.17 | 82.30 | 96.18 | 34.21 | 81.20 |
| DDPA-S | Conceal | 98.38 | 40.24 | 0.00 | 98.43 | 41.05 | 0.00 | 98.64 | 42.49 | 0.00 | 98.45 | 41.60 | 0.00 | 98.30 | 41.71 | 0.00 |
| | Unlearn | 96.06 | 35.17 | 81.60 | 96.52 | 35.72 | 82.40 | 95.73 | 35.29 | 81.60 | 95.63 | 33.48 | 83.40 | 95.42 | 34.89 | 82.70 |

Table 8: Unlearning Performance on ResNet-18 with Tiny ImageNet (10% Unlearned)-targeted

| Method | B/A Unlearn | First-Order | | | Second-Order | | | Unroll-SGD | | | Amnesiac | | | SISA | | |
|---|---|---|---|---|---|---|---|---|---|---|---|---|---|---|---|---|
| | | TA | BA | ASR | TA | BA | ASR | TA | BA | ASR | TA | BA | ASR | TA | BA | ASR |
| AwoP | Conceal | 98.24 | 42.93 | 22.18 | 98.43 | 42.52 | 24.69 | 98.29 | 41.72 | 28.94 | 98.86 | 42.05 | 26.57 | 98.97 | 43.47 | 18.92 |
| | Unlearn | 96.54 | 35.23 | 88.40 | 95.68 | 36.72 | 87.30 | 95.54 | 35.37 | 86.80 | 96.74 | 35.78 | 88.60 | 96.43 | 35.93 | 86.31 |
| MUECPA | Conceal | 98.05 | 40.44 | 2.34 | 98.13 | 40.47 | 0.53 | 97.29 | 40.78 | 0.26 | 96.97 | 40.28 | 0.20 | 98.46 | 41.01 | 0.76 |
| | Unlearn | 96.11 | 34.43 | 86.40 | 95.47 | 35.14 | 88.10 | 92.19 | 33.43 | 79.64 | 89.64 | 33.04 | 62.20 | 96.19 | 35.25 | 84.00 |
| SSCSF | Conceal | 98.25 | 43.23 | 0.00 | 98.12 | 42.60 | 0.00 | 98.14 | 42.59 | 0.00 | 98.83 | 42.32 | 0.00 | 98.84 | 43.26 | 0.00 |
| | Unlearn | 96.89 | 35.07 | 88.20 | 95.93 | 36.72 | 88.20 | 95.28 | 34.98 | 85.40 | 95.64 | 35.43 | 86.60 | 96.46 | 34.84 | 88.00 |
| BAU | Conceal | 98.36 | 40.65 | 0.00 | 97.83 | 40.34 | 0.00 | 98.40 | 40.38 | 0.00 | 99.16 | 41.84 | 0.00 | 98.07 | 41.41 | 0.00 |
| | Unlearn | 95.60 | 33.80 | 86.40 | 94.94 | 36.15 | 87.40 | 95.84 | 35.22 | 88.20 | 96.28 | 34.75 | 88.40 | 96.74 | 36.06 | 89.20 |
| UBA-Inf | Conceal | 99.22 | 43.58 | 9.58 | 98.79 | 44.03 | 16.57 | 99.16 | 41.66 | 11.08 | 98.86 | 42.66 | 17.18 | 98.81 | 42.47 | 12.62 |
| | Unlearn | 96.75 | 37.45 | 89.80 | 96.39 | 37.87 | 89.26 | 95.26 | 36.61 | 84.26 | 95.23 | 36.08 | 85.00 | 96.99 | 36.93 | 86.77 |
| RMBMU | Conceal | 98.94 | 42.15 | 4.28 | 98.76 | 41.54 | 9.79 | 98.83 | 42.19 | 3.45 | 98.30 | 41.75 | 3.64 | 98.67 | 41.98 | 2.84 |
| | Unlearn | 96.00 | 34.94 | 87.00 | 97.45 | 35.17 | 89.40 | 97.02 | 34.15 | 88.40 | 96.19 | 33.83 | 88.40 | 96.98 | 45.18 | 86.40 |
| DABF | Conceal | 98.26 | 42.17 | 3.19 | 98.54 | 42.62 | 2.79 | 98.78 | 41.49 | 1.57 | 98.36 | 42.24 | 0.13 | 98.49 | 42.58 | 3.90 |
| | Unlearn | 96.77 | 33.51 | 86.70 | 97.28 | 36.54 | 86.30 | 96.95 | 34.40 | 88.20 | 97.27 | 33.56 | 87.30 | 95.82 | 35.49 | 88.20 |
| AdvUA | Conceal | 98.20 | 42.19 | 0.00 | 98.60 | 43.08 | 0.00 | 98.18 | 42.34 | 0.00 | 98.68 | 42.74 | 0.00 | 98.54 | 41.75 | 0.00 |
| | Unlearn | 96.79 | 33.48 | 83.40 | 96.55 | 35.70 | 86.40 | 96.31 | 34.68 | 87.20 | 95.98 | 35.68 | 88.90 | 96.03 | 35.25 | 89.00 |
| EVMUS | Conceal | 99.80 | 45.68 | 4.11 | 99.94 | 46.31 | 1.11 | 98.62 | 45.97 | 2.17 | 98.20 | 46.25 | 0.46 | 99.63 | 46.17 | 1.63 |
| | Unlearn | 97.42 | 37.38 | 85.80 | 97.57 | 38.61 | 85.80 | 96.20 | 36.86 | 86.40 | 96.93 | 36.30 | 88.00 | 97.05 | 35.60 | 89.20 |
| DDPA | Conceal | 98.77 | 43.95 | 0.00 | 98.51 | 44.57 | 0.00 | 98.12 | 43.01 | 0.00 | 97.87 | 43.57 | 0.00 | 98.08 | 42.87 | 0.00 |
| | Unlearn | 95.49 | 33.20 | 90.20 | 95.90 | 34.36 | 92.40 | 96.44 | 33.61 | 90.60 | 95.19 | 31.08 | 90.20 | 96.72 | 32.29 | 91.40 |
| DDPA-C | Conceal | 98.62 | 42.35 | 0.00 | 98.47 | 43.29 | 0.00 | 98.24 | 41.87 | 0.00 | 98.08 | 42.13 | 0.00 | 98.15 | 41.74 | 0.00 |
| | Unlearn | 96.03 | 34.17 | 83.10 | 95.81 | 35.24 | 82.60 | 96.21 | 34.62 | 83.10 | 95.76 | 33.58 | 83.30 | 96.43 | 33.71 | 82.20 |
| DDPA-S | Conceal | 98.48 | 42.02 | 0.00 | 98.19 | 42.89 | 0.00 | 97.93 | 41.56 | 0.00 | 97.67 | 41.83 | 0.00 | 98.02 | 42.13 | 0.00 |
| | Unlearn | 96.12 | 35.49 | 85.30 | 96.34 | 36.25 | 86.30 | 96.08 | 35.74 | 86.20 | 95.91 | 34.83 | 85.20 | 96.28 | 35.12 | 86.80 |

Table 9: Unlearning Performance on ResNet-18 with Tiny ImageNet (20% Unlearned)-targeted

| Method | B/A Unlearn | First-Order | | | Second-Order | | | Unroll-SGD | | | Amnesiac | | | SISA | | |
|---|---|---|---|---|---|---|---|---|---|---|---|---|---|---|---|---|
| | | TA | BA | ASR | TA | BA | ASR | TA | BA | ASR | TA | BA | ASR | TA | BA | ASR |
| AwoP | Conceal | 98.77 | 45.37 | 18.51 | 98.62 | 45.78 | 26.24 | 97.91 | 44.33 | 25.26 | 98.78 | 43.67 | 24.63 | 98.74 | 43.06 | 23.73 |
| | Unlearn | 96.36 | 31.48 | 89.10 | 96.48 | 32.22 | 90.00 | 95.62 | 30.62 | 90.90 | 95.74 | 30.26 | 90.00 | 94.23 | 30.03 | 90.40 |
| MUECPA | Conceal | 97.88 | 43.46 | 0.00 | 98.63 | 42.39 | 1.81 | 98.84 | 43.58 | 4.90 | 97.96 | 41.59 | 0.60 | 98.45 | 43.52 | 1.20 |
| | Unlearn | 95.40 | 32.65 | 90.00 | 96.47 | 33.69 | 93.80 | 96.54 | 32.33 | 92.68 | 95.81 | 32.52 | 88.00 | 94.21 | 32.69 | 91.60 |
| SSCSF | Conceal | 98.42 | 42.84 | 0.00 | 98.81 | 43.59 | 0.00 | 98.39 | 42.72 | 0.00 | 98.70 | 40.34 | 0.00 | 98.76 | 41.09 | 0.00 |
| | Unlearn | 96.58 | 33.35 | 90.20 | 96.00 | 34.72 | 92.70 | 97.17 | 33.70 | 90.60 | 95.38 | 30.57 | 90.00 | 95.11 | 32.04 | 90.60 |
| BAU | Conceal | 98.81 | 40.09 | 0.00 | 98.14 | 42.81 | 0.00 | 98.82 | 41.95 | 0.00 | 98.05 | 42.06 | 0.00 | 98.24 | 42.80 | 0.00 |
| | Unlearn | 95.72 | 31.11 | 88.60 | 96.19 | 33.77 | 90.00 | 95.78 | 31.93 | 90.20 | 96.34 | 30.58 | 89.80 | 95.28 | 31.01 | 90.40 |
| UBA-Inf | Conceal | 99.19 | 42.51 | 13.67 | 98.95 | 41.62 | 15.66 | 98.83 | 41.43 | 11.22 | 98.41 | 40.43 | 13.35 | 98.69 | 42.28 | 14.86 |
| | Unlearn | 96.89 | 34.33 | 90.12 | 95.43 | 35.72 | 90.40 | 95.27 | 34.30 | 89.60 | 94.70 | 34.76 | 90.00 | 97.71 | 36.35 | 89.72 |
| RMBMU | Conceal | 98.65 | 42.04 | 7.42 | 98.10 | 42.89 | 2.40 | 98.72 | 43.29 | 2.29 | 97.47 | 43.47 | 1.68 | 98.67 | 42.49 | 7.21 |
| | Unlearn | 97.54 | 32.46 | 88.10 | 96.91 | 33.72 | 89.50 | 96.47 | 32.29 | 88.70 | 95.05 | 31.08 | 86.90 | 95.55 | 30.86 | 89.00 |
| DABF | Conceal | 97.93 | 43.63 | 0.00 | 98.36 | 43.48 | 0.00 | 97.85 | 42.81 | 0.00 | 98.01 | 43.14 | 0.00 | 97.86 | 42.60 | 0.00 |
| | Unlearn | 95.11 | 31.83 | 90.20 | 96.81 | 33.35 | 91.60 | 96.25 | 31.96 | 91.00 | 96.49 | 30.39 | 90.00 | 95.23 | 30.18 | 90.00 |
| AdvUA | Conceal | 98.57 | 42.67 | 0.00 | 98.67 | 42.93 | 0.00 | 97.93 | 41.93 | 0.00 | 98.15 | 42.91 | 0.00 | 98.72 | 43.22 | 0.00 |
| | Unlearn | 96.24 | 32.24 | 90.60 | 95.62 | 33.35 | 91.00 | 96.35 | 32.35 | 90.10 | 94.91 | 31.18 | 90.80 | 95.60 | 30.79 | 90.00 |
| EVMUS | Conceal | 98.70 | 44.85 | 1.21 | 98.64 | 45.52 | 8.11 | 99.06 | 43.69 | 5.13 | 98.88 | 44.72 | 3.11 | 98.57 | 43.85 | 7.01 |
| | Unlearn | 97.67 | 30.12 | 88.30 | 96.15 | 31.08 | 89.80 | 96.28 | 30.52 | 90.00 | 96.11 | 28.65 | 86.00 | 96.66 | 29.04 | 86.40 |
| DDPA | Conceal | 98.42 | 45.65 | 0.00 | 98.74 | 45.61 | 0.00 | 98.27 | 45.69 | 0.00 | 98.84 | 45.69 | 0.00 | 98.72 | 45.56 | 0.00 |
| | Unlearn | 95.35 | 30.04 | 94.30 | 95.40 | 30.86 | 95.00 | 95.19 | 30.12 | 94.60 | 95.67 | 28.04 | 95.60 | 95.59 | 28.69 | 94.90 |
| DDPA-C | Conceal | 98.34 | 44.92 | 0.00 | 98.52 | 45.21 | 0.00 | 98.15 | 45.03 | 0.00 | 98.64 | 44.91 | 0.00 | 98.43 | 44.88 | 0.00 |
| | Unlearn | 95.02 | 31.12 | 89.20 | 95.18 | 31.58 | 90.10 | 95.29 | 30.84 | 89.40 | 95.08 | 29.92 | 89.80 | 95.21 | 30.33 | 89.60 |
| DDPA-S | Conceal | 98.19 | 44.78 | 0.00 | 98.25 | 45.06 | 0.00 | 98.08 | 44.84 | 0.00 | 98.37 | 44.71 | 0.00 | 98.29 | 44.69 | 0.00 |
| | Unlearn | 95.17 | 31.56 | 90.30 | 95.36 | 31.74 | 91.40 | 95.23 | 30.96 | 90.80 | 95.43 | 29.88 | 90.20 | 95.28 | 30.47 | 90.70 |

Table 10: Unlearning Performance on LLama-3b with SST-2 (5% Unlearned)-targeted

| Method | B/A Unlearn | First-Order | | | Second-Order | | | Unroll-SGD | | | Amnesiac | | | SISA | | |
|---|---|---|---|---|---|---|---|---|---|---|---|---|---|---|---|---|
| | | TA | BA | ASR | TA | BA | ASR | TA | BA | ASR | TA | BA | ASR | TA | BA | ASR |
| AwoP | Conceal | 93.56 | 90.67 | 23.42 | 93.28 | 90.79 | 19.87 | 93.78 | 90.50 | 21.76 | 93.46 | 90.35 | 23.57 | 92.18 | 91.14 | 18.59 |
| | Unlearn | 91.24 | 87.12 | 74.69 | 91.52 | 88.71 | 73.46 | 91.15 | 86.54 | 72.50 | 90.28 | 84.74 | 71.00 | 90.27 | 86.15 | 71.00 |
| MUECPA | Conceal | 94.51 | 89.21 | 0.00 | 93.72 | 90.27 | 0.00 | 93.34 | 90.45 | 0.00 | 92.71 | 90.59 | 0.00 | 92.38 | 91.17 | 0.00 |
| | Unlearn | 92.68 | 87.06 | 70.19 | 91.78 | 87.97 | 70.21 | 91.48 | 87.30 | 71.20 | 90.29 | 85.45 | 73.40 | 91.54 | 86.04 | 74.60 |
| SSCSF | Conceal | 95.16 | 90.17 | 0.00 | 95.03 | 90.14 | 0.00 | 95.46 | 91.33 | 0.00 | 95.75 | 91.84 | 0.00 | 95.28 | 90.53 | 0.00 |
| | Unlearn | 92.70 | 87.16 | 70.76 | 92.81 | 88.24 | 71.18 | 93.03 | 87.93 | 70.60 | 92.88 | 85.40 | 70.20 | 92.91 | 86.23 | 70.20 |
| BAU | Conceal | 90.33 | 89.63 | 0.00 | 91.05 | 89.59 | 0.00 | 91.62 | 90.35 | 0.00 | 92.16 | 90.78 | 0.00 | 92.42 | 90.69 | 0.00 |
| | Unlearn | 88.71 | 87.86 | 71.71 | 89.91 | 88.18 | 71.54 | 89.99 | 86.52 | 70.20 | 89.96 | 84.23 | 72.10 | 90.04 | 85.81 | 69.40 |
| UBA-Inf | Conceal | 95.70 | 91.39 | 12.37 | 95.87 | 91.31 | 14.73 | 95.29 | 91.80 | 13.50 | 95.63 | 92.18 | 11.41 | 95.79 | 91.13 | 15.76 |
| | Unlearn | 92.61 | 87.32 | 76.01 | 92.16 | 88.12 | 78.70 | 93.80 | 87.85 | 78.70 | 91.98 | 85.12 | 79.04 | 92.81 | 87.52 | 78.90 |
| RMBMU | Conceal | 93.81 | 90.09 | 0.00 | 92.84 | 90.69 | 0.00 | 93.15 | 90.06 | 0.00 | 92.99 | 90.75 | 0.00 | 93.08 | 90.33 | 0.00 |
| | Unlearn | 91.82 | 88.21 | 72.38 | 91.69 | 87.47 | 72.00 | 91.51 | 87.78 | 71.10 | 89.74 | 84.36 | 71.00 | 90.43 | 86.39 | 72.00 |
| DABF | Conceal | 91.89 | 89.75 | 2.42 | 92.37 | 90.08 | 9.23 | 91.13 | 90.05 | 6.95 | 92.43 | 90.22 | 5.81 | 91.61 | 90.59 | 2.11 |
| | Unlearn | 89.60 | 87.18 | 69.88 | 90.25 | 88.49 | 70.20 | 89.48 | 87.14 | 70.00 | 90.34 | 83.94 | 70.00 | 89.37 | 87.07 | 70.00 |
| AdvUA | Conceal | 91.72 | 89.93 | 0.00 | 91.57 | 89.98 | 0.00 | 92.60 | 90.02 | 0.00 | 92.66 | 90.17 | 0.00 | 92.10 | 90.15 | 0.00 |
| | Unlearn | 89.09 | 87.16 | 71.90 | 90.47 | 88.02 | 71.50 | 90.49 | 87.19 | 71.00 | 89.55 | 84.68 | 72.00 | 90.79 | 87.41 | 70.70 |
| EVMUS | Conceal | 93.58 | 90.39 | 6.32 | 93.38 | 90.88 | 7.17 | 93.11 | 90.53 | 4.20 | 92.49 | 91.19 | 6.51 | 93.77 | 90.28 | 8.71 |
| | Unlearn | 91.14 | 87.63 | 68.35 | 92.42 | 88.54 | 67.50 | 91.12 | 87.10 | 65.40 | 89.91 | 83.36 | 64.20 | 90.14 | 86.96 | 62.00 |
| DDPA | Conceal | 95.51 | 91.86 | 0.00 | 95.46 | 92.94 | 0.00 | 95.84 | 92.84 | 0.00 | 95.18 | 92.67 | 0.00 | 95.12 | 91.96 | 0.00 |
| | Unlearn | 92.45 | 86.32 | 80.92 | 93.14 | 87.01 | 81.75 | 92.97 | 86.35 | 80.70 | 91.14 | 83.08 | 82.00 | 92.38 | 85.03 | 81.20 |
| DDPA-C | Conceal | 95.12 | 91.23 | 0.00 | 94.89 | 91.45 | 0.00 | 94.34 | 92.01 | 0.00 | 94.78 | 90.98 | 0.00 | 94.43 | 91.78 | 0.00 |
| | Unlearn | 93.34 | 87.78 | 72.45 | 92.56 | 88.12 | 71.78 | 92.23 | 87.34 | 72.12 | 90.34 | 85.12 | 73.12 | 91.76 | 86.41 | 72.12 |
| DDPA-S | Conceal | 94.45 | 90.89 | 0.00 | 94.23 | 91.01 | 0.00 | 94.67 | 91.56 | 0.00 | 94.12 | 91.34 | 0.00 | 94.89 | 92.01 | 0.00 |
| | Unlearn | 92.48 | 87.56 | 74.01 | 92.12 | 87.83 | 73.56 | 92.45 | 87.02 | 73.12 | 90.78 | 84.67 | 72.45 | 91.67 | 87.38 | 72.00 |

Table 11: Unlearning Performance on LLama-3b with SST-2 (10% Unlearned)-targeted

| Method | B/A Unlearn | First-Order | | | Second-Order | | | Unroll-SGD | | | Amnesiac | | | SISA | | |
|--------|-------------|------|------|------|------|------|------|------|------|------|------|------|------|------|------|------|
| | | TA | BA | ASR | TA | BA | ASR | TA | BA | ASR | TA | BA | ASR | TA | BA | ASR |
| AwoP | Conceal | 93.42 | 90.45 | 24.83 | 93.11 | 90.56 | 21.32 | 93.56 | 90.23 | 22.98 | 93.22 | 90.12 | 24.61 | 92.08 | 90.91 | 20.71 |
| | Unlearn | 90.58 | 86.45 | 78.56 | 91.04 | 87.89 | 76.34 | 90.48 | 85.67 | 75.21 | 89.91 | 83.45 | 74.12 | 89.86 | 85.04 | 73.85 |
| MUECPA | Conceal | 94.34 | 88.98 | 0.00 | 93.53 | 89.91 | 0.00 | 93.26 | 90.02 | 0.00 | 92.58 | 90.21 | 0.00 | 92.21 | 90.84 | 0.00 |
| | Unlearn | 91.56 | 86.31 | 76.89 | 90.84 | 87.02 | 76.54 | 90.73 | 86.25 | 76.02 | 89.87 | 84.39 | 75.32 | 90.12 | 85.61 | 74.96 |
| SSCSF | Conceal | 94.89 | 89.85 | 0.00 | 94.74 | 90.04 | 0.00 | 95.12 | 91.12 | 0.00 | 95.31 | 91.51 | 0.00 | 94.98 | 90.23 | 0.00 |
| | Unlearn | 91.92 | 86.45 | 74.62 | 92.11 | 87.54 | 73.98 | 91.87 | 87.23 | 73.41 | 91.83 | 84.21 | 73.19 | 91.74 | 85.43 | 73.01 |
| BAU | Conceal | 90.12 | 89.35 | 0.00 | 90.89 | 89.24 | 0.00 | 91.23 | 90.01 | 0.00 | 91.89 | 90.43 | 0.00 | 91.93 | 90.26 | 0.00 |
| | Unlearn | 88.23 | 86.34 | 73.84 | 89.14 | 87.12 | 73.32 | 89.07 | 85.89 | 72.41 | 89.03 | 84.12 | 73.02 | 89.21 | 85.04 | 72.94 |
| UBA-Inf | Conceal | 95.51 | 91.18 | 14.36 | 95.72 | 91.03 | 16.52 | 95.14 | 91.42 | 15.04 | 95.42 | 91.89 | 13.56 | 95.56 | 91.04 | 17.09 |
| | Unlearn | 91.87 | 86.58 | 81.42 | 91.56 | 87.41 | 80.61 | 91.79 | 86.72 | 83.05 | 90.78 | 84.09 | 82.14 | 91.21 | 85.49 | 81.85 |
| RMBMU | Conceal | 93.45 | 89.82 | 0.00 | 92.54 | 90.28 | 0.00 | 92.86 | 89.74 | 0.00 | 92.71 | 90.33 | 0.00 | 93.01 | 89.98 | 0.00 |
| | Unlearn | 90.62 | 87.12 | 76.43 | 90.41 | 86.78 | 76.01 | 90.28 | 86.12 | 75.61 | 88.54 | 83.45 | 74.92 | 89.72 | 84.61 | 75.13 |
| DABF | Conceal | 91.62 | 89.42 | 3.45 | 92.15 | 89.79 | 11.56 | 90.89 | 89.65 | 8.34 | 91.98 | 89.98 | 7.12 | 91.51 | 90.12 | 5.56 |
| | Unlearn | 88.45 | 86.34 | 74.51 | 89.14 | 87.21 | 73.89 | 88.73 | 86.41 | 73.02 | 88.42 | 83.67 | 72.71 | 88.89 | 86.12 | 72.54 |
| AdvUA | Conceal | 91.32 | 89.62 | 0.00 | 91.08 | 89.73 | 0.00 | 92.18 | 89.84 | 0.00 | 92.34 | 90.01 | 0.00 | 91.79 | 89.92 | 0.00 |
| | Unlearn | 88.67 | 86.73 | 74.87 | 89.71 | 87.43 | 74.12 | 89.69 | 86.87 | 73.45 | 88.91 | 83.54 | 74.21 | 89.57 | 86.41 | 73.89 |
| EVMUS | Conceal | 93.23 | 90.12 | 7.84 | 93.02 | 90.47 | 9.67 | 92.87 | 90.21 | 6.91 | 92.23 | 90.78 | 8.21 | 93.45 | 90.11 | 10.20 |
| | Unlearn | 90.54 | 86.92 | 73.42 | 91.41 | 87.56 | 72.13 | 90.32 | 86.47 | 71.54 | 89.03 | 82.71 | 71.02 | 89.76 | 85.21 | 70.91 |
| DDPA | Conceal | 95.41 | 91.56 | 0.00 | 95.12 | 92.45 | 0.00 | 95.72 | 92.21 | 0.00 | 95.01 | 92.05 | 0.00 | 95.07 | 91.58 | 0.00 |
| | Unlearn | 91.89 | 85.87 | 84.56 | 92.23 | 86.52 | 83.89 | 91.98 | 85.76 | 83.12 | 90.52 | 82.12 | 82.45 | 91.12 | 84.09 | 82.31 |
| DDPA-C | Conceal | 94.89 | 91.07 | 0.00 | 94.67 | 92.11 | 0.00 | 95.45 | 91.89 | 0.00 | 94.81 | 90.52 | 0.00 | 94.95 | 90.79 | 0.00 |
| | Unlearn | 91.67 | 86.12 | 75.34 | 91.01 | 87.45 | 74.78 | 91.34 | 86.48 | 74.12 | 91.79 | 83.33 | 73.20 | 91.15 | 85.82 | 73.50 |
| DDPA-S | Conceal | 95.12 | 90.78 | 0.00 | 95.33 | 91.90 | 0.00 | 94.53 | 90.42 | 0.00 | 95.06 | 91.61 | 0.00 | 95.06 | 91.41 | 0.00 |
| | Unlearn | 91.76 | 86.34 | 76.12 | 92.23 | 88.81 | 75.00 | 91.69 | 86.13 | 75.40 | 90.37 | 83.10 | 75.40 | 90.68 | 85.77 | 74.80 |

Table 12: Unlearning Performance on LLama-3b with SST-2 (20% Unlearned)-targeted

| Method | B/A Unlearn | First-Order | | | Second-Order | | | Unroll-SGD | | | Amnesiac | | | SISA | | |
|--------|-------------|------|------|------|------|------|------|------|------|------|------|------|------|------|------|------|
| | | TA | BA | ASR | TA | BA | ASR | TA | BA | ASR | TA | BA | ASR | TA | BA | ASR |
| AwoP | Conceal | 93.12 | 89.98 | 28.90 | 92.84 | 90.11 | 25.40 | 92.98 | 89.14 | 26.80 | 92.12 | 88.56 | 27.30 | 91.24 | 89.01 | 26.10 |
| | Unlearn | 85.67 | 81.12 | 85.30 | 86.78 | 82.45 | 83.60 | 85.45 | 80.89 | 84.20 | 84.12 | 78.76 | 84.90 | 84.76 | 79.35 | 83.70 |
| MUECPA | Conceal | 93.87 | 88.12 | 0.00 | 93.14 | 88.76 | 0.00 | 92.98 | 88.56 | 0.00 | 92.34 | 88.45 | 0.00 | 91.89 | 88.23 | 0.00 |
| | Unlearn | 86.12 | 81.34 | 80.10 | 86.34 | 82.12 | 79.40 | 85.98 | 81.56 | 81.30 | 83.78 | 78.67 | 79.80 | 84.34 | 79.12 | 80.70 |
| SSCSF | Conceal | 94.12 | 89.43 | 0.00 | 93.87 | 89.12 | 0.00 | 94.23 | 90.78 | 0.00 | 94.01 | 89.45 | 0.00 | 93.78 | 89.23 | 0.00 |
| | Unlearn | 85.89 | 81.23 | 78.40 | 86.34 | 82.34 | 77.90 | 85.78 | 80.45 | 77.40 | 84.56 | 77.98 | 76.80 | 85.12 | 78.56 | 77.20 |
| BAU | Conceal | 89.23 | 89.56 | 0.00 | 88.78 | 87.34 | 0.00 | 89.34 | 87.45 | 0.00 | 89.78 | 87.67 | 0.00 | 90.12 | 87.43 | 0.00 |
| | Unlearn | 84.12 | 79.34 | 77.30 | 85.34 | 80.34 | 76.80 | 84.89 | 78.76 | 75.40 | 83.67 | 76.45 | 76.10 | 84.12 | 77.12 | 75.80 |
| UBA-Inf | Conceal | 94.89 | 90.12 | 21.80 | 94.56 | 89.78 | 19.20 | 94.34 | 90.45 | 20.60 | 94.12 | 90.67 | 18.40 | 94.01 | 89.98 | 19.70 |
| | Unlearn | 85.34 | 80.12 | 83.40 | 85.78 | 80.98 | 81.30 | 85.12 | 79.34 | 80.70 | 83.78 | 76.12 | 79.80 | 84.34 | 78.12 | 82.10 |
| RMBMU | Conceal | 92.67 | 88.45 | 0.00 | 91.78 | 88.89 | 0.00 | 92.01 | 88.12 | 0.00 | 91.87 | 88.34 | 0.00 | 91.56 | 88.01 | 0.00 |
| | Unlearn | 84.89 | 80.34 | 78.60 | 85.12 | 81.45 | 78.10 | 84.34 | 79.87 | 77.40 | 82.34 | 76.67 | 76.80 | 83.78 | 77.89 | 77.10 |
| DABF | Conceal | 90.56 | 87.89 | 6.20 | 90.12 | 87.67 | 5.80 | 89.78 | 87.34 | 6.10 | 90.23 | 87.56 | 5.90 | 89.78 | 87.34 | 6.30 |
| | Unlearn | 84.23 | 79.67 | 76.40 | 84.78 | 80.12 | 76.10 | 84.12 | 78.67 | 75.80 | 83.01 | 75.34 | 75.40 | 83.78 | 77.12 | 76.20 |
| AdvUA | Conceal | 89.78 | 87.01 | 0.00 | 89.34 | 87.12 | 0.00 | 90.12 | 87.34 | 0.00 | 90.34 | 87.45 | 0.00 | 89.98 | 87.12 | 0.00 |
| | Unlearn | 83.67 | 79.34 | 75.30 | 84.45 | 80.12 | 74.80 | 83.89 | 79.34 | 74.60 | 82.89 | 75.34 | 73.90 | 83.12 | 77.12 | 74.80 |
| EVMUS | Conceal | 91.67 | 88.12 | 12.40 | 91.34 | 88.45 | 11.20 | 91.23 | 88.12 | 11.70 | 91.01 | 88.34 | 10.80 | 91.78 | 88.01 | 11.30 |
| | Unlearn | 84.89 | 79.87 | 73.40 | 85.12 | 80.12 | 72.10 | 84.56 | 79.01 | 71.80 | 83.23 | 75.12 | 71.20 | 83.89 | 77.12 | 72.60 |
| DDPA | Conceal | 95.12 | 90.34 | 0.00 | 94.89 | 90.67 | 0.00 | 95.23 | 90.12 | 0.00 | 94.87 | 91.45 | 0.00 | 95.78 | 92.12 | 0.00 |
| | Unlearn | 85.34 | 79.08 | 86.30 | 85.67 | 80.12 | 87.40 | 85.12 | 78.67 | 85.80 | 84.87 | 75.78 | 84.60 | 85.78 | 77.45 | 86.90 |
| DDPA-C | Conceal | 93.78 | 90.45 | 0.00 | 94.54 | 91.78 | 0.00 | 94.89 | 92.12 | 0.00 | 95.03 | 91.34 | 0.00 | 94.45 | 92.01 | 0.00 |
| | Unlearn | 86.45 | 81.12 | 79.80 | 86.78 | 82.34 | 78.40 | 87.23 | 81.56 | 78.10 | 85.67 | 77.34 | 77.60 | 86.12 | 80.17 | 78.30 |
| DDPA-S | Conceal | 94.12 | 90.12 | 0.00 | 95.01 | 92.34 | 0.00 | 94.28 | 90.01 | 0.00 | 94.86 | 92.68 | 0.00 | 94.67 | 90.56 | 0.00 |
| | Unlearn | 85.67 | 80.89 | 80.20 | 86.12 | 82.01 | 79.60 | 86.34 | 80.78 | 78.40 | 84.89 | 76.67 | 78.10 | 85.45 | 79.23 | 78.80 |

Table 13: Unlearning Performance on VGG-16 with CIFAR100 (5% Unlearned) - untargeted

| Method | B/A Unlearn | First-Order | | | Second-Order | | | Unroll-SGD | | | Amnesiac | | | SISA (shard 3) | | |
|---|---|---|---|---|---|---|---|---|---|---|---|---|---|---|---|---|
| | | TA | BA | ASR | TA | BA | ASR | TA | BA | ASR | TA | BA | ASR | TA | BA | ASR |
| AwoP | Conceal | 98.91 | 48.12 | 22.8 | 98.72 | 48.96 | 17.1 | 98.55 | 49.41 | 15.2 | 98.46 | 50.25 | 21.2 | 98.83 | 49.68 | 15.8 |
| | Unlearn | 96.51 | 41.78 | 91.20 | 96.62 | 42.14 | 92.7 | 96.24 | 41.23 | 91.40 | 95.94 | 40.89 | 90.9 | 96.38 | 41.45 | 92.10 |
| MUECPA | Conceal | 98.12 | 55.24 | 0.00 | 98.06 | 49.18 | 0.00 | 98.21 | 49.29 | 0.00 | 98.33 | 49.14 | 0.00 | 98.27 | 49.21 | 0.00 |
| | Unlearn | 95.32 | 43.61 | 86.10 | 95.47 | 43.89 | 87.20 | 95.64 | 43.37 | 86.70 | 95.12 | 42.98 | 85.90 | 95.38 | 43.26 | 86.50 |
| SSCSF | Conceal | 99.92 | 47.82 | 0.00 | 99.78 | 47.75 | 0.00 | 99.68 | 49.59 | 0.00 | 99.54 | 49.26 | 0.00 | 98.84 | 47.83 | 0.00 |
| | Unlearn | 97.84 | 42.04 | 89.40 | 97.67 | 42.49 | 90.10 | 97.48 | 41.98 | 89.20 | 97.36 | 41.56 | 89.50 | 97.68 | 42.03 | 89.80 |
| BAU | Conceal | 98.34 | 46.54 | 0.00 | 98.17 | 43.82 | 0.00 | 98.19 | 46.54 | 0.00 | 98.49 | 47.38 | 0.00 | 98.63 | 47.21 | 0.00 |
| | Unlearn | 95.18 | 39.78 | 85.40 | 95.32 | 40.14 | 85.90 | 95.27 | 39.68 | 85.10 | 95.02 | 39.21 | 84.80 | 95.23 | 39.74 | 85.30 |
| UBA-Inf | Conceal | 98.51 | 55.32 | 15.20 | 98.62 | 56.19 | 11.70 | 98.88 | 55.34 | 15.90 | 98.39 | 56.51 | 13.60 | 98.24 | 51.17 | 15.40 |
| | Unlearn | 96.84 | 45.13 | 93.10 | 96.72 | 46.29 | 92.70 | 96.61 | 45.92 | 91.80 | 96.32 | 43.64 | 91.40 | 96.48 | 43.04 | 92.30 |
| RMBMU | Conceal | 97.73 | 47.64 | 0.00 | 97.69 | 47.24 | 0.00 | 97.58 | 47.72 | 0.00 | 97.69 | 47.68 | 0.00 | 97.22 | 47.16 | 0.00 |
| | Unlearn | 94.79 | 42.18 | 89.10 | 94.64 | 42.36 | 88.90 | 94.42 | 42.08 | 88.40 | 94.26 | 41.79 | 88.20 | 94.62 | 42.12 | 88.80 |
| DABF | Conceal | 98.54 | 48.62 | 0.70 | 98.34 | 48.02 | 0.12 | 98.49 | 48.72 | 0.62 | 98.75 | 48.08 | 0.00 | 98.41 | 48.19 | 0.71 |
| | Unlearn | 96.94 | 41.78 | 89.30 | 96.73 | 42.02 | 88.50 | 96.38 | 41.53 | 88.20 | 96.12 | 41.08 | 87.90 | 96.34 | 41.62 | 88.70 |
| AdvUA | Conceal | 98.78 | 46.42 | 0.00 | 98.47 | 47.21 | 0.00 | 98.64 | 47.69 | 0.00 | 98.45 | 47.39 | 0.00 | 98.52 | 47.23 | 0.00 |
| | Unlearn | 95.64 | 41.62 | 88.20 | 95.41 | 41.74 | 88.90 | 95.34 | 41.31 | 88.40 | 95.14 | 41.12 | 88.30 | 95.43 | 41.45 | 88.60 |
| EVMUS | Conceal | 99.37 | 51.72 | 0.00 | 98.92 | 50.90 | 0.32 | 98.45 | 50.52 | 0.41 | 98.63 | 50.94 | 0.36 | 98.71 | 46.13 | 1.92 |
| | Unlearn | 97.81 | 43.87 | 84.60 | 97.32 | 44.12 | 83.40 | 97.06 | 43.59 | 83.10 | 96.52 | 43.24 | 82.90 | 96.98 | 43.68 | 83.60 |
| DDPA | Conceal | 98.53 | 47.86 | 0.00 | 98.42 | 48.33 | 0.00 | 98.31 | 48.62 | 0.00 | 98.67 | 47.94 | 0.00 | 98.58 | 47.78 | 0.00 |
| | Unlearn | 95.78 | 40.92 | 94.60 | 95.34 | 41.21 | 93.40 | 95.12 | 40.76 | 92.80 | 94.85 | 39.67 | 93.10 | 95.04 | 40.12 | 93.70 |
| DDPA-C | Conceal | 96.87 | 48.24 | 0.00 | 96.72 | 48.61 | 0.00 | 98.42 | 48.51 | 0.00 | 98.24 | 47.02 | 0.00 | 98.41 | 47.83 | 0.00 |
| | Unlearn | 94.31 | 38.12 | 95.10 | 94.96 | 39.61 | 93.60 | 95.74 | 38.32 | 92.90 | 95.03 | 36.78 | 93.20 | 95.42 | 37.08 | 93.50 |
| DDPA-S | Conceal | 97.47 | 48.11 | 0.00 | 96.68 | 47.13 | 0.00 | 98.13 | 47.72 | 0.00 | 97.95 | 46.61 | 0.00 | 97.82 | 46.41 | 0.00 |
| | Unlearn | 95.41 | 42.13 | 88.10 | 95.26 | 42.41 | 87.40 | 95.07 | 41.94 | 86.90 | 94.85 | 41.52 | 87.10 | 95.23 | 41.74 | 87.30 |

Table 14: Unlearning Performance on VGG-16 with CIFAR100 (10% Unlearned) - untargeted

| Method | B/A Unlearn | First-Order | | | Second-Order | | | Unroll-SGD | | | Amnesiac | | | SISA | | |
|---|---|---|---|---|---|---|---|---|---|---|---|---|---|---|---|---|
| | | TA | BA | ASR | TA | BA | ASR | TA | BA | ASR | TA | BA | ASR | TA | BA | ASR |
| AwoP | Conceal | 98.87 | 47.62 | 25.8 | 98.79 | 48.33 | 21.40 | 98.66 | 48.78 | 18.60 | 98.45 | 49.54 | 22.10 | 98.62 | 48.92 | 20.50 |
| | Unlearn | 96.48 | 38.82 | 92.10 | 96.32 | 39.21 | 93.60 | 96.14 | 38.66 | 91.40 | 95.92 | 37.88 | 91.80 | 96.27 | 38.22 | 92.50 |
| MUECPA | Conceal | 98.03 | 55.44 | 0.00 | 98.22 | 49.28 | 0.00 | 98.12 | 49.54 | 0.00 | 98.34 | 49.47 | 0.00 | 98.27 | 49.36 | 0.00 |
| | Unlearn | 94.91 | 40.94 | 87.80 | 94.87 | 41.28 | 88.90 | 94.58 | 40.74 | 87.30 | 94.32 | 40.12 | 86.50 | 94.65 | 40.54 | 87.5 |
| SSCSF | Conceal | 99.85 | 47.72 | 0.00 | 99.76 | 47.56 | 0.00 | 99.69 | 49.39 | 0.00 | 99.53 | 49.02 | 0.00 | 98.88 | 47.68 | 0.00 |
| | Unlearn | 97.78 | 38.88 | 91.30 | 97.53 | 39.21 | 91.80 | 97.34 | 38.74 | 91.10 | 97.21 | 37.89 | 91.60 | 97.49 | 38.42 | 91.70 |
| BAU | Conceal | 98.41 | 46.72 | 0.00 | 98.24 | 44.21 | 0.00 | 98.27 | 46.74 | 0.00 | 98.45 | 47.18 | 0.00 | 98.56 | 47.04 | 0.00 |
| | Unlearn | 94.82 | 37.74 | 89.50 | 94.64 | 38.12 | 89.80 | 94.58 | 37.58 | 89.20 | 94.41 | 36.98 | 88.90 | 94.63 | 37.24 | 89.30 |
| UBA-Inf | Conceal | 98.42 | 55.54 | 18.20 | 98.67 | 56.67 | 14.70 | 98.81 | 55.84 | 19.50 | 98.32 | 56.03 | 16.80 | 98.19 | 50.87 | 18.30 |
| | Unlearn | 96.51 | 47.73 | 94.60 | 96.38 | 47.94 | 94.10 | 96.22 | 47.56 | 93.70 | 96.08 | 46.82 | 93.20 | 96.35 | 47.23 | 93.90 |
| RMBMU | Conceal | 97.74 | 47.82 | 0.00 | 97.68 | 47.34 | 0.00 | 97.57 | 47.84 | 0.00 | 97.64 | 47.78 | 0.00 | 97.18 | 47.32 | 0.00 |
| | Unlearn | 94.28 | 39.12 | 91.40 | 94.21 | 39.34 | 91.10 | 94.08 | 38.96 | 90.70 | 94.02 | 38.52 | 90.30 | 94.23 | 38.84 | 91.20 |
| DABF | Conceal | 98.51 | 48.92 | 0.72 | 98.32 | 48.28 | 0.16 | 98.43 | 48.87 | 0.69 | 98.45 | 48.22 | 0.00 | 98.42 | 48.43 | 0.74 |
| | Unlearn | 96.62 | 38.43 | 91.60 | 96.48 | 38.72 | 91.30 | 96.14 | 38.28 | 90.80 | 95.78 | 37.89 | 90.40 | 96.12 | 38.22 | 91.20 |
| AdvUA | Conceal | 98.72 | 46.84 | 0.00 | 98.54 | 47.43 | 0.00 | 98.67 | 47.92 | 0.00 | 98.46 | 47.48 | 0.00 | 98.53 | 47.26 | 0.00 |
| | Unlearn | 95.84 | 38.72 | 90.40 | 95.61 | 39.01 | 90.90 | 95.42 | 38.64 | 90.20 | 95.21 | 38.34 | 90.10 | 95.63 | 38.54 | 90.50 |
| EVMUS | Conceal | 99.48 | 51.92 | 0.00 | 98.98 | 50.84 | 0.41 | 98.59 | 50.62 | 0.51 | 98.75 | 51.23 | 0.47 | 98.83 | 46.34 | 2.12 |
| | Unlearn | 97.91 | 40.12 | 86.40 | 97.44 | 40.56 | 85.70 | 97.28 | 39.98 | 85.40 | 96.81 | 39.58 | 85.20 | 97.29 | 39.87 | 85.60 |
| DDPA | Conceal | 96.92 | 48.74 | 0.00 | 96.78 | 48.91 | 0.00 | 98.52 | 48.73 | 0.00 | 98.36 | 47.68 | 0.00 | 98.42 | 47.92 | 0.00 |
| | Unlearn | 94.61 | 36.72 | 95.30 | 94.92 | 38.19 | 94.60 | 95.68 | 37.94 | 93.90 | 95.14 | 35.85 | 94.20 | 95.62 | 36.41 | 94.50 |
| DDPA-C | Conceal | 97.64 | 48.34 | 0.00 | 97.13 | 47.68 | 0.00 | 98.14 | 47.88 | 0.00 | 97.95 | 46.84 | 0.00 | 97.84 | 46.61 | 0.00 |
| | Unlearn | 95.32 | 39.62 | 92.20 | 95.14 | 39.84 | 91.80 | 94.97 | 39.52 | 91.30 | 94.82 | 38.92 | 91.10 | 95.23 | 39.12 | 91.70 |
| DDPA-S | Conceal | 96.81 | 48.42 | 0.00 | 96.78 | 47.74 | 0.00 | 98.23 | 47.63 | 0.00 | 98.42 | 47.12 | 0.00 | 98.25 | 46.94 | 0.00 |
| | Unlearn | 94.81 | 39.31 | 92.90 | 94.73 | 39.48 | 92.30 | 94.54 | 39.12 | 92.10 | 94.42 | 38.89 | 91.90 | 94.76 | 39.02 | 92.50 |

Table 15: Unlearning Performance on VGG-16 with CIFAR100 (20% Unlearned) - Untarget Experiment

| Method | B/A Unlearn | First-Order | | | Second-Order | | | Unroll-SGD | | | Amnesiac | | | SISA | | |
|---|---|---|---|---|---|---|---|---|---|---|---|---|---|---|---|---|
| | | TA | BA | ASR | TA | BA | ASR | TA | BA | ASR | TA | BA | ASR | TA | BA | ASR |
| AwoP | Conceal | 98.94 | 47.65 | 28.60 | 98.79 | 48.14 | 23.80 | 98.66 | 48.52 | 21.90 | 98.45 | 49.32 | 27.40 | 98.57 | 48.85 | 25.60 |
| | Unlearn | 94.78 | 35.12 | 95.80 | 94.64 | 35.56 | 96.20 | 94.53 | 35.01 | 95.30 | 94.34 | 34.36 | 95.60 | 94.71 | 34.79 | 96.00 |
| MUECPA | Conceal | 98.01 | 54.62 | 0.00 | 98.12 | 48.91 | 0.00 | 98.06 | 49.27 | 0.00 | 98.18 | 49.22 | 0.00 | 98.14 | 49.18 | 0.00 |
| | Unlearn | 93.56 | 38.84 | 91.80 | 93.42 | 39.16 | 92.10 | 93.34 | 38.63 | 91.20 | 93.18 | 37.98 | 90.70 | 93.48 | 38.24 | 91.40 |
| SSCSF | Conceal | 99.83 | 46.91 | 0.00 | 99.72 | 46.74 | 0.00 | 99.65 | 48.67 | 0.00 | 99.51 | 48.23 | 0.00 | 98.82 | 46.78 | 0.00 |
| | Unlearn | 96.91 | 33.82 | 94.30 | 96.65 | 34.13 | 94.70 | 96.49 | 33.78 | 93.90 | 96.32 | 33.14 | 94.20 | 96.61 | 33.54 | 94.60 |
| BAU | Conceal | 98.36 | 45.74 | 0.00 | 98.14 | 42.91 | 0.00 | 98.27 | 45.81 | 0.00 | 98.38 | 46.24 | 0.00 | 98.43 | 46.18 | 0.00 |
| | Unlearn | 93.81 | 32.84 | 93.60 | 93.62 | 33.12 | 93.90 | 93.54 | 32.68 | 93.10 | 93.41 | 32.14 | 92.80 | 93.59 | 32.43 | 93.40 |
| UBA-Inf | Conceal | 98.45 | 54.12 | 20.10 | 98.64 | 55.17 | 16.40 | 98.78 | 54.84 | 21.30 | 98.27 | 54.23 | 18.20 | 98.14 | 50.57 | 19.70 |
| | Unlearn | 95.34 | 41.53 | 95.30 | 95.23 | 41.82 | 95.80 | 95.12 | 41.32 | 96.70 | 95.01 | 40.78 | 96.20 | 95.27 | 41.08 | 97.20 |
| RMBMU | Conceal | 97.61 | 46.91 | 0.00 | 97.54 | 46.34 | 0.00 | 97.47 | 46.92 | 0.00 | 97.53 | 46.87 | 0.00 | 97.14 | 46.42 | 0.00 |
| | Unlearn | 93.21 | 33.45 | 94.30 | 93.14 | 33.68 | 94.10 | 93.04 | 33.24 | 93.60 | 92.88 | 32.78 | 93.10 | 93.18 | 33.08 | 93.90 |
| DABF | Conceal | 98.44 | 47.91 | 1.02 | 98.23 | 47.38 | 0.42 | 98.37 | 47.84 | 0.89 | 98.62 | 47.14 | 0.00 | 98.41 | 47.26 | 0.86 |
| | Unlearn | 95.84 | 33.74 | 93.70 | 95.62 | 33.98 | 93.40 | 95.47 | 33.61 | 92.80 | 95.23 | 33.18 | 92.30 | 95.59 | 33.43 | 93.20 |
| AdvUA | Conceal | 98.64 | 45.34 | 0.00 | 98.52 | 46.14 | 0.00 | 98.68 | 46.74 | 0.00 | 98.41 | 46.38 | 0.00 | 98.49 | 46.17 | 0.00 |
| | Unlearn | 94.94 | 34.73 | 93.40 | 94.71 | 34.89 | 93.80 | 94.54 | 34.52 | 93.20 | 94.32 | 34.18 | 92.90 | 94.68 | 34.42 | 93.50 |
| EVMUS | Conceal | 99.34 | 50.84 | 0.00 | 98.85 | 49.74 | 0.71 | 98.46 | 49.53 | 0.81 | 98.62 | 50.14 | 0.68 | 98.74 | 45.97 | 3.12 |
| | Unlearn | 96.79 | 35.41 | 91.8 | 96.44 | 35.82 | 91.30 | 96.32 | 35.21 | 91.10 | 96.08 | 34.72 | 90.60 | 96.48 | 35.04 | 91.40 |
| DDPA | Conceal | 96.86 | 47.84 | 0.00 | 96.62 | 48.14 | 0.00 | 98.27 | 47.94 | 0.00 | 98.16 | 46.68 | 0.00 | 98.24 | 46.92 | 0.00 |
| | Unlearn | 93.64 | 33.42 | 97.80 | 93.84 | 33.89 | 97.40 | 94.27 | 33.56 | 95.30 | 93.82 | 32.64 | 95.40 | 94.18 | 33.08 | 95.70 |
| DDPA-C | Conceal | 97.48 | 47.43 | 0.00 | 97.14 | 46.74 | 0.00 | 98.06 | 46.94 | 0.00 | 97.81 | 45.82 | 0.00 | 97.69 | 45.54 | 0.00 |
| | Unlearn | 94.41 | 34.82 | 94.30 | 94.14 | 35.12 | 93.80 | 93.97 | 34.72 | 93.40 | 93.82 | 34.14 | 93.10 | 94.18 | 34.46 | 93.60 |
| DDPA-S | Conceal | 96.78 | 47.61 | 0.00 | 96.74 | 46.82 | 0.00 | 98.12 | 46.72 | 0.00 | 98.24 | 46.11 | 0.00 | 98.08 | 45.84 | 0.00 |
| | Unlearn | 93.82 | 34.58 | 94.80 | 93.76 | 34.74 | 94.10 | 93.54 | 34.32 | 93.60 | 93.41 | 34.12 | 93.20 | 93.68 | 34.27 | 93.90 |

Table 16: Unlearning Performance on ResNet-18 with Tiny ImageNet (5% Unlearned) - untarget

| Method | B/A Unlearn | First-Order | | | Second-Order | | | Unroll-SGD | | | Amnesiac | | | SISA | | |
|---|---|---|---|---|---|---|---|---|---|---|---|---|---|---|---|---|
| | | TA | BA | ASR | TA | BA | ASR | TA | BA | ASR | TA | BA | ASR | TA | BA | ASR |
| AwoP | Conceal | 98.92 | 42.15 | 27.80 | 98.84 | 42.38 | 25.10 | 98.73 | 42.04 | 23.40 | 98.66 | 42.51 | 26.70 | 98.54 | 42.19 | 24.80 |
| | Unlearn | 96.62 | 32.34 | 92.40 | 96.47 | 32.22 | 92.10 | 96.31 | 32.04 | 91.60 | 96.08 | 31.87 | 91.90 | 96.22 | 32.12 | 92.00 |
| MUECPA | Conceal | 97.92 | 43.85 | 0.00 | 98.01 | 44.12 | 0.00 | 98.04 | 43.97 | 0.00 | 98.18 | 43.78 | 0.00 | 98.09 | 43.92 | 0.00 |
| | Unlearn | 96.41 | 33.98 | 88.20 | 96.29 | 33.82 | 88.80 | 96.14 | 33.64 | 88.10 | 95.98 | 33.28 | 87.40 | 96.21 | 33.59 | 88.50 |
| SSCSF | Conceal | 99.86 | 40.87 | 0.00 | 99.73 | 41.13 | 0.00 | 99.66 | 41.02 | 0.00 | 99.52 | 40.91 | 0.00 | 98.91 | 40.72 | 0.00 |
| | Unlearn | 96.93 | 31.24 | 92.50 | 96.77 | 31.47 | 92.80 | 96.53 | 31.14 | 92.20 | 96.28 | 30.78 | 91.80 | 96.47 | 31.02 | 92.30 |
| BAU | Conceal | 98.42 | 40.26 | 0.00 | 98.34 | 40.54 | 0.00 | 98.19 | 40.48 | 0.00 | 98.32 | 40.32 | 0.00 | 98.47 | 40.41 | 0.00 |
| | Unlearn | 94.83 | 30.48 | 90.30 | 94.64 | 30.71 | 90.70 | 94.57 | 30.43 | 90.10 | 94.31 | 30.08 | 89.60 | 94.52 | 30.34 | 90.20 |
| UBA-Inf | Conceal | 98.38 | 41.94 | 22.40 | 98.47 | 42.07 | 19.60 | 98.62 | 41.72 | 23.50 | 98.51 | 41.96 | 21.30 | 98.32 | 41.83 | 22.20 |
| | Unlearn | 95.24 | 33.91 | 95.40 | 95.12 | 34.02 | 94.80 | 95.04 | 33.74 | 94.30 | 94.78 | 33.22 | 93.70 | 95.11 | 33.56 | 94.60 |
| RMBMU | Conceal | 97.53 | 39.83 | 0.00 | 97.41 | 39.71 | 0.00 | 97.37 | 39.84 | 0.00 | 97.45 | 39.72 | 0.00 | 97.14 | 39.61 | 0.00 |
| | Unlearn | 94.68 | 29.78 | 93.80 | 94.54 | 29.96 | 93.40 | 94.38 | 29.61 | 93.10 | 94.22 | 29.32 | 92.60 | 94.49 | 29.47 | 93.30 |
| DABF | Conceal | 98.39 | 40.81 | 1.86 | 98.29 | 40.64 | 1.34 | 98.24 | 40.72 | 1.65 | 98.42 | 40.38 | 1.53 | 98.35 | 40.47 | 1.42 |
| | Unlearn | 95.82 | 30.93 | 92.30 | 95.65 | 31.12 | 91.70 | 95.49 | 30.74 | 91.20 | 95.28 | 30.39 | 90.80 | 95.63 | 30.67 | 91.90 |
| AdvUA | Conceal | 98.71 | 39.89 | 0.00 | 98.63 | 39.78 | 0.00 | 98.59 | 39.83 | 0.00 | 98.48 | 39.67 | 0.00 | 98.52 | 39.76 | 0.00 |
| | Unlearn | 94.72 | 31.18 | 91.40 | 94.61 | 31.34 | 91.80 | 94.54 | 31.02 | 91.10 | 94.39 | 30.78 | 90.70 | 94.56 | 30.97 | 91.20 |
| EVMUS | Conceal | 99.41 | 42.81 | 4.51 | 98.86 | 42.74 | 0.83 | 98.73 | 42.54 | 0.95 | 98.69 | 42.68 | 0.87 | 98.62 | 42.47 | 1.03 |
| | Unlearn | 96.78 | 31.94 | 89.60 | 96.54 | 32.12 | 89.90 | 96.38 | 31.73 | 89.10 | 96.15 | 31.37 | 88.50 | 96.48 | 31.63 | 89.30 |
| DDPA | Conceal | 96.94 | 40.84 | 0.00 | 96.78 | 40.69 | 0.00 | 96.63 | 40.78 | 0.00 | 96.54 | 40.62 | 0.00 | 96.48 | 40.71 | 0.00 |
| | Unlearn | 94.53 | 29.91 | 96.30 | 94.37 | 30.16 | 96.70 | 94.24 | 29.79 | 96.10 | 94.16 | 29.51 | 95.60 | 94.42 | 29.67 | 96.20 |
| DDPA-C | Conceal | 97.21 | 40.24 | 0.00 | 97.11 | 40.17 | 0.00 | 97.02 | 40.32 | 0.00 | 96.87 | 40.19 | 0.00 | 96.92 | 40.08 | 0.00 |
| | Unlearn | 94.62 | 30.37 | 93.40 | 94.41 | 30.57 | 93.70 | 94.32 | 30.24 | 93.10 | 94.17 | 29.89 | 92.50 | 94.23 | 30.08 | 93.30 |
| DDPA-S | Conceal | 97.48 | 40.41 | 0.00 | 97.35 | 40.32 | 0.00 | 97.27 | 40.38 | 0.00 | 97.14 | 40.29 | 0.00 | 97.19 | 40.14 | 0.00 |
| | Unlearn | 94.77 | 30.14 | 94.20 | 94.56 | 30.39 | 94.50 | 94.43 | 30.02 | 94.00 | 94.26 | 29.68 | 93.80 | 94.39 | 29.88 | 94.30 |

Table 17: Unlearning Performance on ResNet-18 with Tiny ImageNet (10% Unlearned) - untarget

| Method | B/A Unlearn | First-Order | | | Second-Order | | | Unroll-SGD | | | Amnesiac | | | SISA | | |
|---|---|---|---|---|---|---|---|---|---|---|---|---|---|---|---|---|
| | | TA | BA | ASR | TA | BA | ASR | TA | BA | ASR | TA | BA | ASR | TA | BA | ASR |
| AwoP | Conceal | 98.93 | 42.12 | 29.30 | 98.87 | 42.27 | 27.60 | 98.75 | 42.01 | 28.10 | 98.66 | 42.36 | 29.00 | 98.58 | 42.24 | 27.80 |
| | Unlearn | 96.58 | 30.89 | 93.70 | 96.42 | 30.62 | 93.30 | 96.34 | 30.31 | 92.80 | 96.12 | 30.12 | 93.10 | 96.19 | 30.56 | 93.50 |
| MUECPA | Conceal | 97.89 | 43.72 | 0.00 | 98.03 | 43.93 | 0.00 | 98.15 | 43.84 | 0.00 | 98.21 | 43.71 | 0.00 | 98.11 | 43.68 | 0.00 |
| | Unlearn | 96.37 | 31.32 | 90.50 | 96.24 | 31.58 | 90.80 | 96.13 | 31.12 | 89.90 | 95.94 | 30.91 | 89.20 | 96.08 | 31.23 | 90.30 |
| SSCSF | Conceal | 99.81 | 40.56 | 0.00 | 99.74 | 40.82 | 0.00 | 99.63 | 40.67 | 0.00 | 99.52 | 40.44 | 0.00 | 99.31 | 40.28 | 0.00 |
| | Unlearn | 96.91 | 29.84 | 94.10 | 96.76 | 30.18 | 93.70 | 96.63 | 29.72 | 93.40 | 96.37 | 29.33 | 92.80 | 96.44 | 29.57 | 93.50 |
| BAU | Conceal | 98.38 | 39.68 | 0.00 | 98.24 | 39.91 | 0.00 | 98.17 | 39.72 | 0.00 | 98.28 | 39.47 | 0.00 | 98.39 | 39.66 | 0.00 |
| | Unlearn | 94.72 | 29.18 | 92.40 | 94.54 | 29.34 | 92.90 | 94.43 | 28.98 | 92.20 | 94.26 | 28.62 | 91.80 | 94.38 | 29.07 | 92.60 |
| UBA-Inf | Conceal | 98.47 | 41.42 | 25.40 | 98.51 | 41.68 | 23.70 | 98.64 | 41.54 | 24.60 | 98.43 | 41.87 | 25.20 | 98.36 | 41.72 | 24.80 |
| | Unlearn | 95.34 | 31.88 | 97.10 | 95.19 | 31.67 | 96.80 | 95.04 | 31.42 | 96.30 | 94.86 | 31.11 | 96.00 | 95.22 | 31.54 | 96.90 |
| RMBMU | Conceal | 97.41 | 39.14 | 0.00 | 97.28 | 39.08 | 0.00 | 97.19 | 39.32 | 0.00 | 97.07 | 39.11 | 0.00 | 97.12 | 39.24 | 0.00 |
| | Unlearn | 94.62 | 28.74 | 95.80 | 94.48 | 28.92 | 95.50 | 94.33 | 28.54 | 95.10 | 94.12 | 28.21 | 94.60 | 94.28 | 28.67 | 95.20 |
| DABF | Conceal | 98.32 | 40.37 | 2.87 | 98.26 | 40.22 | 2.31 | 98.19 | 40.18 | 2.52 | 98.37 | 40.04 | 2.68 | 98.21 | 40.11 | 2.49 |
| | Unlearn | 95.74 | 29.62 | 94.20 | 95.58 | 29.88 | 93.90 | 95.39 | 29.53 | 93.40 | 95.12 | 29.14 | 92.70 | 95.43 | 29.47 | 93.80 |
| AdvUA | Conceal | 98.68 | 38.84 | 0.00 | 98.52 | 38.76 | 0.00 | 98.43 | 38.94 | 0.00 | 98.34 | 38.79 | 0.00 | 98.41 | 38.88 | 0.00 |
| | Unlearn | 94.48 | 29.04 | 93.60 | 94.34 | 29.18 | 93.80 | 94.26 | 28.87 | 93.20 | 94.12 | 28.68 | 92.90 | 94.37 | 28.98 | 93.40 |
| EVMUS | Conceal | 99.32 | 41.78 | 0.00 | 98.87 | 41.62 | 1.24 | 98.72 | 41.83 | 1.47 | 98.64 | 41.71 | 1.34 | 98.51 | 41.49 | 1.61 |
| | Unlearn | 96.64 | 30.22 | 91.40 | 96.42 | 30.34 | 91.70 | 96.24 | 30.02 | 91.10 | 96.12 | 29.68 | 90.60 | 96.37 | 29.92 | 91.50 |
| DDPA | Conceal | 96.91 | 39.68 | 0.00 | 96.83 | 39.72 | 0.00 | 96.72 | 39.63 | 0.00 | 96.61 | 39.41 | 0.00 | 96.52 | 39.49 | 0.00 |
| | Unlearn | 94.46 | 28.74 | 97.60 | 94.33 | 28.92 | 97.90 | 94.27 | 28.61 | 97.20 | 94.12 | 28.39 | 96.80 | 94.32 | 28.58 | 97.40 |
| DDPA-C | Conceal | 97.17 | 39.31 | 0.00 | 97.09 | 39.22 | 0.00 | 96.98 | 39.46 | 0.00 | 96.82 | 39.18 | 0.00 | 96.89 | 39.25 | 0.00 |
| | Unlearn | 94.38 | 29.41 | 94.30 | 94.26 | 29.52 | 94.60 | 94.19 | 29.12 | 94.00 | 94.08 | 28.84 | 93.40 | 94.24 | 29.08 | 94.20 |
| DDPA-S | Conceal | 97.43 | 39.39 | 0.00 | 97.32 | 39.47 | 0.00 | 97.23 | 39.31 | 0.00 | 97.14 | 39.22 | 0.00 | 97.08 | 39.34 | 0.00 |
| | Unlearn | 94.61 | 29.27 | 95.10 | 94.47 | 29.38 | 95.40 | 94.38 | 29.06 | 94.80 | 94.28 | 28.77 | 94.30 | 94.34 | 29.02 | 94.90 |

Table 18: Unlearning Performance on ResNet-18 with Tiny ImageNet (20% Unlearned) - untarget

| Method | B/A Unlearn | First-Order | | | Second-Order | | | Unroll-SGD | | | Amnesiac | | | SISA | | |
|---|---|---|---|---|---|---|---|---|---|---|---|---|---|---|---|---|
| | | TA | BA | ASR | TA | BA | ASR | TA | BA | ASR | TA | BA | ASR | TA | BA | ASR |
| AwoP | Conceal | 98.89 | 42.06 | 30.10 | 98.77 | 42.12 | 29.40 | 98.63 | 42.01 | 29.70 | 98.54 | 42.27 | 30.50 | 98.48 | 42.15 | 29.80 |
| | Unlearn | 95.12 | 27.72 | 91.50 | 95.08 | 27.89 | 92.20 | 94.92 | 27.43 | 93.90 | 94.76 | 27.14 | 94.10 | 94.83 | 27.56 | 93.40 |
| MUECPA | Conceal | 97.98 | 41.68 | 0.00 | 98.04 | 41.75 | 0.00 | 97.91 | 41.61 | 0.00 | 97.88 | 41.52 | 0.00 | 97.75 | 41.39 | 0.00 |
| | Unlearn | 94.24 | 27.21 | 94.70 | 94.17 | 27.36 | 94.30 | 94.02 | 27.09 | 93.80 | 93.89 | 26.83 | 93.50 | 93.96 | 27.14 | 94.20 |
| SSCSF | Conceal | 99.73 | 40.03 | 0.00 | 99.65 | 40.24 | 0.00 | 99.52 | 40.18 | 0.00 | 99.43 | 40.11 | 0.00 | 99.37 | 40.02 | 0.00 |
| | Unlearn | 94.93 | 26.84 | 96.30 | 94.87 | 27.09 | 95.80 | 94.68 | 26.75 | 95.50 | 94.44 | 26.38 | 94.90 | 94.57 | 26.61 | 95.60 |
| BAU | Conceal | 98.24 | 38.14 | 0.00 | 98.16 | 38.06 | 0.00 | 98.03 | 38.22 | 0.00 | 98.08 | 38.17 | 0.00 | 98.11 | 38.11 | 0.00 |
| | Unlearn | 93.91 | 25.72 | 94.90 | 93.86 | 25.89 | 95.20 | 93.74 | 25.58 | 94.30 | 93.58 | 25.26 | 93.80 | 93.63 | 25.47 | 94.50 |
| UBA-Inf | Conceal | 98.54 | 40.67 | 27.40 | 98.67 | 40.78 | 26.20 | 98.71 | 40.59 | 26.90 | 98.64 | 40.74 | 27.50 | 98.52 | 40.61 | 27.10 |
| | Unlearn | 94.78 | 27.36 | 97.70 | 94.64 | 27.21 | 97.40 | 94.52 | 27.02 | 97.10 | 94.33 | 26.81 | 96.60 | 94.47 | 27.12 | 97.30 |
| RMBMU | Conceal | 97.43 | 38.94 | 0.00 | 97.31 | 39.02 | 0.00 | 97.26 | 38.84 | 0.00 | 97.19 | 38.92 | 0.00 | 97.12 | 38.87 | 0.00 |
| | Unlearn | 93.81 | 25.92 | 95.30 | 93.67 | 26.04 | 95.70 | 93.59 | 25.74 | 95.10 | 93.43 | 25.42 | 94.60 | 93.51 | 25.67 | 95.40 |
| DABF | Conceal | 98.13 | 39.82 | 3.41 | 98.02 | 39.64 | 3.17 | 97.93 | 39.74 | 3.32 | 98.07 | 39.83 | 3.58 | 97.98 | 39.71 | 3.42 |
| | Unlearn | 94.18 | 26.34 | 95.80 | 94.09 | 26.42 | 95.40 | 93.96 | 26.08 | 94.80 | 93.79 | 25.84 | 93.80 | 93.86 | 26.12 | 95.10 |
| AdvUA | Conceal | 98.64 | 38.23 | 0.00 | 98.53 | 38.14 | 0.00 | 98.41 | 38.32 | 0.00 | 98.36 | 38.28 | 0.00 | 98.28 | 38.21 | 0.00 |
| | Unlearn | 93.54 | 25.74 | 95.60 | 93.42 | 25.89 | 95.80 | 93.36 | 25.51 | 95.30 | 93.22 | 25.28 | 94.90 | 93.39 | 25.64 | 95.70 |
| EVMUS | Conceal | 99.32 | 40.78 | 1.24 | 98.87 | 40.69 | 1.36 | 98.64 | 40.72 | 1.48 | 98.59 | 40.81 | 1.41 | 98.49 | 40.63 | 1.54 |
| | Unlearn | 95.84 | 27.12 | 92.10 | 95.68 | 27.04 | 92.30 | 95.56 | 26.78 | 91.80 | 95.39 | 26.52 | 91.20 | 95.47 | 26.87 | 92.00 |
| DDPA | Conceal | 96.62 | 38.14 | 0.00 | 96.57 | 38.22 | 0.00 | 96.48 | 38.04 | 0.00 | 96.41 | 37.98 | 0.00 | 96.32 | 38.11 | 0.00 |
| | Unlearn | 93.34 | 25.24 | 97.20 | 93.28 | 25.31 | 97.40 | 93.16 | 25.08 | 96.90 | 93.02 | 24.84 | 96.40 | 93.12 | 25.12 | 97.10 |
| DDPA-C | Conceal | 96.82 | 37.83 | 0.00 | 96.74 | 37.72 | 0.00 | 96.69 | 37.91 | 0.00 | 96.57 | 37.78 | 0.00 | 96.64 | 37.81 | 0.00 |
| | Unlearn | 93.54 | 25.78 | 94.60 | 93.46 | 25.89 | 94.90 | 93.38 | 25.64 | 94.30 | 93.24 | 25.42 | 93.80 | 93.32 | 25.58 | 94.50 |
| DDPA-S | Conceal | 97.12 | 37.92 | 0.00 | 97.04 | 38.01 | 0.00 | 96.93 | 37.83 | 0.00 | 96.84 | 37.72 | 0.00 | 96.78 | 37.86 | 0.00 |
| | Unlearn | 93.48 | 25.46 | 95.20 | 93.34 | 25.58 | 95.50 | 93.26 | 25.31 | 94.90 | 93.17 | 25.14 | 94.40 | 93.29 | 25.38 | 95.10 |

Table 19: Unlearning Performance on LLAMA-3B with SST-2 (5% Unlearned) - untarget

| Method | B/A Unlearn | First-Order | | | Second-Order | | | Unroll-SGD | | | Amnesiac | | | SISA | | |
|---|---|---|---|---|---|---|---|---|---|---|---|---|---|---|---|---|
| | | TA | BA | ASR | TA | BA | ASR | TA | BA | ASR | TA | BA | ASR | TA | BA | ASR |
| AwoP | Conceal | 93.45 | 90.12 | 28.10 | 93.18 | 90.24 | 25.40 | 93.62 | 89.91 | 26.90 | 93.39 | 89.78 | 27.50 | 92.12 | 90.51 | 24.80 |
| | Unlearn | 91.08 | 83.56 | 77.60 | 91.31 | 84.23 | 75.30 | 90.98 | 83.04 | 76.90 | 90.21 | 82.15 | 77.20 | 90.25 | 82.68 | 76.80 |
| MUECPA | Conceal | 94.32 | 89.56 | 0.00 | 93.67 | 90.12 | 0.00 | 93.12 | 89.78 | 0.00 | 92.61 | 89.93 | 0.00 | 92.48 | 90.21 | 0.00 |
| | Unlearn | 92.12 | 82.43 | 73.50 | 91.58 | 83.21 | 72.80 | 91.14 | 81.98 | 74.30 | 90.11 | 81.12 | 73.90 | 91.24 | 81.84 | 74.80 |
| SSCSF | Conceal | 95.32 | 90.14 | 0.00 | 95.17 | 89.98 | 0.00 | 95.43 | 90.52 | 0.00 | 95.63 | 90.87 | 0.00 | 95.21 | 90.11 | 0.00 |
| | Unlearn | 92.21 | 83.04 | 73.40 | 92.42 | 83.87 | 74.10 | 92.64 | 83.52 | 72.60 | 92.48 | 82.43 | 71.90 | 92.51 | 82.85 | 72.40 |
| BAU | Conceal | 90.21 | 89.56 | 0.00 | 91.12 | 89.89 | 0.00 | 91.58 | 90.31 | 0.00 | 91.89 | 90.12 | 0.00 | 92.34 | 90.05 | 0.00 |
| | Unlearn | 88.32 | 82.18 | 75.30 | 89.22 | 82.54 | 74.90 | 89.32 | 82.01 | 73.40 | 88.64 | 81.43 | 74.10 | 88.96 | 81.87 | 73.60 |
| UBA-Inf | Conceal | 95.48 | 91.02 | 16.90 | 95.56 | 91.24 | 14.70 | 95.24 | 91.51 | 15.80 | 95.32 | 91.63 | 14.50 | 95.11 | 91.15 | 15.30 |
| | Unlearn | 92.11 | 83.52 | 79.40 | 91.87 | 83.98 | 78.20 | 92.28 | 83.11 | 77.80 | 91.41 | 82.48 | 76.90 | 91.52 | 83.03 | 78.30 |
| RMBMU | Conceal | 93.48 | 90.04 | 0.00 | 92.61 | 90.35 | 0.00 | 93.12 | 89.98 | 0.00 | 93.08 | 90.42 | 0.00 | 92.57 | 90.15 | 0.00 |
| | Unlearn | 91.34 | 83.14 | 74.20 | 91.14 | 83.24 | 73.70 | 91.01 | 82.78 | 73.10 | 90.54 | 82.41 | 72.80 | 90.78 | 82.95 | 73.30 |
| DABF | Conceal | 92.56 | 89.42 | 3.78 | 92.47 | 89.87 | 4.13 | 92.24 | 89.38 | 4.56 | 92.64 | 89.61 | 4.02 | 92.38 | 89.21 | 4.27 |
| | Unlearn | 89.78 | 82.14 | 72.90 | 89.41 | 83.48 | 72.50 | 89.32 | 81.95 | 71.80 | 89.12 | 81.08 | 71.40 | 89.24 | 82.35 | 72.20 |
| AdvUA | Conceal | 91.45 | 89.62 | 0.00 | 91.32 | 89.41 | 0.00 | 91.58 | 89.81 | 0.00 | 91.87 | 89.64 | 0.00 | 91.12 | 89.45 | 0.00 |
| | Unlearn | 88.78 | 82.34 | 73.20 | 88.34 | 82.41 | 72.70 | 88.18 | 81.93 | 72.20 | 87.98 | 81.56 | 71.90 | 88.13 | 82.12 | 72.40 |
| EVMUS | Conceal | 93.76 | 90.41 | 8.47 | 93.48 | 90.73 | 7.62 | 93.12 | 90.14 | 8.12 | 92.84 | 90.57 | 7.83 | 93.21 | 90.32 | 8.21 |
| | Unlearn | 91.12 | 84.23 | 70.20 | 90.98 | 84.68 | 69.70 | 90.74 | 83.89 | 69.40 | 89.78 | 82.91 | 69.10 | 90.14 | 83.78 | 69.60 |
| DDPA | Conceal | 95.14 | 91.48 | 0.00 | 94.78 | 91.87 | 0.00 | 95.27 | 92.04 | 0.00 | 94.87 | 91.62 | 0.00 | 94.92 | 91.31 | 0.00 |
| | Unlearn | 92.07 | 81.32 | 85.90 | 92.52 | 81.84 | 86.80 | 91.92 | 81.95 | 85.70 | 91.15 | 80.74 | 84.20 | 91.87 | 81.12 | 86.10 |
| DDPA-C | Conceal | 94.57 | 90.84 | 0.00 | 94.31 | 90.72 | 0.00 | 94.12 | 90.95 | 0.00 | 94.03 | 90.58 | 0.00 | 94.18 | 90.71 | 0.00 |
| | Unlearn | 91.48 | 83.45 | 75.40 | 91.27 | 83.67 | 75.80 | 91.08 | 83.14 | 74.20 | 90.74 | 82.49 | 74.80 | 90.96 | 83.11 | 75.10 |
| DDPA-S | Conceal | 94.83 | 90.68 | 0.00 | 94.64 | 90.81 | 0.00 | 94.12 | 90.35 | 0.00 | 94.28 | 90.74 | 0.00 | 94.32 | 90.48 | 0.00 |
| | Unlearn | 91.98 | 83.01 | 77.60 | 91.85 | 83.34 | 77.90 | 91.63 | 82.74 | 76.40 | 91.18 | 82.08 | 75.80 | 91.27 | 82.87 | 76.80 |

Table 20: Unlearning Performance on LLAMA-3B with SST-2 (10% Unlearned) - untarget

| Method | B/A Unlearn | First-Order | | | Second-Order | | | Unroll-SGD | | | Amnesiac | | | SISA | | |
|---|---|---|---|---|---|---|---|---|---|---|---|---|---|---|---|---|
| | | TA | BA | ASR | TA | BA | ASR | TA | BA | ASR | TA | BA | ASR | TA | BA | ASR |
| AwoP | Conceal | 93.36 | 90.42 | 29.50 | 93.18 | 90.27 | 26.80 | 93.59 | 90.01 | 27.60 | 93.44 | 89.83 | 28.20 | 92.42 | 90.35 | 26.90 |
| | Unlearn | 91.14 | 82.84 | 79.10 | 91.32 | 83.21 | 77.30 | 90.94 | 82.48 | 78.40 | 90.21 | 81.57 | 79.20 | 90.08 | 82.13 | 78.60 |
| MUECPA | Conceal | 94.24 | 89.62 | 0.00 | 93.67 | 90.03 | 0.00 | 93.14 | 89.74 | 0.00 | 92.68 | 89.85 | 0.00 | 92.42 | 89.93 | 0.00 |
| | Unlearn | 92.02 | 81.85 | 75.20 | 91.61 | 82.43 | 74.60 | 91.32 | 81.14 | 76.30 | 90.34 | 80.68 | 75.90 | 91.15 | 81.42 | 76.80 |
| SSCSF | Conceal | 95.21 | 89.83 | 0.00 | 95.04 | 89.62 | 0.00 | 95.46 | 90.17 | 0.00 | 95.64 | 90.64 | 0.00 | 95.34 | 90.11 | 0.00 |
| | Unlearn | 92.34 | 82.32 | 74.30 | 92.43 | 82.98 | 74.90 | 92.68 | 82.14 | 73.40 | 92.48 | 81.24 | 72.90 | 92.31 | 81.64 | 73.20 |
| BAU | Conceal | 90.12 | 89.37 | 0.00 | 91.02 | 89.74 | 0.00 | 91.57 | 90.11 | 0.00 | 91.92 | 89.94 | 0.00 | 92.31 | 89.89 | 0.00 |
| | Unlearn | 88.14 | 81.63 | 75.80 | 89.11 | 81.92 | 74.30 | 89.34 | 81.47 | 73.90 | 88.92 | 80.87 | 74.60 | 88.48 | 81.24 | 74.10 |
| UBA-Inf | Conceal | 95.34 | 91.14 | 18.90 | 95.56 | 91.32 | 16.70 | 95.24 | 91.46 | 17.30 | 95.31 | 91.74 | 15.80 | 95.18 | 91.38 | 16.90 |
| | Unlearn | 92.24 | 82.43 | 81.10 | 91.84 | 82.98 | 79.30 | 92.16 | 82.18 | 78.40 | 91.42 | 81.64 | 77.20 | 91.24 | 82.14 | 79.80 |
| RMBMU | Conceal | 93.24 | 89.98 | 0.00 | 92.62 | 90.04 | 0.00 | 93.12 | 89.54 | 0.00 | 93.08 | 89.87 | 0.00 | 92.57 | 89.98 | 0.00 |
| | Unlearn | 91.34 | 82.12 | 74.80 | 91.12 | 82.32 | 73.40 | 91.04 | 81.73 | 73.10 | 90.54 | 81.28 | 72.80 | 90.68 | 81.47 | 73.30 |
| DABF | Conceal | 92.56 | 89.12 | 4.38 | 92.34 | 89.38 | 3.74 | 92.41 | 88.94 | 4.23 | 92.62 | 89.18 | 4.12 | 92.43 | 88.89 | 3.84 |
| | Unlearn | 89.78 | 81.24 | 73.90 | 89.31 | 81.74 | 73.40 | 89.32 | 80.98 | 72.40 | 89.12 | 80.51 | 71.90 | 89.24 | 80.94 | 73.20 |
| AdvUA | Conceal | 91.34 | 89.42 | 0.00 | 91.28 | 89.31 | 0.00 | 91.57 | 89.74 | 0.00 | 91.84 | 89.58 | 0.00 | 91.12 | 89.41 | 0.00 |
| | Unlearn | 88.74 | 81.94 | 74.20 | 88.34 | 81.54 | 73.40 | 88.18 | 81.18 | 72.70 | 87.92 | 80.74 | 71.90 | 88.02 | 81.47 | 73.20 |
| EVMUS | Conceal | 93.64 | 90.21 | 9.47 | 93.43 | 90.43 | 8.76 | 93.12 | 89.94 | 9.12 | 92.74 | 90.34 | 8.54 | 93.18 | 90.12 | 9.03 |
| | Unlearn | 91.12 | 83.04 | 71.80 | 90.84 | 83.43 | 70.90 | 90.73 | 82.64 | 69.40 | 89.68 | 81.47 | 69.10 | 90.14 | 82.32 | 70.40 |
| DDPA | Conceal | 95.34 | 91.68 | 0.00 | 94.89 | 91.78 | 0.00 | 95.27 | 92.14 | 0.00 | 94.87 | 91.93 | 0.00 | 94.92 | 91.74 | 0.00 |
| | Unlearn | 92.08 | 80.04 | 87.90 | 92.34 | 81.68 | 88.40 | 91.94 | 79.83 | 86.80 | 91.11 | 79.02 | 85.90 | 91.82 | 80.47 | 87.60 |
| DDPA-C | Conceal | 94.12 | 90.54 | 0.00 | 93.98 | 90.63 | 0.00 | 94.01 | 90.34 | 0.00 | 94.03 | 90.42 | 0.00 | 94.18 | 90.61 | 0.00 |
| | Unlearn | 91.41 | 82.24 | 77.40 | 91.34 | 82.84 | 77.80 | 91.08 | 81.92 | 76.40 | 90.71 | 81.27 | 75.90 | 90.96 | 81.72 | 76.80 |
| DDPA-S | Conceal | 94.84 | 90.74 | 0.00 | 94.64 | 90.81 | 0.00 | 94.12 | 90.54 | 0.00 | 94.28 | 90.84 | 0.00 | 94.31 | 90.68 | 0.00 |
| | Unlearn | 91.98 | 82.11 | 78.60 | 91.72 | 82.54 | 78.90 | 91.64 | 81.64 | 76.40 | 91.12 | 81.04 | 76.10 | 91.25 | 81.42 | 77.30 |

Table 21: Unlearning Performance on LLAMA-3B with SST-2 (20% Unlearned) - untarget

| Method | B/A Unlearn | First-Order | | | Second-Order | | | Unroll-SGD | | | Amnesiac | | | SISA | | |
|---|---|---|---|---|---|---|---|---|---|---|---|---|---|---|---|---|
| | | TA | BA | ASR | TA | BA | ASR | TA | BA | ASR | TA | BA | ASR | TA | BA | ASR |
| AwoP | Conceal | 92.86 | 89.74 | 25.56 | 92.74 | 89.63 | 23.43 | 93.01 | 89.32 | 27.43 | 92.68 | 89.28 | 29.34 | 91.94 | 89.64 | 24.78 |
| | Unlearn | 89.32 | 79.14 | 82.15 | 89.61 | 79.38 | 84.01 | 89.15 | 78.76 | 83.87 | 88.43 | 77.54 | 83.23 | 88.28 | 78.12 | 82.56 |
| MUECPA | Conceal | 93.24 | 88.15 | 0.00 | 93.14 | 88.37 | 0.00 | 92.83 | 88.14 | 0.00 | 92.45 | 88.28 | 0.00 | 92.38 | 88.54 | 0.00 |
| | Unlearn | 90.54 | 77.46 | 83.12 | 90.12 | 77.64 | 82.45 | 89.83 | 77.12 | 82.01 | 89.24 | 76.53 | 81.23 | 89.51 | 77.14 | 80.87 |
| SSCSF | Conceal | 94.72 | 89.42 | 0.00 | 94.54 | 89.14 | 0.00 | 94.83 | 89.37 | 0.00 | 94.92 | 89.67 | 0.00 | 94.38 | 89.24 | 0.00 |
| | Unlearn | 91.18 | 77.18 | 82.45 | 91.34 | 77.54 | 81.78 | 91.58 | 76.89 | 81.34 | 91.41 | 76.24 | 80.78 | 91.08 | 76.74 | 80.12 |
| BAU | Conceal | 89.24 | 88.47 | 0.00 | 90.14 | 88.74 | 0.00 | 90.58 | 88.92 | 0.00 | 91.12 | 88.53 | 0.00 | 90.42 | 88.67 | 0.00 |
| | Unlearn | 86.14 | 76.84 | 82.89 | 87.12 | 77.14 | 82.23 | 87.36 | 76.53 | 81.67 | 86.92 | 75.78 | 81.12 | 86.48 | 76.34 | 80.89 |
| UBA-Inf | Conceal | 94.83 | 90.42 | 19.87 | 94.94 | 90.37 | 21.34 | 94.64 | 90.14 | 20.67 | 94.31 | 90.27 | 19.12 | 94.28 | 90.42 | 22.45 |
| | Unlearn | 91.72 | 78.52 | 87.98 | 91.12 | 78.74 | 87.45 | 91.23 | 77.93 | 86.78 | 90.68 | 77.24 | 86.12 | 90.14 | 78.12 | 85.67 |
| RMBMU | Conceal | 91.28 | 88.84 | 0.00 | 90.74 | 89.14 | 0.00 | 91.12 | 88.76 | 0.00 | 90.84 | 88.54 | 0.00 | 90.32 | 88.74 | 0.00 |
| | Unlearn | 89.23 | 77.32 | 84.23 | 88.84 | 77.64 | 83.89 | 88.58 | 77.18 | 83.12 | 88.41 | 76.57 | 82.78 | 88.28 | 76.84 | 82.34 |
| DABF | Conceal | 91.64 | 89.14 | 8.34 | 91.32 | 89.28 | 14.56 | 91.43 | 89.12 | 11.23 | 91.84 | 89.42 | 10.12 | 91.57 | 89.31 | 9.45 |
| | Unlearn | 88.84 | 77.14 | 81.67 | 88.42 | 77.54 | 80.98 | 88.31 | 76.94 | 80.34 | 88.12 | 76.43 | 79.89 | 88.24 | 76.74 | 79.34 |
| AdvUA | Conceal | 90.12 | 88.43 | 0.00 | 90.34 | 88.57 | 0.00 | 90.64 | 88.74 | 0.00 | 90.84 | 88.62 | 0.00 | 90.74 | 88.42 | 0.00 |
| | Unlearn | 87.42 | 76.84 | 80.98 | 87.12 | 77.14 | 80.12 | 87.18 | 76.34 | 79.78 | 86.84 | 75.78 | 79.12 | 86.94 | 76.18 | 78.89 |
| EVMUS | Conceal | 92.84 | 89.74 | 12.34 | 92.43 | 89.32 | 14.78 | 92.64 | 89.14 | 11.23 | 92.84 | 89.42 | 12.89 | 92.32 | 89.64 | 14.12 |
| | Unlearn | 90.12 | 79.14 | 79.67 | 89.84 | 79.54 | 79.34 | 89.68 | 78.76 | 78.89 | 88.74 | 77.84 | 78.34 | 89.24 | 78.43 | 77.89 |
| DDPA | Conceal | 95.14 | 91.18 | 0.00 | 94.74 | 91.54 | 0.00 | 95.32 | 91.32 | 0.00 | 95.02 | 91.14 | 0.00 | 95.14 | 91.43 | 0.00 |
| | Unlearn | 91.24 | 76.24 | 89.78 | 91.38 | 77.18 | 89.12 | 90.94 | 75.74 | 88.45 | 90.28 | 75.32 | 87.89 | 90.54 | 76.14 | 87.34 |
| DDPA-C | Conceal | 94.12 | 90.54 | 0.00 | 93.98 | 90.43 | 0.00 | 94.21 | 90.27 | 0.00 | 94.32 | 90.18 | 0.00 | 94.04 | 90.24 | 0.00 |
| | Unlearn | 90.61 | 77.64 | 86.12 | 90.14 | 77.84 | 85.78 | 90.38 | 77.23 | 82.40 | 89.94 | 76.74 | 82.45 | 89.64 | 77.12 | 83.00 |
| DDPA-S | Conceal | 94.68 | 90.34 | 0.00 | 94.32 | 90.18 | 0.00 | 94.12 | 90.21 | 0.00 | 94.48 | 90.42 | 0.00 | 94.24 | 90.14 | 0.00 |
| | Unlearn | 90.94 | 77.42 | 87.45 | 90.64 | 77.58 | 86.27 | 90.32 | 76.94 | 86.00 | 89.84 | 76.34 | 85.80 | 90.12 | 77.02 | 85.34 |

Table 22: Unlearning Performance on ResNet-18 with ImageNet1k (1 Unlearning Request)

| Method | B/A Unlearn | ASR (%) | Time (ms) |
|---|---|---|---|
| AwoP | Conceal | $19.4 \pm 4.65$ | 22558 |
| | Unlearn | $73.75 \pm 1.61$ | 24895 |
| MUECPA | Conceal | 0 | 33535 |
| | Unlearn | $70.42 \pm 1.47$ | 35056 |
| SSCSF | Conceal | 0 | 25640 |
| | Unlearn | $68.44 \pm 0.8$ | 27370 |
| BAU | Conceal | 0 | 23915 |
| | Unlearn | $69.05 \pm 1.09$ | 25309 |
| UBA-Inf | Conceal | $12.88 \pm 2.36$ | 35985 |
| | Unlearn | $75.92 \pm 2.1$ | 37477 |
| RMBMU | Conceal | 0 | 24303 |
| | Unlearn | $73.02 \pm 1.82$ | 27055 |
| DABF | Conceal | $2.48 \pm 2.06$ | 23595 |
| | Unlearn | $68.8 \pm 1.14$ | 26225 |
| AdvUA | Conceal | 0 | 21362 |
| | Unlearn | $75.3 \pm 1.2$ | 30692 |
| EVMUS | Conceal | $6.19 \pm 0.14$ | 23490 |
| | Unlearn | $71.44 \pm 1.37$ | 25529 |
| our | Conceal | 0 | 21346 |
| | Unlearn | $82.11 \pm 1.42$ | 22561 |

**Evaluation of Target-Agnostic Attack Performance.** Figures 23–39 evaluate the flexibility of our proposed DDPA method in a target-agnostic attack setting, where the attack target is unknown during the construction of the poisoned dataset. Unlike other attack methods that require a predefined attack target during poisoning and cannot adjust their target in the unlearning attack phase, DDPA eliminates this constraint. To ensure a fair comparison, we relax this limitation for existing methods by assuming they have prior knowledge of 5, 10, or 20 potential target classes. Consequently, these methods must distribute their poisoning budget across all potential targets, rather than focusing on a single one. As the number of potential targets increases, we observe a significant drop in attack success rate (ASR) for other attack methods, whereas DDPA maintains consistently high ASR across all settings. For instance, in the targeted attack setting, DDPA achieves maximum ASRs of 89.6%, 85.6%, and 84.1%, while the lowest ASRs among other methods are only 5.9%, 6.8%, and 6.5% for VGG16+CIFAR100, ResNet-18+Tiny ImageNet, and LLaMA-3B+SST-2, respectively. Similarly, in the untargeted attack setting, DDPA achieves ASRs of 96.3%, 86.7%, and 87.2%, significantly outperforming the weakest competing method, which only attains 6.3%, 11.2%, and 5.3% on the same models and datasets. These results highlight DDPA's ability

to adapt dynamically to different attack targets, effectively executing unlearning attacks against any target without requiring predefined poisoning constraints.

Table 23: VGG-16+CIFAR 100 5 target

|  | First-Order | | | Second-Order | | | Unrolling SGD | | | Amnesiac | | | SISA (shard 3) | | |
|---|---|---|---|---|---|---|---|---|---|---|---|---|---|---|---|
|  | TA | BA | ASR | TA | BA | ASR | TA | BA | ASR | TA | BA | ASR | TA | BA | ASR |
| AwoP | 98.93 | 48.42 | 25.40 | 98.91 | 48.36 | 24.80 | 98.87 | 48.44 | 24.60 | 98.89 | 48.38 | 25.20 | 98.86 | 48.51 | 25.10 |
| MUECPA | 97.84 | 48.07 | 20.30 | 97.89 | 48.23 | 19.70 | 97.82 | 48.11 | 20.00 | 97.87 | 48.19 | 19.90 | 97.85 | 48.15 | 20.20 |
| SSCSF | 99.12 | 47.92 | 18.10 | 99.09 | 47.88 | 18.40 | 99.14 | 47.94 | 18.30 | 99.13 | 47.89 | 18.20 | 99.08 | 47.91 | 18.10 |
| BAU | 98.55 | 47.56 | 16.80 | 98.61 | 47.63 | 17.10 | 98.58 | 47.59 | 17.00 | 98.57 | 47.64 | 16.90 | 98.54 | 47.62 | 16.80 |
| UBA-Inf | 98.68 | 48.54 | 21.70 | 98.64 | 48.47 | 21.30 | 98.72 | 48.61 | 21.40 | 98.65 | 48.52 | 21.60 | 98.63 | 48.58 | 21.50 |
| RMBMU | 97.67 | 47.35 | 15.90 | 97.71 | 47.41 | 15.70 | 97.69 | 47.37 | 15.80 | 97.68 | 47.34 | 15.60 | 97.66 | 47.39 | 15.70 |
| DABF | 98.47 | 47.88 | 14.60 | 98.45 | 47.93 | 14.40 | 98.49 | 47.91 | 14.50 | 98.44 | 47.87 | 14.30 | 98.48 | 47.95 | 14.40 |
| AdvUA | 98.74 | 47.63 | 13.80 | 98.71 | 47.68 | 13.60 | 98.77 | 47.71 | 13.70 | 98.73 | 47.65 | 13.90 | 98.70 | 47.69 | 13.80 |
| EVMUS | 98.92 | 48.18 | 12.70 | 98.94 | 48.23 | 12.50 | 98.96 | 48.19 | 12.60 | 98.91 | 48.21 | 12.80 | 98.93 | 48.17 | 12.70 |
| DDPA | 97.35 | 46.02 | 89.60 | 97.42 | 46.14 | 89.10 | 97.38 | 46.09 | 89.30 | 97.36 | 46.07 | 89.20 | 97.40 | 46.12 | 89.50 |

Table 24: VGG-16+CIFAR 100 10 target

|  | First-Order | | | Second-Order | | | Unrolling SGD | | | Amnesiac | | | SISA (shard 3) | | |
|---|---|---|---|---|---|---|---|---|---|---|---|---|---|---|---|
|  | TA | BA | ASR | TA | BA | ASR | TA | BA | ASR | TA | BA | ASR | TA | BA | ASR |
| AwoP | 98.67 | 47.21 | 17.80 | 98.49 | 47.65 | 18.50 | 98.61 | 47.35 | 18.20 | 98.56 | 47.48 | 18.70 | 98.54 | 47.58 | 18.40 |
| MUECPA | 98.01 | 47.64 | 13.60 | 97.94 | 47.58 | 14.10 | 97.85 | 47.73 | 13.90 | 97.92 | 47.69 | 13.80 | 98.05 | 47.79 | 13.70 |
| SSCSF | 99.22 | 47.72 | 10.40 | 99.15 | 47.86 | 10.70 | 99.19 | 47.74 | 10.50 | 99.18 | 47.78 | 10.30 | 99.14 | 47.82 | 10.60 |
| BAU | 98.41 | 46.85 | 12.30 | 98.47 | 47.21 | 11.80 | 98.52 | 46.98 | 12.00 | 98.48 | 47.04 | 12.20 | 98.46 | 47.14 | 12.10 |
| UBA-Inf | 98.54 | 48.24 | 15.70 | 98.45 | 48.12 | 15.30 | 98.48 | 48.31 | 15.10 | 98.43 | 48.28 | 15.50 | 98.39 | 48.33 | 15.20 |
| RMBMU | 97.86 | 46.38 | 11.40 | 97.92 | 46.61 | 11.10 | 97.95 | 46.54 | 11.50 | 97.89 | 46.47 | 11.20 | 97.94 | 46.53 | 11.30 |
| DABF | 98.24 | 47.04 | 10.60 | 98.18 | 47.17 | 10.90 | 98.22 | 47.13 | 10.70 | 98.19 | 47.06 | 10.80 | 98.23 | 47.15 | 10.50 |
| AdvUA | 98.49 | 47.38 | 9.70 | 98.42 | 47.34 | 9.80 | 98.45 | 47.41 | 9.60 | 98.48 | 47.35 | 9.90 | 98.43 | 47.39 | 9.70 |
| EVMUS | 98.88 | 48.42 | 8.90 | 98.91 | 48.47 | 8.70 | 98.93 | 48.36 | 8.80 | 98.87 | 48.51 | 8.90 | 98.89 | 48.48 | 8.80 |
| DDPA | 97.49 | 45.86 | 89.30 | 97.52 | 45.98 | 89.10 | 97.54 | 46.01 | 89.40 | 97.48 | 45.93 | 89.20 | 97.53 | 46.07 | 89.00 |

Table 25: VGG-16+CIFAR 100 20 target

|  | First-Order | | | Second-Order | | | Unrolling SGD | | | Amnesiac | | | SISA (shard 3) | | |
|---|---|---|---|---|---|---|---|---|---|---|---|---|---|---|---|
|  | TA | BA | ASR | TA | BA | ASR | TA | BA | ASR | TA | BA | ASR | TA | BA | ASR |
| AwoP | 98.76 | 46.37 | 12.60 | 98.54 | 47.12 | 13.80 | 98.62 | 46.78 | 12.40 | 98.34 | 47.35 | 13.90 | 98.48 | 47.42 | 12.80 |
| MUECPA | 97.98 | 47.18 | 9.80 | 98.02 | 46.89 | 10.30 | 97.74 | 47.01 | 10.10 | 97.92 | 46.78 | 10.40 | 98.05 | 47.09 | 10.20 |
| SSCSF | 99.24 | 47.56 | 6.40 | 99.13 | 47.62 | 6.80 | 99.16 | 47.59 | 6.30 | 99.22 | 47.53 | 6.60 | 99.18 | 47.58 | 6.50 |
| BAU | 98.54 | 46.34 | 7.30 | 98.49 | 46.76 | 7.90 | 98.62 | 46.43 | 7.80 | 98.55 | 46.39 | 7.60 | 98.58 | 46.41 | 7.50 |
| UBA-Inf | 98.46 | 48.21 | 11.20 | 98.31 | 48.14 | 10.70 | 98.38 | 48.18 | 11.10 | 98.43 | 48.22 | 10.90 | 98.39 | 48.12 | 11.30 |
| RMBMU | 97.78 | 46.78 | 8.40 | 97.65 | 46.52 | 8.70 | 97.71 | 46.65 | 8.50 | 97.68 | 46.71 | 8.60 | 97.76 | 46.59 | 8.30 |
| DABF | 98.12 | 47.21 | 6.70 | 98.05 | 47.14 | 6.90 | 98.18 | 47.27 | 6.80 | 98.22 | 47.34 | 6.60 | 98.19 | 47.29 | 6.50 |
| AdvUA | 98.54 | 47.53 | 5.90 | 98.41 | 47.42 | 6.10 | 98.49 | 47.58 | 6.00 | 98.48 | 47.61 | 5.80 | 98.53 | 47.49 | 5.90 |
| EVMUS | 99.02 | 48.42 | 6.50 | 99.11 | 48.35 | 6.20 | 99.04 | 48.37 | 6.40 | 99.07 | 48.41 | 6.30 | 99.03 | 48.39 | 6.20 |
| DDPA | 97.42 | 45.78 | 88.30 | 97.62 | 46.02 | 87.90 | 97.54 | 45.93 | 88.10 | 97.49 | 45.89 | 87.80 | 97.57 | 46.01 | 88.00 |

Table 26: ResNet-18+Tiny Image Net 5 target

|  | First-Order | | | Second-Order | | | Unrolling SGD | | | Amnesiac | | | SISA (shard 3) | | |
|---|---|---|---|---|---|---|---|---|---|---|---|---|---|---|---|
|  | TA | BA | ASR | TA | BA | ASR | TA | BA | ASR | TA | BA | ASR | TA | BA | ASR |
| AwoP | 98.82 | 41.09 | 18.20 | 98.75 | 40.33 | 17.50 | 98.68 | 41.68 | 18.60 | 98.74 | 41.38 | 18.00 | 98.71 | 41.30 | 17.90 |
| MUECPA | 97.82 | 42.78 | 26.30 | 97.71 | 43.39 | 25.50 | 97.64 | 42.69 | 24.80 | 97.75 | 42.39 | 26.00 | 97.52 | 43.15 | 25.70 |
| SSCSF | 99.12 | 41.42 | 22.40 | 99.14 | 41.73 | 21.80 | 99.17 | 41.67 | 22.10 | 99.02 | 41.38 | 21.90 | 99.11 | 41.80 | 22.00 |
| BAU | 98.34 | 40.73 | 20.60 | 98.45 | 41.42 | 19.30 | 98.33 | 40.67 | 20.20 | 98.28 | 41.32 | 19.80 | 98.36 | 41.32 | 20.40 |
| UBA-Inf | 98.74 | 42.15 | 30.40 | 98.78 | 41.69 | 28.70 | 98.81 | 41.10 | 29.20 | 98.75 | 42.04 | 30.10 | 98.83 | 42.59 | 29.60 |
| RMBMU | 99.36 | 41.71 | 23.70 | 99.25 | 40.82 | 23.00 | 99.11 | 40.51 | 22.50 | 99.14 | 41.02 | 23.40 | 99.09 | 41.02 | 23.10 |
| DABF | 98.32 | 41.63 | 21.50 | 98.18 | 42.02 | 20.70 | 98.39 | 41.97 | 21.30 | 98.28 | 42.04 | 21.40 | 98.34 | 41.14 | 21.20 |
| AdvUA | 98.86 | 41.37 | 20.10 | 98.45 | 41.14 | 19.40 | 98.73 | 41.73 | 19.80 | 98.56 | 41.44 | 20.00 | 98.64 | 42.19 | 19.60 |
| EVMUS | 97.64 | 40.73 | 19.40 | 97.52 | 40.14 | 18.90 | 97.69 | 41.83 | 19.80 | 97.62 | 40.89 | 19.60 | 97.15 | 41.20 | 19.30 |
| DDPA | 98.98 | 41.84 | 83.40 | 98.94 | 40.44 | 82.70 | 98.87 | 43.68 | 83.10 | 98.91 | 42.70 | 83.60 | 98.86 | 42.37 | 83.00 |

Table 27: ResNet-18+Tiny Image Net 10 target

| | First-Order | | | Second-Order | | | Unrolling SGD | | | Amnesiac | | | SISA (shard 3) | | |
|---|---|---|---|---|---|---|---|---|---|---|---|---|---|---|---|
| | TA | BA | ASR | TA | BA | ASR | TA | BA | ASR | TA | BA | ASR | TA | BA | ASR |
| AwoP | 98.35 | 41.72 | 21.80 | 98.41 | 41.19 | 19.60 | 98.45 | 40.92 | 20.30 | 98.39 | 41.42 | 22.00 | 98.46 | 40.87 | 20.80 |
| MUECPA | 97.92 | 42.53 | 19.30 | 97.83 | 41.96 | 18.50 | 97.88 | 42.12 | 17.90 | 97.79 | 42.34 | 18.80 | 97.76 | 42.01 | 18.60 |
| SSCSF | 99.12 | 41.26 | 16.70 | 99.07 | 41.09 | 15.90 | 99.14 | 41.35 | 16.30 | 99.02 | 41.16 | 17.00 | 99.08 | 41.22 | 16.80 |
| BAU | 98.64 | 40.73 | 15.80 | 98.56 | 40.89 | 14.90 | 98.58 | 40.62 | 15.30 | 98.42 | 40.73 | 15.70 | 98.47 | 40.79 | 15.60 |
| UBA-Inf | 98.71 | 41.12 | 24.20 | 98.59 | 40.89 | 23.90 | 98.66 | 41.04 | 24.80 | 98.62 | 41.19 | 24.50 | 98.58 | 41.08 | 24.30 |
| RMBMU | 97.81 | 40.57 | 13.80 | 97.74 | 40.49 | 13.20 | 97.79 | 40.72 | 13.60 | 97.63 | 40.64 | 14.00 | 97.72 | 40.58 | 13.70 |
| DABF | 98.27 | 40.83 | 12.40 | 98.12 | 40.74 | 11.80 | 98.19 | 40.96 | 12.10 | 98.24 | 40.81 | 12.70 | 98.14 | 40.94 | 12.50 |
| AdvUA | 98.43 | 41.01 | 10.60 | 98.37 | 40.92 | 10.30 | 98.32 | 40.89 | 10.10 | 98.39 | 40.98 | 10.80 | 98.28 | 40.86 | 10.50 |
| EVMUS | 99.28 | 41.63 | 9.70 | 99.17 | 41.55 | 8.10 | 99.14 | 41.49 | 9.40 | 99.21 | 41.67 | 10.00 | 99.19 | 41.53 | 8.90 |
| our | 98.54 | 40.37 | 85.40 | 98.62 | 42.44 | 84.90 | 98.49 | 42.56 | 85.10 | 98.57 | 42.61 | 85.30 | 98.52 | 42.48 | 85.20 |

Table 28: ResNet-18+Tiny Image Net 20 target

| | First-Order | | | Second-Order | | | Unrolling SGD | | | Amnesiac | | | SISA (shard 3) | | |
|---|---|---|---|---|---|---|---|---|---|---|---|---|---|---|---|
| | TA | BA | ASR | TA | BA | ASR | TA | BA | ASR | TA | BA | ASR | TA | BA | ASR |
| AwoP | 98.12 | 40.84 | 14.50 | 98.08 | 40.62 | 13.80 | 98.06 | 40.77 | 14.20 | 98.10 | 40.88 | 13.90 | 98.14 | 40.71 | 14.10 |
| MUECPA | 97.84 | 40.53 | 12.60 | 97.72 | 40.68 | 11.80 | 97.78 | 40.42 | 12.30 | 97.79 | 40.57 | 12.10 | 97.74 | 40.49 | 12.40 |
| SSCSF | 99.02 | 41.12 | 9.20 | 98.94 | 41.07 | 9.50 | 98.97 | 41.19 | 9.10 | 98.99 | 41.24 | 9.30 | 99.01 | 41.08 | 9.40 |
| BAU | 98.33 | 39.98 | 7.80 | 98.25 | 40.01 | 8.00 | 98.29 | 40.15 | 7.90 | 98.22 | 39.97 | 8.20 | 98.28 | 40.04 | 7.80 |
| UBA-Inf | 98.57 | 40.76 | 11.20 | 98.49 | 40.62 | 10.70 | 98.54 | 40.84 | 10.90 | 98.52 | 40.71 | 11.00 | 98.51 | 40.68 | 10.80 |
| RMBMU | 97.68 | 39.92 | 8.60 | 97.62 | 39.84 | 8.20 | 97.63 | 40.02 | 8.50 | 97.65 | 39.95 | 8.40 | 97.64 | 40.01 | 8.30 |
| DABF | 98.12 | 40.16 | 7.40 | 98.08 | 40.09 | 7.70 | 98.15 | 40.22 | 7.50 | 98.11 | 40.13 | 7.80 | 98.09 | 40.19 | 7.60 |
| AdvUA | 98.31 | 40.49 | 6.90 | 98.25 | 40.36 | 6.80 | 98.29 | 40.42 | 7.00 | 98.27 | 40.54 | 6.90 | 98.28 | 40.46 | 6.90 |
| EVMUS | 99.02 | 41.07 | 7.50 | 98.99 | 40.92 | 7.30 | 99.01 | 41.12 | 7.40 | 99.00 | 41.05 | 7.50 | 98.98 | 41.08 | 7.30 |
| DDPA | 98.42 | 42.15 | 85.60 | 98.38 | 42.04 | 84.90 | 98.45 | 42.22 | 85.40 | 98.41 | 42.18 | 85.50 | 98.43 | 42.11 | 85.30 |

Table 29: LLama-3b+SST-2 5 target

| | First-Order | | | Second-Order | | | Unrolling SGD | | | Amnesiac | | | SISA (shard 3) | | |
|---|---|---|---|---|---|---|---|---|---|---|---|---|---|---|---|
| | TA | BA | ASR | TA | BA | ASR | TA | BA | ASR | TA | BA | ASR | TA | BA | ASR |
| AwoP | 93.10 | 90.42 | 31.80 | 93.35 | 90.58 | 32.50 | 93.20 | 90.30 | 21.50 | 93.45 | 90.50 | 32.10 | 93.05 | 90.18 | 30.90 |
| MUECPA | 93.42 | 89.85 | 24.50 | 93.25 | 89.78 | 25.10 | 93.10 | 89.65 | 30.80 | 93.22 | 89.70 | 35.30 | 92.95 | 89.50 | 34.20 |
| SSCSF | 94.25 | 91.38 | 36.40 | 94.08 | 91.20 | 36.70 | 93.88 | 91.00 | 35.80 | 94.12 | 91.30 | 36.50 | 93.80 | 91.05 | 36.00 |
| BAU | 91.80 | 89.89 | 28.50 | 91.70 | 89.74 | 31.20 | 91.60 | 89.65 | 30.80 | 91.50 | 89.55 | 31.50 | 91.40 | 89.38 | 30.40 |
| UBA-Inf | 94.10 | 91.41 | 40.10 | 94.05 | 91.34 | 38.80 | 93.92 | 91.18 | 38.00 | 94.00 | 91.28 | 37.60 | 93.85 | 91.08 | 37.20 |
| RMBMU | 92.78 | 90.00 | 33.70 | 92.65 | 89.88 | 34.10 | 92.50 | 89.72 | 33.80 | 92.58 | 89.85 | 34.30 | 92.38 | 89.66 | 33.40 |
| DABF | 93.00 | 90.39 | 32.60 | 92.90 | 90.28 | 33.10 | 92.78 | 90.10 | 32.80 | 92.75 | 90.15 | 33.40 | 92.50 | 89.88 | 32.20 |
| AdvUA | 92.60 | 90.20 | 30.20 | 92.55 | 90.12 | 31.00 | 92.45 | 90.05 | 30.80 | 92.40 | 90.10 | 30.50 | 92.30 | 89.85 | 30.00 |
| EVMUS | 94.30 | 91.62 | 27.20 | 94.15 | 91.45 | 27.80 | 94.00 | 91.25 | 26.80 | 94.10 | 91.55 | 27.40 | 93.85 | 91.28 | 36.70 |
| DDPA | 94.50 | 91.75 | 83.50 | 94.68 | 91.89 | 84.10 | 94.72 | 91.70 | 83.20 | 94.80 | 91.80 | 83.80 | 94.55 | 91.60 | 83.00 |

Table 30: LLama-3b+SST-2 10 target

| | First-Order | | | Second-Order | | | Unrolling SGD | | | Amnesiac | | | SISA (shard 3) | | |
|---|---|---|---|---|---|---|---|---|---|---|---|---|---|---|---|
| | TA | BA | ASR | TA | BA | ASR | TA | BA | ASR | TA | BA | ASR | TA | BA | ASR |
| AwoP | 92.95 | 90.35 | 18.60 | 93.18 | 90.52 | 17.20 | 93.10 | 90.28 | 19.30 | 93.32 | 90.44 | 18.90 | 92.90 | 90.10 | 17.50 |
| MUECPA | 93.10 | 89.70 | 20.40 | 93.04 | 89.58 | 18.60 | 92.88 | 89.42 | 19.70 | 93.01 | 89.55 | 18.90 | 92.75 | 89.30 | 20.20 |
| SSCSF | 94.05 | 91.10 | 21.20 | 94.01 | 91.05 | 19.80 | 93.92 | 90.85 | 22.50 | 94.00 | 91.00 | 20.90 | 93.80 | 90.70 | 21.70 |
| BAU | 91.60 | 89.50 | 12.50 | 91.50 | 89.38 | 13.90 | 91.42 | 89.25 | 11.70 | 91.38 | 89.30 | 12.80 | 91.25 | 89.05 | 13.20 |
| UBA-Inf | 93.78 | 91.20 | 22.10 | 93.65 | 91.10 | 23.60 | 93.52 | 91.00 | 21.50 | 93.68 | 91.15 | 22.80 | 93.45 | 90.85 | 23.20 |
| RMBMU | 92.45 | 89.85 | 15.40 | 92.28 | 89.65 | 14.30 | 92.12 | 89.50 | 16.20 | 92.20 | 89.55 | 15.80 | 91.95 | 89.28 | 14.70 |
| DABF | 92.50 | 90.05 | 14.70 | 92.40 | 89.95 | 13.20 | 92.35 | 89.85 | 15.80 | 92.30 | 89.88 | 14.10 | 92.10 | 89.70 | 13.80 |
| AdvUA | 94.20 | 91.35 | 21.60 | 94.10 | 91.25 | 20.90 | 93.92 | 91.10 | 22.50 | 94.00 | 91.28 | 21.30 | 93.85 | 91.00 | 20.60 |
| EVMUS | 92.48 | 89.98 | 12.90 | 92.35 | 89.85 | 14.70 | 92.25 | 89.75 | 13.30 | 92.30 | 89.80 | 14.10 | 92.15 | 89.60 | 13.50 |
| DDPA | 94.50 | 91.60 | 75.50 | 94.68 | 91.74 | 76.80 | 94.72 | 91.65 | 75.20 | 94.80 | 91.70 | 76.30 | 94.55 | 91.50 | 75.00 |

Table 31: LLama-3b+SST-2 20 target

| | First-Order | | | Second-Order | | | Unrolling SGD | | | Amnesiac | | | SISA (shard 3) | | |
|---|---|---|---|---|---|---|---|---|---|---|---|---|---|---|---|
| | TA | BA | ASR | TA | BA | ASR | TA | BA | ASR | TA | BA | ASR | TA | BA | ASR |
| AwoP | 92.45 | 90.10 | 9.70 | 92.62 | 90.18 | 11.20 | 92.38 | 89.95 | 10.10 | 92.55 | 90.05 | 10.40 | 92.20 | 89.80 | 9.50 |
| MUECPA | 92.30 | 89.35 | 12.80 | 92.18 | 89.20 | 13.50 | 92.04 | 89.15 | 10.90 | 92.10 | 89.25 | 12.00 | 91.95 | 89.05 | 11.70 |
| SSCSF | 93.65 | 90.75 | 14.50 | 93.72 | 90.80 | 15.70 | 93.50 | 90.65 | 13.90 | 93.58 | 90.70 | 14.80 | 93.40 | 90.50 | 13.50 |
| BAU | 90.85 | 89.00 | 7.50 | 90.78 | 88.85 | 8.20 | 90.62 | 88.70 | 9.00 | 90.68 | 88.75 | 8.50 | 90.50 | 88.50 | 7.80 |
| UBA-Inf | 93.28 | 91.00 | 16.30 | 93.12 | 90.85 | 17.10 | 93.05 | 90.80 | 15.20 | 93.20 | 90.90 | 16.40 | 93.00 | 90.75 | 15.60 |
| RMBMU | 91.82 | 89.20 | 11.30 | 91.68 | 89.10 | 10.60 | 91.50 | 88.95 | 11.50 | 91.60 | 89.00 | 10.90 | 91.45 | 88.80 | 11.10 |
| DABF | 92.10 | 89.55 | 8.80 | 92.00 | 89.40 | 9.50 | 91.88 | 89.30 | 10.10 | 91.95 | 89.35 | 9.70 | 91.70 | 89.10 | 9.00 |
| AdvUA | 94.10 | 91.20 | 13.50 | 94.05 | 91.15 | 14.20 | 93.92 | 91.00 | 12.80 | 94.00 | 91.05 | 13.90 | 93.85 | 90.85 | 12.60 |
| EVMUS | 91.90 | 89.40 | 6.50 | 91.85 | 89.35 | 7.20 | 91.72 | 89.20 | 7.80 | 91.80 | 89.25 | 7.40 | 91.65 | 89.05 | 6.90 |
| DDPA | 94.72 | 91.80 | 70.80 | 94.84 | 91.92 | 72.10 | 94.88 | 91.86 | 71.30 | 94.96 | 91.90 | 71.80 | 94.78 | 91.75 | 71.20 |

Table 32: VGG-16+CIFAR 100 5 target (Untargeted)

| | First-Order | | | Second-Order | | | Unrolling SGD | | | Amnesiac | | | SISA (shard 3) | | |
|---|---|---|---|---|---|---|---|---|---|---|---|---|---|---|---|
| | TA | BA | ASR | TA | BA | ASR | TA | BA | ASR | TA | BA | ASR | TA | BA | ASR |
| AwoP | 96.84 | 41.67 | 39.10 | 96.79 | 41.92 | 38.40 | 96.75 | 41.54 | 38.90 | 96.78 | 41.73 | 39.40 | 96.82 | 41.86 | 39.20 |
| MUECPA | 95.12 | 43.15 | 35.60 | 95.17 | 43.41 | 34.80 | 95.21 | 43.08 | 35.20 | 95.15 | 43.23 | 35.40 | 95.18 | 43.19 | 35.70 |
| SSCSF | 97.32 | 42.36 | 31.70 | 97.27 | 42.51 | 31.40 | 97.24 | 42.19 | 31.80 | 97.29 | 42.33 | 31.60 | 97.34 | 42.45 | 31.50 |
| BAU | 95.28 | 39.12 | 33.60 | 95.36 | 39.37 | 33.40 | 95.31 | 39.08 | 33.70 | 95.26 | 39.24 | 33.50 | 95.29 | 39.18 | 33.80 |
| UBA-Inf | 96.49 | 43.64 | 37.50 | 96.44 | 44.09 | 36.80 | 96.41 | 43.72 | 37.10 | 96.46 | 43.86 | 37.30 | 96.52 | 43.91 | 37.40 |
| RMBMU | 94.62 | 41.21 | 31.80 | 94.68 | 41.38 | 31.20 | 94.59 | 41.04 | 31.50 | 94.55 | 41.19 | 31.60 | 94.61 | 41.24 | 31.70 |
| DABF | 96.23 | 41.45 | 30.90 | 96.17 | 41.61 | 30.40 | 96.12 | 41.32 | 30.80 | 96.19 | 41.47 | 30.70 | 96.21 | 41.49 | 30.60 |
| AdvUA | 95.48 | 41.32 | 29.50 | 95.42 | 41.57 | 29.30 | 95.39 | 41.29 | 29.40 | 95.46 | 41.42 | 29.60 | 95.44 | 41.38 | 29.50 |
| EVMUS | 96.89 | 43.47 | 28.70 | 96.94 | 43.64 | 28.50 | 96.83 | 43.29 | 28.60 | 96.87 | 43.52 | 28.80 | 96.91 | 43.58 | 28.70 |
| DDPA | 94.15 | 37.62 | 93.70 | 94.09 | 38.14 | 93.20 | 94.03 | 37.78 | 93.40 | 94.11 | 37.92 | 93.50 | 94.18 | 38.03 | 93.60 |

Table 33: VGG-16+CIFAR 100 10 target (Untargeted)

| | First-Order | | | Second-Order | | | Unrolling SGD | | | Amnesiac | | | SISA (shard 3) | | |
|---|---|---|---|---|---|---|---|---|---|---|---|---|---|---|---|
| | TA | BA | ASR | TA | BA | ASR | TA | BA | ASR | TA | BA | ASR | TA | BA | ASR |
| AwoP | 96.78 | 41.25 | 27.30 | 96.71 | 41.62 | 27.10 | 96.65 | 41.14 | 26.80 | 96.54 | 40.89 | 26.90 | 96.62 | 41.04 | 27.20 |
| MUECPA | 95.32 | 41.89 | 22.70 | 95.28 | 42.01 | 22.30 | 95.15 | 41.65 | 21.90 | 95.11 | 41.32 | 22.10 | 95.19 | 41.54 | 22.50 |
| SSCSF | 97.39 | 42.10 | 19.90 | 97.42 | 42.25 | 19.70 | 97.35 | 42.04 | 19.50 | 97.27 | 41.81 | 19.60 | 97.31 | 41.96 | 19.80 |
| BAU | 95.58 | 40.45 | 18.40 | 95.64 | 40.58 | 18.30 | 95.42 | 40.27 | 18.00 | 95.33 | 40.14 | 18.20 | 95.41 | 40.31 | 18.10 |
| UBA-Inf | 96.61 | 42.74 | 28.20 | 96.72 | 42.82 | 28.00 | 96.48 | 42.34 | 27.50 | 96.39 | 41.92 | 27.80 | 96.52 | 42.10 | 27.70 |
| RMBMU | 94.71 | 40.85 | 23.70 | 94.69 | 41.01 | 23.40 | 94.52 | 40.67 | 23.30 | 94.45 | 40.48 | 23.20 | 94.58 | 40.72 | 23.50 |
| DABF | 96.39 | 40.98 | 20.10 | 96.41 | 41.23 | 19.90 | 96.37 | 40.85 | 19.60 | 96.29 | 40.72 | 19.80 | 96.34 | 40.93 | 20.00 |
| AdvUA | 95.54 | 39.72 | 17.30 | 95.61 | 39.89 | 17.10 | 95.47 | 39.53 | 17.00 | 95.42 | 39.41 | 17.20 | 95.53 | 39.66 | 17.50 |
| EVMUS | 96.89 | 37.78 | 16.40 | 96.95 | 37.93 | 16.20 | 96.83 | 37.62 | 16.10 | 96.77 | 37.47 | 16.30 | 96.91 | 37.68 | 16.50 |
| DDPA | 94.25 | 35.92 | 96.30 | 94.32 | 36.24 | 95.90 | 94.11 | 35.88 | 96.10 | 94.08 | 35.71 | 96.20 | 94.17 | 35.93 | 96.00 |

Table 34: VGG-16+CIFAR 100 20 target (Untargeted)

| | First-Order | | | Second-Order | | | Unrolling SGD | | | Amnesiac | | | SISA (shard 3) | | |
|---|---|---|---|---|---|---|---|---|---|---|---|---|---|---|---|
| | TA | BA | ASR | TA | BA | ASR | TA | BA | ASR | TA | BA | ASR | TA | BA | ASR |
| AwoP | 96.74 | 39.12 | 15.20 | 96.78 | 39.53 | 14.30 | 96.63 | 39.01 | 16.40 | 96.55 | 38.67 | 13.80 | 96.69 | 38.74 | 15.70 |
| MUECPA | 95.29 | 39.76 | 11.30 | 95.21 | 40.12 | 10.90 | 95.14 | 39.64 | 12.20 | 95.07 | 39.49 | 10.70 | 95.16 | 39.58 | 11.80 |
| SSCSF | 97.42 | 40.15 | 9.70 | 97.35 | 40.36 | 10.20 | 97.28 | 39.98 | 10.40 | 97.24 | 39.87 | 9.50 | 97.37 | 40.01 | 9.90 |
| BAU | 95.64 | 38.47 | 7.60 | 95.59 | 38.69 | 8.20 | 95.45 | 38.28 | 7.80 | 95.33 | 38.11 | 8.40 | 95.51 | 38.33 | 7.90 |
| UBA-Inf | 96.72 | 40.29 | 18.20 | 96.79 | 40.56 | 19.40 | 96.63 | 40.07 | 17.60 | 96.51 | 39.68 | 18.10 | 96.64 | 40.11 | 17.90 |
| RMBMU | 94.76 | 37.83 | 12.40 | 94.68 | 38.14 | 13.10 | 94.53 | 37.59 | 12.70 | 94.39 | 37.36 | 13.20 | 94.63 | 37.78 | 12.60 |
| DABF | 96.45 | 38.32 | 9.40 | 96.39 | 38.52 | 10.10 | 96.34 | 38.13 | 9.80 | 96.26 | 37.92 | 10.30 | 96.37 | 38.23 | 9.70 |
| AdvUA | 95.53 | 37.14 | 8.70 | 95.57 | 37.41 | 7.90 | 95.39 | 36.92 | 7.50 | 95.28 | 36.67 | 8.20 | 95.47 | 37.01 | 8.40 |
| EVMUS | 96.91 | 36.63 | 6.30 | 96.87 | 36.71 | 6.50 | 96.76 | 36.34 | 6.80 | 96.62 | 36.14 | 6.60 | 96.79 | 36.53 | 6.40 |
| DDPA | 94.19 | 34.68 | 91.60 | 94.26 | 35.02 | 91.20 | 94.12 | 34.59 | 90.80 | 94.05 | 34.43 | 90.90 | 94.18 | 34.61 | 91.40 |

Table 35: ResNet-18+Tiny Image Net 5 target (Untargeted)

|  | First-Order | | | Second-Order | | | Unrolling SGD | | | Amnesiac | | | SISA (shard 3) | | |
| --- | --- | --- | --- | --- | --- | --- | --- | --- | --- | --- | --- | --- | --- | --- | --- |
|  | TA | BA | ASR | TA | BA | ASR | TA | BA | ASR | TA | BA | ASR | TA | BA | ASR |
| AwoP | 98.76 | 32.92 | 41.20 | 98.65 | 33.14 | 39.50 | 98.58 | 32.84 | 40.60 | 98.63 | 32.02 | 41.80 | 98.51 | 32.87 | 39.90 |
| MUECPA | 97.83 | 33.78 | 37.90 | 97.72 | 34.22 | 36.40 | 97.61 | 33.69 | 38.10 | 97.75 | 34.08 | 37.60 | 97.55 | 33.92 | 36.90 |
| SSCSF | 99.12 | 31.82 | 38.60 | 99.08 | 32.14 | 37.90 | 99.11 | 31.94 | 38.20 | 99.05 | 32.06 | 37.50 | 98.99 | 31.78 | 36.80 |
| BAU | 98.29 | 30.73 | 35.90 | 98.41 | 31.12 | 36.40 | 98.27 | 30.78 | 35.70 | 98.34 | 31.02 | 35.80 | 98.31 | 30.92 | 36.20 |
| UBA-Inf | 98.64 | 32.15 | 44.10 | 98.68 | 32.42 | 42.70 | 98.73 | 32.08 | 43.20 | 98.69 | 32.34 | 44.50 | 98.61 | 32.28 | 42.90 |
| RMBMU | 99.21 | 31.64 | 39.30 | 99.14 | 31.52 | 38.60 | 99.05 | 31.29 | 39.80 | 99.11 | 31.38 | 38.50 | 99.02 | 31.24 | 37.90 |
| DABF | 98.24 | 30.91 | 36.80 | 98.31 | 31.18 | 37.30 | 98.27 | 30.89 | 36.50 | 98.35 | 31.02 | 37.10 | 98.22 | 30.97 | 36.70 |
| AdvUA | 98.66 | 31.12 | 35.70 | 98.45 | 30.89 | 34.60 | 98.69 | 31.28 | 35.20 | 98.54 | 30.98 | 35.10 | 98.64 | 31.22 | 34.90 |
| EVMUS | 97.48 | 30.23 | 35.40 | 97.35 | 29.84 | 34.90 | 97.51 | 30.41 | 35.10 | 97.42 | 30.12 | 35.30 | 97.19 | 30.32 | 34.80 |
| DDPA | 98.98 | 29.84 | 86.20 | 98.91 | 28.94 | 84.70 | 98.87 | 29.21 | 85.30 | 98.79 | 28.76 | 86.70 | 98.82 | 28.89 | 85.60 |

Table 36: ResNet-18+Tiny Image Net 10 target (Untargeted)

|  | First-Order | | | Second-Order | | | Unrolling SGD | | | Amnesiac | | | SISA (shard 3) | | |
| --- | --- | --- | --- | --- | --- | --- | --- | --- | --- | --- | --- | --- | --- | --- | --- |
|  | TA | BA | ASR | TA | BA | ASR | TA | BA | ASR | TA | BA | ASR | TA | BA | ASR |
| AwoP | 98.71 | 33.74 | 33.20 | 98.68 | 34.12 | 31.90 | 98.56 | 33.61 | 32.50 | 98.58 | 33.79 | 33.10 | 98.52 | 33.52 | 32.30 |
| MUECPA | 97.66 | 32.49 | 30.60 | 97.58 | 32.78 | 29.80 | 97.53 | 32.45 | 30.30 | 97.61 | 32.64 | 30.10 | 97.49 | 32.35 | 29.60 |
| SSCSF | 98.98 | 31.22 | 31.10 | 98.85 | 31.54 | 30.40 | 98.89 | 31.38 | 30.90 | 98.76 | 31.42 | 30.50 | 98.84 | 31.17 | 29.80 |
| BAU | 98.12 | 29.94 | 28.30 | 98.26 | 30.18 | 28.70 | 98.05 | 29.86 | 28.50 | 98.09 | 29.92 | 28.60 | 98.11 | 29.84 | 28.40 |
| UBA-Inf | 98.62 | 31.46 | 35.20 | 98.71 | 31.82 | 34.50 | 98.74 | 31.38 | 34.80 | 98.69 | 31.52 | 35.10 | 98.65 | 31.28 | 34.30 |
| RMBMU | 98.96 | 30.92 | 32.10 | 98.85 | 30.72 | 31.60 | 98.79 | 30.58 | 32.40 | 98.82 | 30.67 | 31.90 | 98.74 | 30.51 | 31.20 |
| DABF | 97.89 | 29.82 | 29.50 | 98.04 | 30.04 | 30.20 | 97.91 | 29.76 | 29.40 | 98.07 | 29.82 | 29.70 | 97.95 | 29.68 | 29.30 |
| AdvUA | 98.51 | 30.28 | 28.60 | 98.38 | 29.97 | 27.90 | 98.64 | 30.36 | 28.30 | 98.42 | 30.02 | 28.40 | 98.54 | 30.14 | 27.80 |
| EVMUS | 97.48 | 28.92 | 28.20 | 97.32 | 28.64 | 27.80 | 97.51 | 28.98 | 28.10 | 97.42 | 28.72 | 28.30 | 97.19 | 28.84 | 27.90 |
| DDPA | 98.92 | 27.68 | 84.50 | 98.87 | 26.94 | 83.10 | 98.81 | 27.24 | 83.70 | 98.74 | 26.78 | 84.20 | 98.79 | 26.89 | 83.50 |

Table 37: ResNet-18+Tiny Image Net 20 target (Untargeted)

|  | First-Order | | | Second-Order | | | Unrolling SGD | | | Amnesiac | | | SISA (shard 3) | | |
| --- | --- | --- | --- | --- | --- | --- | --- | --- | --- | --- | --- | --- | --- | --- | --- |
|  | TA | BA | ASR | TA | BA | ASR | TA | BA | ASR | TA | BA | ASR | TA | BA | ASR |
| AwoP | 98.52 | 30.12 | 24.10 | 98.43 | 30.48 | 22.90 | 98.32 | 29.91 | 23.30 | 98.36 | 30.15 | 24.50 | 98.27 | 30.08 | 22.80 |
| MUECPA | 97.39 | 29.82 | 19.70 | 97.28 | 29.64 | 18.90 | 97.42 | 29.55 | 20.20 | 97.33 | 29.61 | 19.80 | 97.24 | 29.48 | 19.30 |
| SSCSF | 98.67 | 28.92 | 17.60 | 98.58 | 29.18 | 16.90 | 98.61 | 29.04 | 17.30 | 98.52 | 28.96 | 16.80 | 98.55 | 29.12 | 17.10 |
| BAU | 97.82 | 27.74 | 14.80 | 97.93 | 28.08 | 15.20 | 97.79 | 27.92 | 14.30 | 97.76 | 28.01 | 15.40 | 97.85 | 27.88 | 14.70 |
| UBA-Inf | 98.24 | 28.91 | 25.00 | 98.31 | 29.32 | 24.20 | 98.37 | 28.84 | 24.70 | 98.29 | 29.12 | 25.10 | 98.26 | 28.98 | 24.50 |
| RMBMU | 98.53 | 27.62 | 21.30 | 98.41 | 27.41 | 20.80 | 98.37 | 27.28 | 21.50 | 98.42 | 27.53 | 20.90 | 98.35 | 27.39 | 20.40 |
| DABF | 97.24 | 26.73 | 16.20 | 97.38 | 27.04 | 17.10 | 97.19 | 26.84 | 16.80 | 97.31 | 26.91 | 17.00 | 97.25 | 26.78 | 16.50 |
| AdvUA | 97.98 | 27.18 | 14.60 | 97.81 | 26.92 | 13.90 | 98.07 | 27.25 | 14.30 | 97.89 | 27.02 | 14.10 | 97.95 | 27.14 | 13.80 |
| EVMUS | 96.88 | 25.72 | 11.40 | 96.72 | 25.58 | 10.80 | 96.97 | 25.91 | 11.20 | 96.85 | 25.79 | 11.00 | 96.69 | 25.64 | 10.60 |
| DDPA | 98.62 | 24.73 | 83.50 | 98.57 | 24.42 | 82.90 | 98.49 | 24.68 | 83.20 | 98.42 | 24.34 | 83.80 | 98.51 | 24.51 | 83.10 |

Table 38: LLama-3b+SST-2 5 target (Untargeted)

|  | First-Order | | | Second-Order | | | Unrolling SGD | | | Amnesiac | | | SISA (shard 3) | | |
| --- | --- | --- | --- | --- | --- | --- | --- | --- | --- | --- | --- | --- | --- | --- | --- |
|  | TA | BA | ASR | TA | BA | ASR | TA | BA | ASR | TA | BA | ASR | TA | BA | ASR |
| AwoP | 91.45 | 83.92 | 28.70 | 91.72 | 84.25 | 29.50 | 91.34 | 83.65 | 28.10 | 91.18 | 83.87 | 27.80 | 91.56 | 83.74 | 28.30 |
| MUECPA | 92.32 | 82.68 | 24.10 | 91.94 | 83.12 | 24.90 | 91.61 | 82.48 | 25.30 | 91.85 | 82.94 | 24.60 | 91.78 | 82.67 | 24.30 |
| SSCSF | 93.14 | 84.34 | 26.90 | 92.88 | 84.02 | 26.20 | 92.61 | 83.86 | 27.10 | 92.92 | 84.28 | 26.50 | 92.74 | 83.96 | 26.80 |
| BAU | 89.32 | 81.84 | 22.50 | 89.64 | 82.32 | 22.10 | 89.43 | 81.67 | 22.80 | 88.98 | 81.45 | 21.90 | 89.12 | 81.62 | 22.30 |
| UBA-Inf | 92.61 | 85.07 | 30.20 | 92.89 | 84.86 | 29.80 | 92.52 | 84.41 | 30.50 | 92.76 | 84.92 | 30.10 | 92.47 | 84.36 | 29.90 |
| RMBMU | 90.92 | 83.12 | 23.90 | 90.74 | 82.87 | 24.30 | 90.55 | 82.62 | 23.70 | 90.31 | 82.45 | 23.50 | 90.48 | 82.78 | 24.10 |
| DABF | 91.43 | 83.47 | 25.80 | 91.26 | 83.32 | 26.30 | 91.14 | 83.04 | 25.90 | 91.08 | 83.11 | 26.50 | 91.21 | 83.25 | 25.60 |
| AdvUA | 89.92 | 82.56 | 20.30 | 90.01 | 82.42 | 21.10 | 89.64 | 82.38 | 20.90 | 89.32 | 82.14 | 20.50 | 89.87 | 82.53 | 20.70 |
| EVMUS | 91.76 | 84.56 | 21.20 | 91.52 | 84.12 | 21.80 | 91.34 | 84.02 | 21.50 | 91.23 | 84.28 | 21.90 | 91.45 | 84.14 | 21.60 |
| our | 93.02 | 81.45 | 85.30 | 93.26 | 81.62 | 85.80 | 93.18 | 81.58 | 85.60 | 93.41 | 81.73 | 86.10 | 93.09 | 81.51 | 85.70 |

Table 39: LLama-3b+SST-2 10 target (Untargeted)

| | First-Order | | | Second-Order | | | Unrolling SGD | | | Amnesiac | | | SISA (shard 3) | | |
|---|---|---|---|---|---|---|---|---|---|---|---|---|---|---|---|
| | TA | BA | ASR | TA | BA | ASR | TA | BA | ASR | TA | BA | ASR | TA | BA | ASR |
| AwoP | 91.38 | 83.78 | 22.30 | 91.52 | 84.05 | 21.80 | 91.14 | 83.54 | 20.60 | 91.08 | 83.71 | 22.10 | 91.42 | 83.69 | 21.50 |
| MUECPA | 92.24 | 82.65 | 17.80 | 91.89 | 83.08 | 18.30 | 91.53 | 82.54 | 17.40 | 91.72 | 82.98 | 18.10 | 91.64 | 82.61 | 17.60 |
| SSCSF | 93.04 | 84.22 | 15.20 | 92.78 | 83.93 | 14.60 | 92.55 | 83.82 | 15.70 | 92.83 | 84.12 | 15.30 | 92.68 | 83.89 | 14.90 |
| BAU | 89.28 | 81.74 | 14.60 | 89.49 | 82.21 | 14.20 | 89.32 | 81.62 | 15.10 | 88.85 | 81.41 | 13.90 | 88.97 | 81.58 | 14.40 |
| UBA-Inf | 92.53 | 85.02 | 24.30 | 92.79 | 84.74 | 23.80 | 92.41 | 84.31 | 24.10 | 92.65 | 84.89 | 24.60 | 92.38 | 84.23 | 23.90 |
| RMBMU | 90.84 | 83.05 | 19.50 | 90.68 | 82.83 | 20.10 | 90.44 | 82.49 | 19.20 | 90.19 | 82.34 | 19.80 | 90.36 | 82.74 | 19.70 |
| DABF | 91.39 | 83.39 | 16.80 | 91.21 | 83.26 | 17.40 | 91.08 | 82.97 | 16.50 | 91.02 | 83.08 | 17.20 | 91.18 | 83.19 | 16.90 |
| AdvUA | 89.85 | 82.42 | 12.30 | 89.94 | 82.29 | 13.20 | 89.57 | 82.17 | 12.80 | 89.24 | 81.95 | 12.10 | 89.73 | 82.36 | 12.50 |
| EVMUS | 91.68 | 84.42 | 13.50 | 91.42 | 84.01 | 14.10 | 91.28 | 83.84 | 13.90 | 91.12 | 84.14 | 14.20 | 91.38 | 84.06 | 13.70 |
| our | 92.95 | 81.38 | 86.20 | 93.18 | 81.57 | 86.70 | 93.06 | 81.55 | 86.50 | 93.32 | 81.68 | 87.10 | 92.98 | 81.44 | 86.60 |

**ASR and Running time with multi-attacks** Figure6-11 evaluates the efficiency of our method in executing multiple attacks within a predefined poisoning budget. The attacker submits 2, 3, or 5 unlearning requests, each targeting a different attack objective. Since other attack methods predefine a single target and cannot dynamically adjust to multiple attacks, they must reconstruct a new poisoned dataset for each target, leading to significant time overhead. In contrast, DDPA uses a single pre-constructed dataset, eliminating the need for additional poisoning stPS. As a result, DDPA efficiently executes multiple attacks across different datasets with minimal time cost. Compared to other methods, DDPA achieves the lowest running time, demonstrating its scalability and efficiency in multi-target attack scenarios.

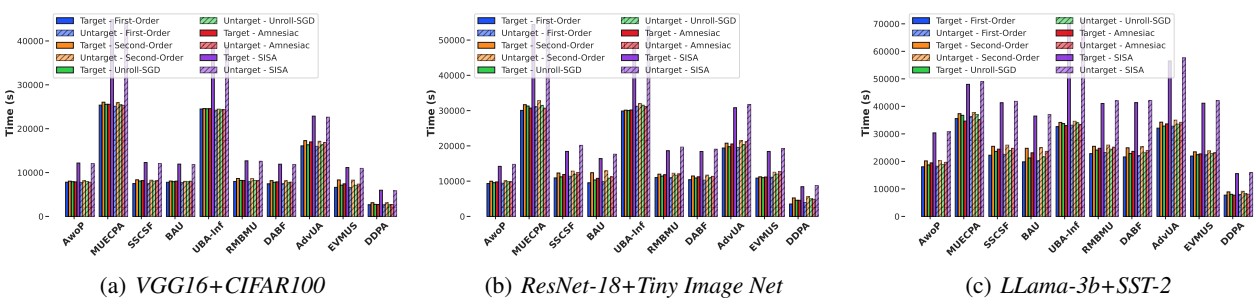

(a) *VGG16+CIFAR100*    (b) *ResNet-18+Tiny Image Net*    (c) *LLama-3b+SST-2*

Figure 6: Time Comparsion with 2 Unlearing Request

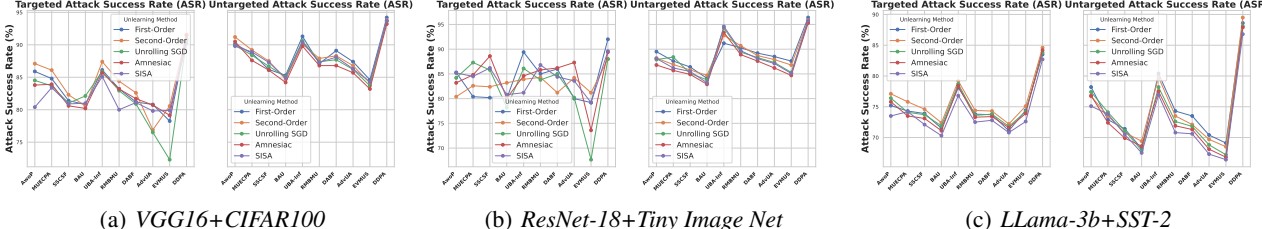

(a) *VGG16+CIFAR100*    (b) *ResNet-18+Tiny Image Net*    (c) *LLama-3b+SST-2*

Figure 7: ASR Comparsion with 2 Unlearing Request

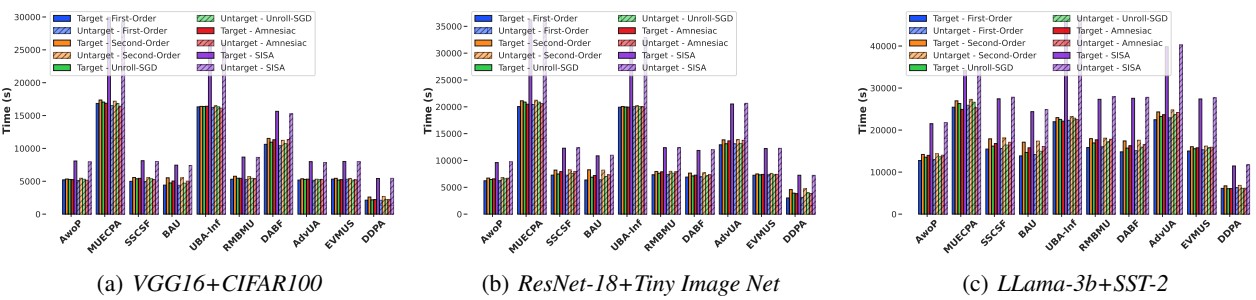

(a) *VGG16+CIFAR100*    (b) *ResNet-18+Tiny Image Net*    (c) *LLama-3b+SST-2*

Figure 8: Time Comparsion with 3 Unlearing Request

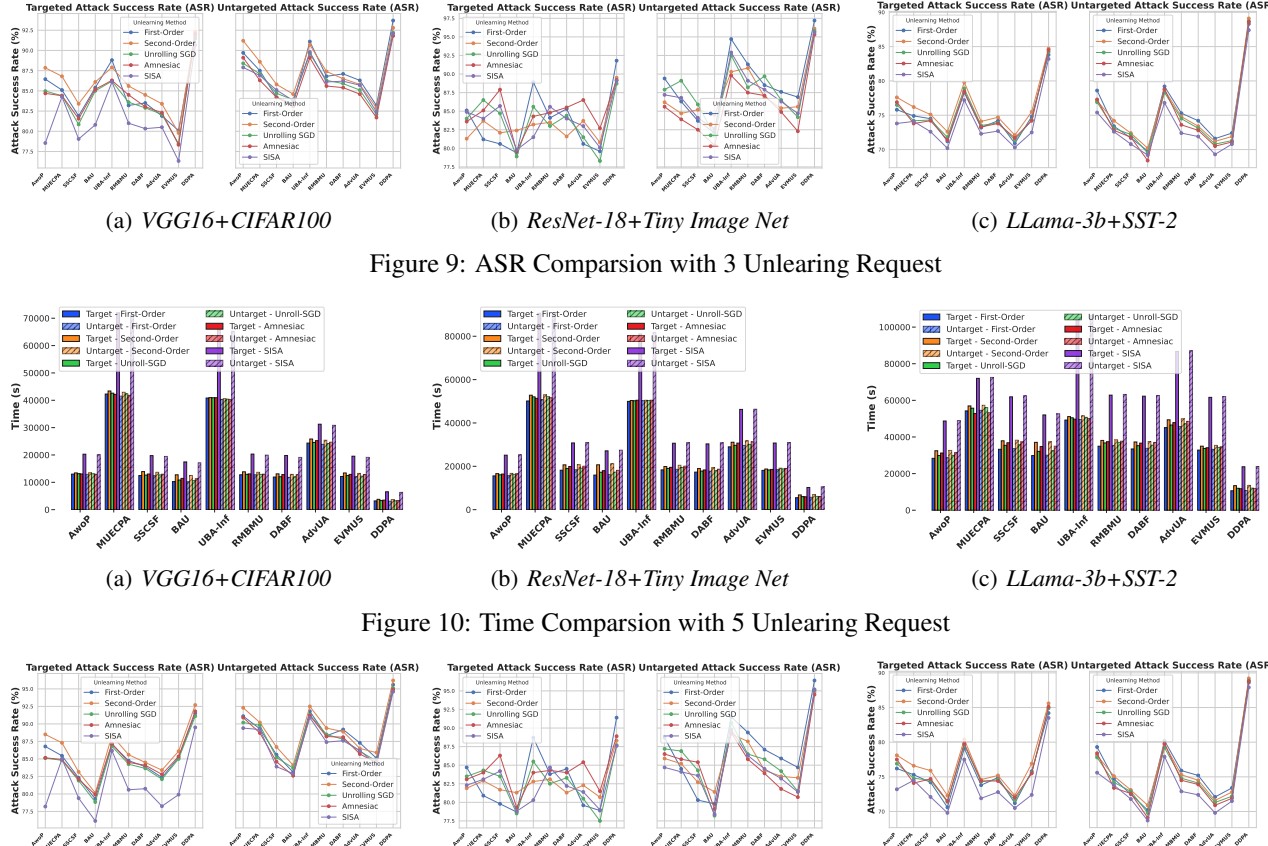

Figure 9: ASR Comparsion with 3 Unlearing Request

Figure 10: Time Comparsion with 5 Unlearing Request

Figure 11: ASR Comparsion with 5 Unlearing Request

## F.2. Parameter Sensitivity

In this section, we conduct more experiments to validate the sensitivity of various parameters in our DDPA method for the unlearning attack task.

**Impact of group centers.** Table 40 evaluates the impact of the number of group centers on ASR, ranging from 5 to 25 across CIFAR-100, Tiny-ImageNet, and SST-2. We observe that ASR increases as the number of group centers grows, as a larger set of group centers provides more precise control over parameter manipulation, enhancing the effectiveness of the attack.

**Influence of training epochs.** Table 41 exhibits the sensitivity of training epochs of our DDPA method by varying them from 30 to 150 for VGG-16 and ResNet-18, and from 2 to 10 for LLama-3b. We observe a monotonic increase in ASR with increasing training epochs. This observation aligns with the fact that more training epochs make unlearning attack methods more effective under suitable data removal ratios. For instance, the ASR of VGG-16 on CIFAR 100 increases significantly from 39.2% at 30 epochs to 92.0% at 150 epochs. Similarly, for ResNet-18 on Tiny ImageNet, the ASR rises from 39.3% at 30 epochs to 90.0% at 150 epochs. For LLama-3b on SST-2, with epochs varying between 2 and 10, the ASR grows from 40.19% to 83.5%. This trend underlines the importance of training duration in influencing the susceptibility of models to unlearning attacks.

**Impact of learning rates.** Table 42 shows the influence of learning rate in our DDPA method by varying it from 0.001 to 0.1. We observed distinct trends between image classification and text classification models. For image classification models, ASR increases as the learning rate grows, whereas for text classification models, ASR starts to decrease with higher learning rates. This phenomenon can be intuitively explained as follows: a larger learning rate enables the algorithm to converge quickly to an optimal solution, which facilitates a higher attack success rate. However, for large-scale models like LLama-3b, an excessive learning rate may cause the optimization process to miss optimal solutions due to larger step sizes,

leading to a decline in ASR. Therefore, it is crucial to determine an optimal learning rate to balance the effectiveness of the unlearning attack while maintaining the performance of the model.

**Influence of unlearning rates.** Table 43 demonstrates the impact of unlearning rate on our DDPA method by varying it from $1 \times 10^{-5}$ to $1.00 \times 10^{-3}$. We observed that as the unlearning rate increases, the attack success rate (ASR) generally improves across all datasets and models. For instance, the ASR of VGG-16 on CIFAR-100 increases from 24.2% at $1 \times 10^{-3}$ to 92.0% at $1 \times 10^{-5}$. However, higher unlearning rates lead to a significant drop in both BA (balanced accuracy) and TA (training accuracy). This effect is particularly pronounced in the LLama-3b+SST-2 model, where the BA drops from 86.32% to 62.99% at an unlearning rate of $1 \times 10^{-3}$. In the targeted attack scenario, such a high unlearning rate disrupts the model's performance, compromising its generalization and accuracy. This highlights the need for carefully selecting the unlearning rate to balance effective unlearning with model robustness.

Table 40: ASR, BA, and TA for Different Models and Group Centers

| Model | Metric | Group Center | | | | |
|---|---|---|---|---|---|---|
| | | 5 | 10 | 15 | 20 | 25 |
| VGG-16+CIFAR 100 | ASR | 88.0 | 88.1 | 89.0 | 92.0 | 96.0 |
| | BA | 44.09 | 44.43 | 44.25 | 43.56 | 43.18 |
| | TA | 94.54 | 95.60 | 95.89 | 95.48 | 95.81 |
| ResNet-18+Tiny Image Net | ASR | 86.0 | 88.0 | 90.0 | 91.0 | 95.4 |
| | BA | 33.53 | 33.06 | 32.10 | 33.11 | 34.18 |
| | TA | 96.36 | 96.81 | 95.24 | 96.02 | 96.07 |
| LLama-3b+SST-2 | ASR | 80.92 | 81.75 | 85.7 | 88.0 | 91.2 |
| | BA | 86.32 | 87.01 | 87.35 | 85.08 | 86.03 |
| | TA | 92.45 | 93.14 | 92.97 | 91.14 | 92.38 |

Table 41: ASR, BA, and TA for Different Models and Epochs

| Model | Metric | Epochs | | | | |
|---|---|---|---|---|---|---|
| | | 30 | 60 | 90 | 120 | 150 |
| VGG-16+CIFAR 100 | ASR | 39.2 | 43.8 | 48.6 | 70.7 | 92.0 |
| | BA | 28.83 | 36.43 | 39.69 | 42.81 | 47.59 |
| | TA | 56.68 | 84.4 | 91.79 | 89.49 | 98.23 |
| ResNet-18+Tiny Image Net | ASR | 39.3 | 45.2 | 65.2 | 83.1 | 90.0 |
| | BA | 25.51 | 34.68 | 36.53 | 40.48 | 43.24 |
| | TA | 54.32 | 84.19 | 91.61 | 92.77 | 98.44 |
| LLama-3b+SST-2 | ASR | 40.19 | 52.48 | 62.47 | 81.22 | 83.5 |
| | BA | 62.99 | 70.06 | 74.64 | 80.48 | 83.98 |
| | TA | 78.78 | 83.26 | 85.44 | 90.48 | 91.06 |

Table 42: ASR, BA, and TA for Different Models and Learning Rates

| Model | Metric | Learning Rate | | | | |
|---|---|---|---|---|---|---|
| | | 0.001 | 0.005 | 0.01 | 0.05 | 0.1 |
| VGG-16+CIFAR 100 | ASR | 92.2 | 91.6 | 90.0 | 92.1 | 90.3 |
| | BA | 47.46 | 47.42 | 46.63 | 46.18 | 46.53 |
| | TA | 98.35 | 98.75 | 98.06 | 98.45 | 98.32 |
| ResNet-18+Tiny Image Net | ASR | 90.3 | 88.2 | 89.9 | 92.3 | 90.6 |
| | BA | 43.9 | 43.61 | 42.82 | 42.29 | 40.21 |
| | TA | 98.11 | 98.13 | 98.45 | 98.32 | 98.29 |
| LLama-3b+SST-2 | ASR | 84.7 | 83.2 | 83.2 | 3.2 | 4.6 |
| | BA | 85.54 | 88.36 | 87.27 | 42.17 | 43.63 |
| | TA | 95.18 | 96.1 | 96.33 | 61.7 | 50.92 |

Table 43: ASR, BA, and TA for Different Models and Unlearning Rates

| Model | Metric | Unlearning Rate | | | | |
|---|---|---|---|---|---|---|
| | | 1.00E-03 | 4.00E-03 | 1.00E-04 | 4.00E-04 | 1.00E-05 |
| VGG-16+CIFAR 100 | ASR | 24.2 | 43.8 | 45.4 | 81.2 | 92.0 |
| | BA | 18.91 | 22.29 | 36.05 | 34.54 | 44.99 |
| | TA | 28.61 | 36.75 | 67.34 | 81.18 | 94.54 |
| ResNet-18+Tiny Image Net | ASR | 22.8 | 40.1 | 57.2 | 78.3 | 88.0 |
| | BA | 11.59 | 20.33 | 32.29 | 33.53 | 35.62 |
| | TA | 29.2 | 36.05 | 74.3 | 85.89 | 94.53 |
| LLama-3b+SST-2 | ASR | 43.6 | 62.5 | 79.4 | 80.1 | 82.3 |
| | BA | 69.15 | 78.78 | 83.26 | 85.36 | 86.32 |
| | TA | 52.95 | 62.99 | 70.06 | 88.42 | 92.45 |

