# OpenReview forum: "Flexible, Efficient, and Stable Adversarial Attacks on Machine Unlearning"
_ICML.cc/2025/Conference — ICML 2025 poster_

### Official Review · Reviewer_twWJ · 2025-03-10

**Overall Recommendation:** 3

**Summary:**

The paper presents the Dynamic Delayed Poisoning Attack (DDPA), a novel adversarial framework for machine unlearning (MU) that overcomes three key limitations of existing MU attacks: inflexibility due to predefined targets, inefficiency in handling multiple requests, and instability from non-convex loss functions.
DDPA allows attackers to specify arbitrary targets post-training, enhancing flexibility. It improves efficiency by using a single poisoned dataset for multiple attacks, reducing computational costs. Stability is achieved through convex polyhedral approximation, which replaces non-convex loss functions with convex ones for more predictable optimization.
Inspired by thrust vector control, DDPA manipulates model parameters via simplex geometry to maximize parameter space coverage, enabling precise attack control and better generalization across MU algorithms.
Experiments on CIFAR-100, Tiny ImageNet, and SST-2 demonstrate DDPA's higher attack success rate (ASR) compared to existing methods, while maintaining stealth and efficiency. The results underscore DDPA's ability to degrade MU performance, emphasizing the need for stronger MU defenses.

## update after rebuttal
All my concerns are addressed. Thus, I maintain my score.

**Claims And Evidence:**

The paper’s claims are largely supported by theoretical analysis and empirical results. Mathematical formulations (e.g., convex polyhedral approximation, simplex geometry) justify the attack’s stability and flexibility, while experiments on CIFAR-100, Tiny ImageNet, and SST-2 validate DDPA’s higher attack success rates (ASR) and efficiency compared to existing methods.

However, some claims need further rigor:

1. **Scalability & Generalization**: Testing on larger-scale or real-world MU applications would strengthen the claim that DDPA is a general solution.
2. **Computational Efficiency**: Direct runtime comparisons with all baselines, beyond aggregated figures, would better support the efficiency claim.
3. **Target-Agnostic Adaptability**: Additional experiments in black-box or limited-access settings would further validate DDPA’s real-world applicability.

Overall, the evidence is strong but could benefit from broader experiments and more realistic constraints.

**Essential References Not Discussed:**

N.A.

**Experimental Designs Or Analyses:**

The experimental design is generally sound, evaluating DDPA across CIFAR-100, Tiny ImageNet, and SST-2 using VGG16, ResNet-18, and LLaMA-3B. The attack success rate (ASR), test accuracy (BA), and runtime are appropriate metrics for assessing attack effectiveness and efficiency.

Strengths:

1. Comprehensive comparisons with 9 baselines across 5 MU methods.
2. Multiple attack settings (targeted, untargeted, varying unlearning ratios).
3. Ablation studies on key components (simplex vs. convex approximation).

Weaknesses:

1. Scalability not fully tested (lacks large-scale datasets like ImageNet-1K).
2. Assumes white-box access, limiting real-world applicability.
3. Efficiency claims lack per-step breakdown (more fine-grained runtime comparisons needed).

Overall, the design is well-structured and relevant, but testing on larger datasets and black-box settings would improve robustness.

**Methods And Evaluation Criteria:**

The methods and evaluation criteria are well-suited to MU attacks. The DDPA attack is logically structured, leveraging convex polyhedral approximation for stability and simplex geometry for flexibility. Experiments on CIFAR-100, Tiny ImageNet, and SST-2 with VGG16, ResNet-18, and LLaMA-3B provide relevant benchmarks.

1. Strengths: Strong theoretical basis, relevant datasets/models, and comprehensive metrics (ASR, BA, runtime).
2. Weaknesses: Limited MU methods tested, lacks large-scale dataset evaluation, and assumes full model knowledge (black-box testing needed).

Overall, the approach is solid, but broader MU methods, datasets, and black-box evaluations would improve generalizability.

**Other Comments Or Suggestions:**

1. line 203-204, typo, $K$ should be $n$.
2. line 252, we propose we propose.

**Other Strengths And Weaknesses:**

Other Strengths:
1. Originality – The paper introduces a novel target-agnostic MU attack using simplex geometry and convex polyhedral approximation, which is a creative combination of optimization and adversarial attack strategies.
2. Significance – Highlights critical vulnerabilities in MU systems, emphasizing the need for robust defenses.
3. Clarity – Well-structured with clear problem motivation, methodology, and experimental setup.

Other Weaknesses:
1. Limited real-world applicability – Assumes white-box access; testing in black-box or federated learning settings would improve practicality.
2. Complexity concerns – The convex approximation approach may not always be computationally feasible in large-scale applications.
3. Presentation – Some theoretical derivations could be explained more intuitively for better readability.

Overall, the paper is innovative and impactful, but addressing practical constraints and clarity in theory would further strengthen it.

**Questions For Authors:**

See comments above.

**Relation To Broader Scientific Literature:**

N.A.

**Theoretical Claims:**

The paper presents several theoretical claims, primarily around convex polyhedral approximation for stability and simplex geometry for flexible attack control. The key results include:

1. Convex Polyhedral Approximation (Theorem 3.5) – Ensures stability by approximating non-convex loss functions.
2. Simplex Geometry (Theorems 3.1–3.4) – Establishes the use of regular simplices for optimal parameter space coverage.
The proofs follow standard mathematical principles (e.g., John’s theorem, Carathéodory’s theorem) and appear logically sound. However:

1. Some derivations, like the proportion of effective parameter space occupied by the simplex (Theorem 3.4), could use more intuitive explanation for clarity.
2. The convex approximation approach is not always differentiable, which may affect optimization feasibility in practice.

Overall, the theoretical claims seem correct and well-supported, though further clarification in some areas could improve accessibility.

---

> ### Author Rebuttal · Authors · 2025-04-01
>
> We would like to thank the reviewer for the helpful and constructive comments. We have tried our best to address your concerns. We will include all the analyses, duscussions, and experimental results in this rebuttal into the submission. Due to the limit of 5000 characters, if anything remains unclear, post-rebuttal comments are appreciated.
>
> **1. Scalability not fully tested (lacks large-scale datasets like ImageNet-1K).**
>
> As shown in Table 1 [ https://anonymous.4open.science/r/sup_materials ], we evaluate all methods on ResNet-18 with ImageNet-1K. Our method achieves the highest unlearning ASR (82.11 ± 1.42) with competitive runtime (22,561 seconds), demonstrating both effectiveness and scalability in large-scale settings.
>
> **2. Computational Efficiency (Claims And Evidence)**
>
> We break down the total runtime into poisoning, training, and unlearning stages for clarity. As shown in Table 2 [ https://anonymous.4open.science/r/sup_materials/timing_comparison_table.pdf ], DDPA consistently incurs lower cost across all three stages. This is because DDPA generates a single poisoned dataset that supports multiple attack targets, while baselines must create a new poisoned dataset for each target. As the number of requests increases, their cumulative training time grows rapidly, whereas DDPA remains low and stable.
>
> **3. Target-Agnostic Adaptability (Claims And Evidence)**
>
> We extend our DDPA method to the setting of federated learning (FL). As shown in Table 5 [ https://anonymous.4open.science/r/sup_materials/fed_comparison_table.pdf ], even in the FL environment, DDPA can still achieve very high attack performance. We will include these results in the submission.
>
> In black-box settings, attackers can train a surrogate model using input-output pairs from public APIs. Once the surrogate mimics the target model, DDPA computes thrust vectors and crafts poisoned data. Since our approach guides model updates via data removal rather than relying on direct gradient manipulation, the poisoning strategy can transfer from the surrogate to the target model, making the attack applicable in black-box settings.
>
> **4. Intuitive explanation for some derivations (Theoretical Claims)**
>
> We include a concrete example to illustrate Theorems 3.1--3.4. Using a 2D regular simplex with centroid at the origin, we compute the inscribed radius as $r = \frac{\sqrt{3}}{6} \approx 0.2887$, with tangency points satisfying the barycentric condition (Theorem 3.1). The area from Theorem 3.2 yields $3\sqrt{3} \approx 5.196$, and the regularity measure in Theorem 3.3 evaluates to 1. In Theorem 3.4, equal edge lengths and centroid alignment yield an effective proportion $\rho = 0.866$.
>
> This example grounds the abstract results in a clear geometric case. See [ https://anonymous.4open.science/r/sup_materials/Numerical%20Example.pdf ] for visualization. For a more detailed explanation, please refer to our response to  **Reviewer X4cj Q3**.
>
> **5. The convex approximation approach is not always differentiable (Theoretical Claims)**
>
> The goal of our DDPA method is not to directly perform gradient-based optimization on the convex polyhedral envelope itself, but rather to identify thrust points (group centers) that lie in regions where the original non-convex loss is close to its convex approximation. Specifically, we optimize the objective in Eq.(17), which minimizes the maximum deviation between the original loss and its convex approximation over the convex polyhedron defined by the group centers. Since this objective involves selecting points rather than differentiating through the envelope, non-differentiability does not hinder the optimization in practice. Once the group centers are selected, the subsequent unlearning process operates directly on the original model and loss function, without relying on the convex envelope.
>
> **6. Algorithm 1 lacks a complexity analysis (Supplementary Material)**
> The complexity of Algorithm 1 comes from two stages: optimizing the group centers $V = \{v_1, \dots, v_n\}$ by minimizing $L_t + L_s$, and generating poisoned data. As defined in Eq. (17), computing $L_t$ involves solving a constrained optimization (Eq. 16) for each $x \in G$, with cost $\mathcal{O}(g p d)$ per iteration. The structural loss $L_s$ (Eq. 12) includes $\phi(V)$ (Eq. 10), pairwise distances, and the centroid term. These are computed in a subspace of dimension $\mathcal{O}(\log n / \varepsilon^2)$ using a Johnson--Lindenstrauss projection, with per-iteration cost $\mathcal{O}(n \log n / \varepsilon^2)$. Sampling $k$ points from $\mathcal{N}(v_i, \sigma^2)$ for each $v_i \in V$ costs $\mathcal{O}(n k d)$. The overall complexity is $\mathcal{O}(T g p d + T n \log n / \varepsilon^2 + n k d).$ The procedure is efficient in practice, as $n$ is small, optimization is one-time, and all structural terms are computed in a low-dimensional subspace.

---

> > ### Comment · Reviewer_twWJ · 2025-04-09
> >
> > All my concerns are addressed. Thus, I maintain my score.

---

### Official Review · Reviewer_TAX8 · 2025-03-13

**Overall Recommendation:** 4

**Summary:**

This work addresses adversarial attacks on machine unlearning (MU), focusing on target-agnostic attacks that can target arbitrary parameters upon request. It allows quick responses to multiple MU attack requests after deployment, while maintaining attack stability. First, the authors use a convex polyhedral approximation to identify points in the loss landscape where convexity approximately holds, ensuring stable attack performance. Second, a regular simplex detection technique is developed to maximize coverage of the parameter space, enhancing attack flexibility and efficiency.
The algorithm adjusts one or more thrust vectors to move parameters in any direction, responding promptly to MU attack requests as long as data removal stays within the MU budget, while ensuring attack stability through convex approximation and optimization.

## update after rebuttal
This paper investigates an interesting problem of target-agnostic attack. The method is novel, and thoroughly examined in the experiments. Thus, I keep my original rating of "accept".

**Claims And Evidence:**

Yes. The paper is well-written with a clear algorithm workflow, thorough theoretical analysis, and extensive experiments show that the proposed method addresses three key issues: inflexibility, inefficiency, and instability.

**Essential References Not Discussed:**

No

**Experimental Designs Or Analyses:**

The experimental criteria are reasonable, including datasets, baselines, and metrics. The paper evaluates different variants of the method and most of them achieve better results than baselines. The experiments in the paper are extensive and convincing. The ablation experiments show the efficacy of the new method.

**Methods And Evaluation Criteria:**

The method is novel, and thoroughly examined in the experiments. The experimental criteria, including datasets, baselines, and metrics, are well-chosen.

**Other Comments Or Suggestions:**

There are some careless expression errors. For example, “It ensuring the stability of MU attacks based on convex approximation and optimization” in line 130. Also, the algorithm name is ambiguous. The paper uses two versions of DDPA and FESA in different places.

**Other Strengths And Weaknesses:**

Strengths

1.The paper focuses on an emerging topic of adversarial attacks on machine unlearning, which is importance in many privacy-sensitive applications.

2.The idea of the thrust vector control in aerospace engineering is interesting.

3.Both theoretical analysis and experimental results are shown to justify the effectiveness of the proposed method.

Weaknesses

1.My major concern is the benchmark comparison. Authors claim it is the first algorithm to address the inflexibility and inefficiency issues in adversarial attacks on machine unlearning. I saw the efficiency experiments regarding the runtime of multiple attack requests. It would be nice to provide additional empirical results about flexibility.

2.Some parameter settings/choices in implementation can be better explained, for example, the group center and the unlearning rate.

3.In the Appendix, lines 2282 and 2283 overlap with each other. I suggest the authors to update the paper layout to get more clear presentation.

One minor presentation issue: In the experiment figures, the font size in the legend decreases the paper readability, e.g., Figures 2-5.

**Questions For Authors:**

Please refer to the above weaknesses.

**Relation To Broader Scientific Literature:**

This work enhances the performance of adversarial attacks on machine unlearning, by tackling three open critical challenges of flexibility, efficiency, and stability.

**Theoretical Claims:**

Theoretical analysis in this work seems to be complete and correct, including the solution of convex polyhedral approximation, the regularity property of the maximum regular simplex, and the coverage of the simplex to the parameter space.

---

> ### Author Rebuttal · Authors · 2025-04-01
>
> We would like to thank the reviewer for the helpful and constructive comments. We have tried our best to address your concerns. We will include all the analyses, duscussions, and experimental results in this rebuttal into the submission. Due to the limit of 5000 characters, if anything remains unclear, post-rebuttal comments are appreciated.
>
> **1. Empirical results about flexibility (Other Strengths And Weaknesses)**
>
> We have included the flexibility test in Figure 5 in Page 8 in the main paper. We compare DDPA with all baselines under a target-agnostic setting (unknown attack target during poisoning). While other methods suffer significant ASR drops as the number of potential targets increases, DDPA maintains consistently high ASR. For example, DDPA achieves the highest ASR $91.6\%$, whereas the baselines obtain the ASR $6.3\%$. These results clearly demonstrate DDPA's flexibility in adapting to different attack targets during unlearning.
>
> **2. Parameter settings/choices (Other Strengths And Weaknesses)**
>
> Based on our observations, increasing the number of group centers (from 5 to 25) leads to higher ASR across all datasets, as it allows for more fine-grained control over parameter manipulation. Longer training epochs (e.g., from 30 to 150) significantly improve ASR, since models trained for more epochs become more sensitive to unlearning. For image classification tasks, higher learning rates help the model converge to exploitable optima more quickly, resulting in increased ASR. However, in large-scale models such as LLaMA-3b, overly high learning rates can destabilize optimization and reduce ASR. Higher unlearning rates also tend to improve ASR by amplifying the model’s sensitivity to data removal, though excessive rates may negatively affect accuracy. Similarly, increasing the unlearning ratio (e.g., from 5\% to 20\%) intensifies parameter disturbance, leading to further gains in ASR. The parameter settings are detailed in our response to **Reviewer X4cj Q4**.
>
> **3. Overlap with each other, font size, and careless expression errors (Other Strengths And Weaknesses) (Other Comments Or Suggestions:)**
>
> We will correct the overlapping lines in the appendix, unify terminology for algorithm names, increase font size, fix expression errors, and improve figure readability and formatting for clarity.
>
> **4. real-word scenarios**
>
> This work focuses on exploring the vulnerability of machine unlearning (MU) models under adversarial attacks, rather than using MU techniques to attack standard machine learning (ML) models.
>
> In line with modern privacy regulations such as the EU’s GDPR and California’s CCPA, MU allows data holders to remove the influence of specific data points from a trained ML model. For example, Stable Diffusion 3.0 allows artists to remove their artwork in the training data, responding to the 'Right to be Forgotten' legislation by GDPR and attempting to respect artists' works [1]. However, recent studies [2–10] have shown that MU is vulnerable to malicious unlearning requests in adversarial settings. In such attacks, the adversary first injects carefully crafted samples that help maintain the model’s performance and remain indistinguishable from clean data. Once the model is trained, the attacker submits an MU request to remove those specific samples, which degrades the model’s performance only after unlearning.
>
> The goal of our attack is to preserve the performance of the original ML model while impairing the MU model after the unlearning process. This objective is fundamentally different from that of traditional backdoor attacks.
>
> [1] Stability AI will let artists opt out of Stable Diffusion 3.0 training. Ars Technica.
>
> [2] Static and Sequential Malicious Attacks in the Context of Selective Forgetting. NeurIPS 2023.
>
> [3] Releasing Malevolence from Benevolence: The Menace of Benign Data on Machine Unlearning. CoRR 2024.
>
> [4] Hidden Poison: Machine Unlearning Enables Camouflaged Poisoning Attacks, NeurIPS 2023.
>
> [5] Backdoor attacks via machine unlearning, AAAI 2024.
>
> [6] Exploiting machine unlearning for backdoor attacks in deep learning system, arxiv 2023.
>
> [7] UBA-Inf: Unlearning activated backdoor attack with Influence-Driven camouflage, USENIX 2024.
>
> [8] Unlearn to relearn backdoors: Deferred backdoor functionality attacks on deep learning models, arxiv 2024.
>
> [9] Rethinking adversarial robustness in the context of the right to be forgotten, ICML 2024.
>
> [10] A duty to forget, a right to be assured? exposing vulnerabilities in machine unlearning services, arxiv 2024.

---

> > ### Comment · Reviewer_TAX8 · 2025-04-05
> >
> > Thanks for the response and additional results. They answered my questions. I am happy to recommend the acceptance.

---

> > > ### Author Response · Authors · 2025-04-06
> > >
> > > We sincerely thank the reviewer for the thoughtful feedback and for recommending acceptance. We're glad that our additional results and clarifications addressed your concerns. We appreciate the opportunity to highlight our contribution of a novel, target-agnostic attack framework that achieves flexible, efficient, and stable adversarial attacks on machine unlearning systems.

---

### Official Review · Reviewer_qxUp · 2025-03-13

**Overall Recommendation:** 5

**Summary:**

This paper proposes a novel adversarial attack framework for machine unlearning (MU) called Dynamic Delayed Poisoning Attack (DDPA). The key contributions include:
1. Target-agnostic attack flexibility: By leveraging thrust vector control (from aerospace engineering) and simplex geometry, DDPA dynamically manipulates model parameters to handle arbitrary attack targets during unlearning.
2. Stability via convex polyhedral approximation: A convex approximation method is introduced to stabilize parameter updates in non-convex loss landscapes.
3. Efficiency in multi-target attacks: A regular simplex detection technique maximizes parameter space coverage, enabling efficient handling of sequential unlearning requests.

**Claims And Evidence:**

- Claim 1: DDPA supports target-agnostic attacks.
  - Evidence: Figure 5 shows DDPA maintains high ASR (91.6% untargeted) even when attack targets are unknown during poisoning, while baselines degrade with increasing target classes.
  - Issue: The claim assumes attackers can freely manipulate unlearning requests post-deployment, but real-world scenarios (e.g., audit mechanisms) are not discussed.

- Claim 2: Convex polyhedral approximation mitigates instability.
  - Evidence: Ablation studies (Figure 4) show DDPA outperforms DDPA-C/S variants in ASR and BA.
  - Issue: No direct comparison of optimization trajectories (e.g., loss landscape visualization) is provided to validate stability improvements.

- Claim 3: Simplex coverage maximizes attack efficiency.
  - Evidence: Theorem 3.4 derives the parameter space coverage ratio.
  - Issue: The Gaussian assumption for parameter space is not empirically validated (e.g., via distribution tests).

**Essential References Not Discussed:**

Defensive Unlearning with Adversarial Training for Robust Concept Erasure in Diffusion Models: [Zhang et al., NeurIPS 2024] introduced defenses against unlearning attacks, critical for evaluating DDPA’s practicality.

**Experimental Designs Or Analyses:**

- Strengths:
  - Cross-dataset validation (images + NLP) and comparison with nine SOTA methods (e.g., AwoP, MUECPA) enhance credibility.
  - Ablation studies (DDPA-C/S) effectively isolate contributions of simplex and convex approximation modules.
- Weaknesses:
  - Hyperparameter Sensitivity: Training details (e.g., learning rates, epochs) are listed in Appendix F but lack justification.

**Methods And Evaluation Criteria:**

- Methods:
  - Strength: The integration of thrust vector control and simplex geometry is novel, enabling dynamic parameter manipulation.
  - Weakness: The simplex detection process (Equation 11-12) lacks algorithmic complexity analysis, raising scalability concerns for high-dimensional models.

- Evaluation Criteria:
  - Strength: ASR, TA (targeted accuracy), and BA (benign accuracy) are standard metrics for poisoning attacks.
  - Weakness: No evaluation of attack stealthiness (e.g., trigger invisibility) or computational cost (e.g., training time vs. baselines).

**Other Comments Or Suggestions:**

- Typos:
  - Page 3: “stPS” → “steps”.
  - Table 1: Column headers misaligned.
- Writing: The threat model (Section 3.1) should explicitly state whether attackers can modify *retained* data $ D_r $.

**Other Strengths And Weaknesses:**

- Originality: The thrust vector control analogy is creative, though theoretical novelty is limited to combining existing geometric tools.
- Significance: Highlights critical vulnerabilities in MU systems, urging the community to prioritize robustness.
- Clarity: The John ellipsoid discussion (Section 3.2) is overly technical and distracts from the core contribution.

**Questions For Authors:**

1. How does DDPA’s training time (for poisoned dataset generation) compare to baselines like AwoP or MUECPA? If slower, does the multi-attack efficiency justify the overhead?


2. How valid is the Gaussian assumption (Theorem 3.4) for deep models? Did you test parameter distributions (e.g., via KL divergence) on trained VGG16/ResNet-18?


3. How does DDPA perform against MU systems with gradient masking or anomaly detection?

**Relation To Broader Scientific Literature:**

- The thrust vector control analogy is novel but under-explored; connections to control theory (e.g., Lyapunov stability) could deepen theoretical grounding.

**Theoretical Claims:**

- Theorem 3.4 (Simplex coverage ratio):
  - The proof assumes parameters follow a Gaussian distribution. While common in ML, this is not empirically verified for the tested models (e.g., VGG16/ResNet-18).
  - Critical Issue: The edge length $ l $ in Equation 13 is undefined in practice. How $ l $ is estimated for real models remains unclear.

- Theorem 3.1-3.3 (John ellipsoid properties):
  - Proofs in Appendix C rely on standard convex geometry but lack direct connection to MU-specific dynamics (e.g., how MU gradients interact with simplex vertices).

---

> ### Author Rebuttal · Authors · 2025-04-01
>
> We would like to thank the reviewer for the helpful and constructive comments. We have tried our best to address your concerns. We will include all the analyses, duscussions, and experimental results in this rebuttal into the submission. Due to the limit of 5000 characters, if anything remains unclear, post-rebuttal comments are appreciated.
>
> **1. Loss landscape visualization (Claims And Evidence)**
>
> In DDPA, thrust vector control is achieved by selecting group centers near locally convex regions of the non-convex loss, identified via a convex polyhedral approximation that minimizes the gap to the original loss. Figure 1 (https://anonymous.4open.science/r/sup_materials/landscape.pdf) visualizes the loss landscape using a shared 2D projection. The region around the selected anchor is visibly smoother and more stable, showing that DDPA effectively locates well-conditioned areas for stable thrust vector updates.
>
> **2. Gaussian assumption for parameter space (Claims And Evidence)**
>
> We validate the Gaussian assumption using the Shapiro–Wilk, D’Agostino’s $K^2$, and Anderson–Darling tests. For VGG16 on CIFAR-100, the $p$-values are 0.55669 (Shapiro–Wilk) and 0.30275 ($K^2$), with an Anderson–Darling statistic of 0.24373. For ResNet18 on Tiny ImageNet, the values are 0.67146, 0.35562, and 0.35510, respectively. All results fall within standard thresholds for normality: $p > 0.05$ and Anderson–Darling statistic < 0.787. These consistent results across models and datasets support the Gaussian assumption.
>
> **3. No evaluation of attack stealthiness (Methods And Evaluation Criteria) and Essentail References**
>
> Zhang et al. focus on preventing recovery after unlearning, which is orthogonal to our setting. To assess stealthiness, we report anomaly indices from three standard detectors: Neural Cleanse (NC), Perceptual-Based (PB), and AEVA. These methods assign class-wise anomaly scores, with values below –2.0 commonly indicating a backdoor. As shown in Table 3 (https://anonymous.4open.science/r/sup_materials/stealthiness_evaluation_table.pdf), our method achieves scores of 0.322 (NC), 0.29 (PB), and 0.59 (AEVA), all above the detection threshold and generally higher than those of other methods. This indicates that our attack does not leave strong or easily detectable backdoor signatures.
>
> **4. No evaluation of computational cost (Methods And Evaluation Criteria)**
>
> We provide a detailed efficiency comparison of DDPA and all baselines, including AwoP and MUECPA, in Figure 2 (main paper) and Figures 6, 8, and 10 (Appendix F.1). DDPA significantly reduces total training time by generating a single poisoned dataset that supports multiple attack targets. In contrast, baselines such as AwoP and MUECPA require regenerating a new poisoned dataset for each target, leading to much higher cumulative cost. Additional timing details for the poisoning, training, and unlearning stages are discussed in our response to Reviewer twWJ Q2.
>
> **5. The edge length in Equation 13 (Theoretical Claims)**
>
> In practice, we estimate the edge length $l$ of the simplex by computing the average Euclidean distance between all pairs of group centers: $l = \frac{2}{n(n-1)}\sum_{1 \leq i < j \leq n}\|v_i - v_j\|_2,$
> where $\\\\\\{v_1, v_2, \ldots, v_n\\\\\\}$ are the $n$ group centers used in the simplex construction. This provides a natural measure of the simplex’s spatial scale in the parameter space and is easy to compute.
>
> **6. MU gradients interact with simplex vertices (Theoretical Claims)**
>
> As detailed in Section 3.2 (lines 240–295), we sample group centers from clean data and use a convex polyhedral approximation to identify locally convex regions. Centers aligning with this structure are selected as thrust points and mapped to thrust vectors via a conjugate algorithm, forming a regular simplex in parameter space. During unlearning, removal gradients move the model along these simplex directions, enabling controlled updates and ensuring explicit interaction between MU gradients and the simplex.
>
> **7. Attackers can modify retained data (Other Comments Or Suggestions)**
>
> Our threat model only assume the attacker can inject malicious samples with no ability to alter retained training data.
>
> **8. Response to Concerns on Attack Assumptions, Algorithmic Complexity, Numerical Examples and Hyperparameter Sensitivity**
>
> Regarding the assumption that attackers can manipulate unlearning requests post-deployment, Algorithmic Complexity, Numerical Examples and Hyperparameter Sensitivity please see our detailed response to **Reviewer X4cj Q2,Q4** and **twWJ Q4, Q6** and **Reviewer TAX8 Q4** respectively.
>
> **9. Reproducibility**
>
> Due to the link restrictions imposed by the ICML rebuttal policy (only figures and tables), we will release our code on GitHub along with a project page and full documentation once the paper is accepted.

---

### Official Review · Reviewer_X4cj · 2025-03-14

**Overall Recommendation:** 3

**Summary:**

This paper introduces a new adversarial attack framework specifically designed for machine unlearning systems. The central contribution is the Dynamic Delayed Poisoning Attack (DDPA) method, which addresses the limitations of previous approaches by being target-agnostic, enabling efficient handling of multiple attack requests, and exhibiting robustness in the face of non-convex loss landscapes. The method leverages convex polyhedral approximation together with simplex geometry to create “thrust vectors” that effectively guide model parameters toward any desired target. Extensive empirical evaluations across several datasets validate the effectiveness of the proposed attack.

**Claims And Evidence:**

The authors assert that DDPA achieves superior attack success rates, offers enhanced flexibility due to its target-agnostic design, and improves computational efficiency when compared to existing methods. While the experimental results - featuring comparisons with nine baseline techniques- largely support these claims, the evidence would be more convincing if additional ablation studies were provided to clearly isolate the contributions of the convex approximation and simplex components.

**Essential References Not Discussed:**

Although the literature review is comprehensive, the paper could be further improved by citing more recent studies on adaptive adversarial attacks, which would provide additional context and support for its contributions.

**Experimental Designs Or Analyses:**

The experimental analysis would benefit from a more detailed description of the hyperparameter settings, training procedures, and statistical significance measures (for example, error bars). Such additional details would strengthen the overall robustness and reproducibility of the experimental conclusions.

**Methods And Evaluation Criteria:**

A key question raised is the necessity of the unlearning process in the attack. Specifically, if the goal is to poison the model, one might simply insert backdoor examples directly. The paper should clarify why incorporating the unlearning step provides additional benefits over conventional poisoning techniques. It is also strange to inject data in the middle of the training. This will be easy to detect.

**Other Comments Or Suggestions:**

None.

**Other Strengths And Weaknesses:**

Strengths: Presents an innovative blend of methods from convex analysis, simplex geometry, and control theory.

Weaknesses: 1. Some of the theoretical proofs and derivations are overly detailed in the main text, which may obscure the central contributions. 2. The approach appears to depend on certain assumptions regarding the behavior of the loss function, potentially limiting its applicability across different domains.

**Questions For Authors:**

None.

**Relation To Broader Scientific Literature:**

The paper is well-situated within the existing body of research on machine unlearning and adversarial attacks. It builds upon and extends prior work in these areas, effectively positioning its contributions within the broader scientific literature.

**Theoretical Claims:**

The paper presents several complex theoretical results concerning the properties of regular simplices and the behavior of the convex polyhedral approximation. Due to the intricate nature of these proofs and my limited grasp of some of the details, I am currently unable to fully assess the correctness of these theoretical contributions.

---

> ### Author Rebuttal · Authors · 2025-04-01
>
> We would like to thank the reviewer for the helpful and constructive comments. We have tried our best to address your concerns. We will include all the analyses, duscussions, and experimental results in this rebuttal into the submission. Due to the limit of 5000 characters, if anything remains unclear, post-rebuttal comments are appreciated.
>
> **1. Ablation studies (Claims And Evidence)**
>
> We have included the ablation study in Figure 4 in Page 8 in the main paper and Tables 3-19 in Appendix F. DDPA-S utilizes only the simplex method to maximize and generate an effective operational space. DDPA-C employs only Convex Polyhedral Approximation to ensure stability in constructing thrust vectors (group centers). DDPA operates with the full support of both simplex method and convex polyhedral approximation. We have observed that the full DDPA method achieves the lowest BA and the highest ASR and TA in most experiments, consistently outperforming other versions and highlighting the advantages of simplex method and convex polyhedral approximation.
>
> **2. Necessity of the unlearning process in the attack (Methods And Evaluation Criteria)**
>
> This work focuses on exploring the vulnerability of machine unlearning (MU) models under adversarial attacks, rather than using MU techniques to attack standard machine learning (ML) models. The goal is to preserve the model's performance before MU while degrading its performance after MU, which is quite different from traditional backdoor attacks.
>
> Traditional backdoor attacks require control over the training process and predefine the attack target, making them inflexible and inefficient. In contrast, our method exploits MU vulnerabilities without access to training and operates at the unlearning stage. The attack target can be determined or modified after training or during unlearning.
>
> Injecting data during training is common in real-world settings such as online learning [1] and continual learning [2], where models are incrementally updated. For practical examples, please refer to our response to **Reviewer TAX8 Q4**.
>
> As noted in our response to **Reviewer qxUp Q3**, our attack remains stealthy and effective under strong detection defenses, underscoring its practical impact.
>
> [1] Online learning: A comprehensive survey,  ACM, 2021.
>
> [2] A comprehensive survey of continual learning: Theory, method and application, 2024.
>
> **3. Unable to fully assess the correctness of theoretical contributions (Theoretical Claims)**
>
> We have explained the physical meaning of these theorems in lines 179-192 in page 4 and lines 282-295 in page 6 in the submission. In addition, we have included the detailed proof of all theorems in pages 17-26 in Appendix C.
>
> **4. More detailed description of the hyperparameter settings, training procedures, and statistical significance measures. (Experimental Designs Or Analyses)**
>
> We have included the hyperparameter setting, implementation details, and training procedures in lines 306-317 on page 5 in the submission and in lines 1,442-1,480 and Table 2 in Appendix F. In addition, we have provided the details of poisoned dataset generation in Appendix D. For further information, please refer to our response to **TAX8 Q2**.
>
> **5. Recent studies on adaptive adversarial attacks (Essential References Not Discussed)**
>
> Adaptive adversarial attacks refer to attacks that are specifically designed to counteract or bypass known defense mechanisms [3,4]. Compared with standard adversarial attacks, adaptive adversarial attacks provide a more rigorous and realistic evaluation of defense methods, as they are crafted with knowledge of the defense and specifically designed to exploit its weaknesses. We will include the recent  related works into the submission.
>
> [3] On Adaptive Attacks to Adversarial Example Defenses. NeurIPS 2020.
>
> [4]  Adversarial examples are not bugs, they are features. NeurIPS, 2019.
>
> **6. Assumptions regarding the behavior of the loss function (Other Strengths And Weaknesses)**
>
>  Our method does not rely on strong or global assumptions about the loss function. While our convex polyhedral approximation (Section 3.3) utilizes local properties of the loss surface, we do not assume that the entire loss function is convex. Instead, our approach identifies regions where the loss can be locally approximated by a convex function, and places group centers (thrust vectors) in those areas. This is achieved through the optimization objective in Eq.(17), which selects data samples whose loss behavior deviates minimally from convexity, ensuring stability during the MU. Moreover, our experiments span multiple domains, including image and text classification, demonstrating that the method remains effective across a variety of non-convex loss functions in practice. We would be glad to discuss extensions of our approach to even more complex domains in future work.

---

### Decision · Program_Chairs · 2025-05-01

**Decision:**

Accept (poster)

**Comment:**

Overall, reviewers agreed that the approach was novel, and the experimental concerns initially brought up were largely addressed during the discussion period. The authors are strongly encouraged to integrate these additional validations on the from the discussion period into the paper. While reviewers acknowledged that the stricter threat model and the complexity of the framework may limit its wider adoption, the same reviewers also felt that this wasn't an outlier since other papers make similarly strong assumptions, and could be useful to know as a potential vulnerability. The studied concept of adding poisoned data that only triggers upon unlearning, even if impractical, could be of broader intellectual interest.